# Federated Learning under Periodic Client Participation and Heterogeneous Data: A New Communication-Efficient Algorithm and Analysis

**Michael Crawshaw**
Department of Computer Science
George Mason University
Fairfax, VA 22030
mcrawsha@gmu.edu

**Mingrui Liu**
Department of Computer Science
George Mason University
Fairfax, VA 22030
mingruil@gmu.edu

## Abstract

In federated learning, it is common to assume that clients are always available to participate in training, which may not be feasible with user devices in practice. Recent works analyze federated learning under more realistic participation patterns, such as cyclic client availability or arbitrary participation. However, all such works either require strong assumptions (e.g., all clients participate almost surely within a bounded window), do not achieve linear speedup and reduced communication rounds, or are not applicable in the general non-convex setting. In this work, we focus on nonconvex optimization and consider participation patterns in which the chance of participation over a fixed window of rounds is equal among all clients, which includes cyclic client availability as a special case. Under this setting, we propose a new algorithm, named Amplified SCAFFOLD, and prove that it achieves linear speedup, reduced communication, and resilience to data heterogeneity simultaneously. In particular, for cyclic participation, our algorithm is proved to enjoy $\mathcal{O}(\epsilon^{-2})$ communication rounds to find an $\epsilon$-stationary point in the non-convex stochastic setting. In contrast, the prior work under the same setting requires $\mathcal{O}(\kappa^2 \epsilon^{-4})$ communication rounds, where $\kappa$ denotes the data heterogeneity. Therefore, our algorithm significantly reduces communication rounds due to better dependency in terms of $\epsilon$ and $\kappa$. Our analysis relies on a fine-grained treatment of the nested dependence between client participation and errors in the control variates, which results in tighter guarantees than previous work. We also provide experimental results with (1) synthetic data and (2) real-world data with a large number of clients ($N = 250$), demonstrating the effectiveness of our algorithm under periodic client participation.

## 1 Introduction

Federated learning (FL) [27, 21, 16, 53] is a distributed learning paradigm that emphasizes client privacy [28, 41, 29], limited communication [19, 25], and data heterogeneity across clients [17, 55]. FL has attracted attention in recent years due to the ability to leverage data and compute from user devices while respecting privacy [49, 9]. For large-scale FL, it is common to limit the number of simultaneously participating devices, and many works do so by assuming that a random subset of clients can be sampled independently at each round [27, 3, 50, 22, 38]. However, this pattern of client participation is not always practical. If clients are user devices like mobile phones, they may not have 24/7 availability due to low battery or bad internet connection [15, 31]. In particular, if client availability is correlated with geographical location (e.g. mobile phones charging at night), then

38th Conference on Neural Information Processing Systems (NeurIPS 2024).

Table 1: Communication and computation complexity of various methods to find an $\epsilon$-stationary point for $L$-smooth, non-convex objectives. $N$: number of clients, $\kappa$: data heterogeneity $\sup_{\boldsymbol{x}} \|\nabla f_i(\boldsymbol{x}) - \nabla f(\boldsymbol{x})\| \leq \kappa$. $S$: number of participating clients per round, $\bar{K}$: number of groups for cyclic participation. See Section 3.2 for a description of each participation pattern. We say that an algorithm exhibits reduced communication if its dependence in terms of $\epsilon$ is strictly smaller than $\mathcal{O}(\epsilon^{-4})$. Derivation of complexities for Amplified FedAvg can be found in Appendix C.

| Setting | Communication Complexity ($R$) | Iteration Complexity ($RI$) | Reduced Communication | Unaffected by Heterogeneity |
|---|---|---|---|---|
| i.i.d. Participation ($S$) | | | | |
| FedAvg [17] | $\frac{\Delta \kappa^2}{S\epsilon^4}\left(1-\frac{S}{N}\right) + \frac{\sqrt{L}\kappa}{\epsilon^3}$ | $\frac{\Delta L \sigma^2}{S\epsilon^4}$ | ✗ | ✗ |
| SCAFFOLD [17] | $\frac{\Delta L}{\epsilon^2}\left(\frac{N}{S}\right)^{2/3}$ | $\frac{\Delta L \sigma^2}{S\epsilon^4}$ | ✓ | ✓ |
| Amplified FedAvg [39] | $\frac{\Delta L N \kappa^2}{S\epsilon^4}\left(1-\frac{S}{N}\right) + \frac{\Delta L + \kappa^2}{\epsilon^2}$ | $\frac{\Delta L \sigma^2}{S\epsilon^4}$ | ✗ | ✗ |
| Amplified SCAFFOLD (ours) | $\frac{\Delta L}{\epsilon^2}\frac{N}{S}$ | $\frac{\Delta L \sigma^2}{S\epsilon^4}$ | ✓ | ✓ |
| Regularized Participation ($P, \rho$) | | | | |
| Amplified FedAvg [39] | $\frac{\Delta L P + \kappa^2 + \sigma^2}{\epsilon^2}$ | $\frac{\Delta L \rho^2 \sigma^2}{\epsilon^4}$ | ✓ | ✗ |
| Amplified SCAFFOLD (ours) | $\frac{\Delta L P}{\epsilon^2}$ | $\frac{\Delta L \rho^2 \sigma^2}{\epsilon^4}$ | ✓ | ✓ |
| Cyclic Participation ($\bar{K}, S$) | | | | |
| Amplified FedAvg [39] | $\frac{\Delta L N \kappa^2}{S\epsilon^4}\left(1-\frac{S\bar{K}}{N}\right) + \frac{\Delta L \bar{K} + \kappa^2}{\epsilon^2}$ | $\frac{\Delta L \sigma^2}{S\epsilon^4}$ | ✗ | ✗ |
| Amplified SCAFFOLD (ours) | $\frac{\Delta L \bar{K}}{\epsilon^2}\frac{N}{S}$ | $\frac{\Delta L \sigma^2}{S\epsilon^4}$ | ✓ | ✓ |

client availability follows a cyclic pattern [54]. Therefore, it remains an important open question to design federated optimization algorithms with provable efficiency under non-i.i.d client participation.

Several works have investigated optimization in FL under non-i.i.d. client participation [10, 2, 8, 39, 46]. However, to the best of our knowledge, no existing algorithm in a non-i.i.d. participation setting provably exhibits reduced communication cost, linear speedup with respect to the number of clients, and resilience to client data heterogeneity for general non-convex optimization.

In this work, we consider FL under an arbitrary participation framework [39], where client participation during each round is a random variable with potentially unknown distribution. We focus on client participation patterns that are periodic, in the sense that all clients are expected to participate with equal frequency over a window of multiple training rounds.

For this setting, we propose Amplified SCAFFOLD, an optimization algorithm for FL under periodic client participation. Amplified SCAFFOLD utilizes (a) amplified updates across participation periods and (b) control variates computed across entire participation periods, to eliminate the effect of data heterogeneity even under non-i.i.d. participation. We show that Amplified SCAFFOLD exhibits significantly reduced communication cost, linear speedup, and is unaffected by client data heterogeneity. To the best of our knowledge, this is the first result demonstrating reduced communication or resilience to data heterogeneity without assuming i.i.d. participation. The complexity of Amplified SCAFFOLD is compared against baselines in Table 1. For cyclic participation, Amplified SCAFFOLD improves the previous best communication cost from $\mathcal{O}(\kappa^2\epsilon^{-4})$ to $\mathcal{O}(\epsilon^{-2})$.

The main challenges of achieving these properties are (1) simultaneously handling randomness from stochastic gradients and non-i.i.d. participation; and (2) controlling the error of control variates under non-i.i.d. participation. Previous work in this setting [39] performs an in-expectation analysis, by taking expectation only over randomness from stochastic gradients; this avoids (1) but cannot leverage properties of the participation pattern to reduce communication. We present a tighter analysis that addresses (1) by taking expectation over both client participation and the stochastic gradients throughout the analysis, and carefully treating the trajectory variables which depend on both sources of randomness. We address (2) by recursively bounding the control variate errors, which involves a non-uniform average of non-uniform averages of error terms resulting from non-i.i.d. participation. We show that this nested non-uniform average can be bounded using mild regularity conditions on the participation pattern.

Our contributions are summarized below.

- We introduce Amplified SCAFFOLD, an optimization algorithm for federated learning under non-i.i.d. client participation. Our convergence analysis demonstrates its computational and communication efficiency: Amplified SCAFFOLD exhibits reduced communication, linear speedup, and is unaffected by data heterogeneity. These guarantees are achieved with a tighter analysis than used in previous work [39], with a fine-grained treatment of the two sources of randomness: client participation and stochastic gradients. In the case of cyclic participation, we reduce the previous best communication cost of $\mathcal{O}(\kappa^2\epsilon^{-4})$ to $\mathcal{O}(\epsilon^{-2})$.

- Experimental results show that Amplified SCAFFOLD converges faster than baselines on both synthetic and real-world problems under realistic non-i.i.d. client participation patterns. We also include an ablation study which demonstrates the robustness of our algorithm to changes in data heterogeneity, the number of participating clients per round, and the number of client groups in cyclic participation.

The paper is outlined as follows. We discuss related work in Section 2, and Section 3 provides a formal specification of the optimization problem. Amplified SCAFFOLD is introduced and theoretically analyzed in Section 4, and we provide experiments in Section 5. We conclude with Section 6.

## 2 Related Work

**Federated Optimization.** FedAvg [27] characterizes partial client participation and local updates in each round. FedAvg was analyzed in the full participation setting [35, 37, 50, 51, 43, 42, 18, 12]. Other federated optimization algorithms aim to improve communication efficiency [32, 52] and tackle data heterogeneity [22, 17]. The analysis of FL optimization algorithms typically either assumes full client participation or partial client participation where clients are sampled uniformly randomly [47, 38, 17, 23, 43, 1]. [30] provides lower bounds for distributed stochastic, smooth optimization with intermittent communication and non-convex objectives, both in the full and partial participation settings. They also include algorithms employing variance reduction which match (or closely match) lower bounds in the full and partial participation settings. However, none of the works above are applicable for general participation patterns such as periodic participation.

**Client Participation.** Cyclic data sampling was considered for stochastic convex optimization in [10], where they propose "pluralistic" solutions instead of learning a single model for all clients. There is a recent line of work considering various participation patterns, including client selection [11, 4, 33] biased participation [34, 6, 7], independent participation across rounds [17, 22, 23], unbiased participation [36, 13], bounded rounds of unavailability [46, 14, 48], asynchronous participation [24, 2], cyclic participation [8], and arbitrary participation [39, 40]. However, none of these works enjoy linear speedup, reduced communication rounds, and resilience to data heterogeneity under the general setting of non-convex objectives and periodic client participation. Concurrent work [44] considers non-stationary client participation under the condition that for every client and every round, the probability of participation is bounded away from zero; for this setting, they propose an algorithm with linear speedup and dependence on gradient dissimilarity for non-convex, Lipschitz objectives.

See Appendix F for a detailed discussion comparing our results with a small number of closely related baselines.

## 3 Problem Setup

We consider a federated learning problem with $N$ clients, with the overall objective

$$\min_{\boldsymbol{x}\in\mathbb{R}^d}\left\{f(x):=\frac{1}{N}\sum_{i=1}^{N}f_i(x)\right\},$$

where each $f_i:\mathbb{R}^d\to\mathbb{R}$ is the local objective of one client. We consider the stochastic optimization problem, so that $f_i(\boldsymbol{x})=\mathbb{E}_{\xi\sim\mathcal{D}_i}[F(\boldsymbol{x};\xi)]$, and the optimization algorithm can access $F_i(\boldsymbol{x};\xi)$ and $\nabla F_i(\boldsymbol{x};\xi)$ for individual values of $\xi$. We make the following assumptions about the objectives:

**Assumption 1.** *(a)* $f(\boldsymbol{x}_0)-\min_{\boldsymbol{x}\in\mathbb{R}^d}f(\boldsymbol{x})\leq\Delta$. *(b) Each $f_i$ is $L$-smooth, i.e.,* $\|\nabla f_i(\boldsymbol{x})-\nabla f_i(\boldsymbol{y})\|\leq L\|\boldsymbol{x}-\boldsymbol{y}\|$ *for all* $\boldsymbol{x},\boldsymbol{y}\in\mathbb{R}^d$. *(c) The stochastic gradient has variance $\sigma^2$, i.e.,* $\mathbb{E}_{\xi\sim\mathcal{D}_i}[\|\nabla F_i(\boldsymbol{x};\xi)-\nabla f_i(\boldsymbol{x})\|^2]\leq\sigma^2$ *for all* $\boldsymbol{x}\in\mathbb{R}^d$.

Since each $f_i$ may be non-convex, we consider the problem of finding an $\epsilon$-stationary point of $f$, that is, a point $\boldsymbol{x} \in \mathbb{R}^d$ such that $\|\nabla f(\boldsymbol{x})\| \leq \epsilon$.

## 3.1 Participation Framework

We consider a federated learning framework consisting of $R$ rounds. For any round $r \in \{0, \ldots, R-1\}$ and client $i \in [N]$, the availability of client $i$ at round $r$ is a random variable $q_r^i$, following the arbitrary participation framework of [39]. If $q_r^i = 0$, then client $i$ may not participate during round $i$. For example, under the conventional i.i.d. sampling of clients, at each round $r$ a subset of clients $\mathcal{S}_r \subset [N]$ is sampled uniformly without replacement, and the weights are set as $q_r^i = \frac{\mathbb{1}\{i \in \mathcal{S}_r\}}{S}$.

For some $P \in \mathbb{N}$, let $\mathcal{Q}_{r_0}$ be the filtration generated by $\{q_r^i : r_0 \leq r < r_0 + P, i \in [N]\}$, let $\mathcal{Q}$ be the filtration generated by $\mathcal{Q}_0, \ldots, \mathcal{Q}_{R-P}$, and let $\mathcal{G}$ be the filtration generated by $\{\xi_{r,k}^i : 0 \leq r < R, 0 \leq k < I, i \in [N]\}$, where $\xi_{r,k}^i$ is the random sampling of the stochastic gradient of round $r$, step $k$, client $i$. We make the following assumptions about the participation distribution.

**Assumption 2.** *For all* $r \in \{0, \ldots, R - 1\}$: *(a)* $\sum_{i=1}^N q_r^i = 1$ *and* $\sum_{i=1}^N (q_r^i)^2 \leq \rho^2$. *(b) The distribution of* $\{q_r^i\}$ *is unbiased across clients over every window of* $P$ *rounds, i.e.,* $\mathbb{E}_{\mathcal{Q}_{r_0}}[\frac{1}{P} \sum_{r=mP}^{(m+1)P-1} q_r^i] = 1/N$ *for every* $m < R/P$ *and* $i \in [N]$. *(c) Each client has a non-zero probability of being sampled over every window of* $P$ *rounds, i.e.,* $\mathbb{P}_{\mathcal{Q}_{r_0}}(\frac{1}{P} \sum_{r=mP}^{(m+1)P-1} q_r^i > 0) > p_{sample}$ *for every* $m < R/P$ *and* $i \in [N]$. *(d)* $\mathcal{Q}$ *and* $\mathcal{G}$ *are independent.*

For each round, Assumption 2(a) enforces that the participation weights $q_r^i$ are normalized to sum to 1, and characterizes the spread of participation weights across clients with the constant $\rho^2$. Assumption 2(b) requires that the set of rounds can be partitioned into windows of length $P$ within which clients are expected to participate with equal frequency. Lastly, Assumption 2(c) enforces that within each window, for each client the probability of being sampled is nonzero. Conventional i.i.d client sampling satisfies Assumption 2 with $\rho = S^{-1/2}, P = 1$, and $p_{sample} = S/N$.

An important difference from conventional i.i.d. participation is that here, client participation is not necessarily independent across rounds. Accordingly, we emphasize that the expectation and probability in Assumptions 2(b)-(c) are taken only over $\mathcal{Q}_{r_0}$. Therefore, the mean participation weight in Assumption 2(b) may itself be a random variable if client participation at some rounds is dependent on the outcome of participation in previous rounds. Similarly, the sampling probability in Assumption 2(c) may be a random variable. For the participation patterns considered in the next section, Assumption 2 is satisfied even when client sampling is not independent across rounds.

## 3.2 Specific Participation Patterns

**Regularized Participation** We say that client participation is *regularized* [39] if $\bar{q}_{r_0}^i = \frac{1}{N}$ almost surely for all $r_0$ and $i$, where $\bar{q}_{r_0}^i$ is defined on Line 18 of Algorithm 1 as the participation of client $i$ averaged over rounds $r_0, \ldots, r_0 + P - 1$. In this case, Assumption 2 is satisfied with $p_{sample} = 1$, while $P$ and $\rho^2$ are parameters of the participation pattern. Regularized participation is a relatively strong constraint, since every client must participate within each window, which may not be practical. However, it is flexible in that there is no constraint on how clients participate within each window. Regularized participation was also considered for strongly convex objectives [26].

**Cyclic Participation** Following the CyCP framework [8], $N$ clients are partitioned into $\bar{K}$ equally sized subsets, and at round $r$ only clients in group $(r \bmod \bar{K})$ may participate. $S$ clients are sampled without replacement from group $(r \bmod \bar{K})$, for whom the participation weight is $q_r^i = 1/S$. All other clients are assigned $q_r^i = 0$. Cyclic participation satisfies Assumption 2 with $P = \bar{K}, \rho = S^{-1/2}$, and $p_{sample} = S\bar{K}/N$. Notice that i.i.d. client sampling is the special case where $\bar{K} = 1$.

Cyclic participation can model a situation where each client group is available at a different time of day. For example, if client devices are mobile phones, then clients are available for participation at night, when phones are charging, likely to have internet connection, and otherwise idle. If devices are spread across the globe, then client groups are naturally formed by time zones. Cyclic participation is less stringent than regularized participation since not all clients are required to participate within each window. FedAvg was analyzed under cyclic participation for PL objectives [8], although this analysis is not applicable in our setting, which uses general non-convex objectives.

---

**Algorithm 1** Amplified SCAFFOLD

---

1: Initialize $\bar{\boldsymbol{x}}_0, \boldsymbol{G}_0^i, \boldsymbol{G}_0 \leftarrow \frac{1}{N}\sum_{i=1}^{N} \boldsymbol{G}_0^i, r_0 \leftarrow 0, \boldsymbol{u} \leftarrow 0$
2: **for** $r = 0, 1, \ldots, R-1$ **do**
3:    **for** $i \in [N]$ **do**
4:       $\boldsymbol{x}_{r,0}^i \leftarrow \bar{\boldsymbol{x}}_r$                                           `client`
5:       **for** $k = 0, \ldots, I-1$ **do**
6:          Sample $\xi_{r,k}^i \sim \mathcal{D}_i$
7:          $\boldsymbol{g}_{r,k}^i \leftarrow \nabla F_i(\boldsymbol{x}_{r,k}^i; \xi_{r,k}^i)$
8:          $\boldsymbol{x}_{r,k+1}^i \leftarrow \boldsymbol{x}_{r,k}^i - \eta(\boldsymbol{g}_{r,k}^i - \boldsymbol{G}_{r_0}^i + \boldsymbol{G}_{r_0})$
9:       **end for**
10:       $\Delta_r^i \leftarrow \boldsymbol{x}_{r,I}^i - \bar{\boldsymbol{x}}_r$
11:    **end for**
12:    $\bar{\boldsymbol{x}}_{r+1} \leftarrow \bar{\boldsymbol{x}}_r + \sum_{i=1}^{N} q_r^i \Delta_r^i$                            `server`
13:    $\boldsymbol{u} \leftarrow \boldsymbol{u} + \sum_{i=1}^{N} q_r^i \Delta_r^i$
14:    **if** $r = r_0 + P - 1$ **then**
15:       $\bar{\boldsymbol{x}}_{r_0+P} \leftarrow \bar{\boldsymbol{x}}_{r_0} + \gamma \boldsymbol{u}$
16:       **for** $i \in [N]$ **do**
17:          $\bar{q}_{r_0}^i \leftarrow \frac{1}{P} \sum_{s=r_0}^{r_0+P-1} q_s^i$
18:          $\boldsymbol{g}_{r_0}^i \leftarrow \frac{1}{IP\bar{q}_{r_0}^i} \sum_{s=r_0}^{r_0+P-1} \sum_{k=0}^{I-1} q_s^i \boldsymbol{g}_{r,k}^i$
19:          $\boldsymbol{G}_{r_0+P}^i \leftarrow \mathbb{1}\left\{\bar{q}_{r_0}^i > 0\right\} \boldsymbol{g}_{r_0}^i + \mathbb{1}\left\{\bar{q}_{r_0}^i = 0\right\} \boldsymbol{G}_{r_0}^i$
20:       **end for**
21:       $\boldsymbol{G}_{r_0+P} \leftarrow \frac{1}{N} \sum_{i=1}^{N} \boldsymbol{G}_{r_0+P}^i$
22:       $r_0 \leftarrow r_0 + P$
23:       $\boldsymbol{u} \leftarrow 0$
24:    **end if**
25: **end for**

---

# 4 Algorithm and Analysis

In this section, we present Amplified SCAFFOLD, our algorithm to solve the FL problem described in Section 3. Pseudocode for Amplified SCAFFOLD is shown in Algorithm 1. The main components of the Amplified SCAFFOLD algorithm are (1) amplified updates and (2) long-range control variates.

## 4.1 Algorithm Overview

To deal with the non-stationarity of client availability, Amplified SCAFFOLD performs amplified updates based on information accumulated over a window of $P$ rounds. In Algorithm 1, the variable $\boldsymbol{u}$ holds a weighted average of local updates to client models, weighted by client participation. Every $P$ rounds, the global model is updated in the direction $\boldsymbol{u}$ scaled by the amplification factor $\gamma$. Informally, the direction $\boldsymbol{u}$ includes information from all clients with equal representation, according to Assumption 2(b). Similar amplified updates are used in Amplified FedAvg [39].

Control variates for heterogeneous federated learning were first introduced by SCAFFOLD [17]. However, SCAFFOLD-style control variates are updated every time a client participates, which may not be appropriate under periodic availability. For example, under non-i.i.d. participation control variates for different clients would be updated with different frequencies, so that some clients may have consistently less accurate control variates than others. Informally, this may lead to a bias in which some clients' objective is underweighted relative to others. To deal with this issue, Amplified SCAFFOLD updates control variates based on information accumulated over a window of $P$ rounds, which enforces equal representation of all clients in expectation, according to Assumption 2(b).

**Comparison with [17, 39]** Although the two algorithmic components of Amplified SCAFFOLD individually appear in previous work [17, 39], we emphasize that our complexity results cannot be achieved by simply combining the analyses of these two works. The analysis of [39] requires $\epsilon^{-4}$ communication cost due to their treatment of the randomness in client participation. Here, we present

a tighter analysis with $\epsilon^{-2}$ cost from a more fine-grained treatment of the two sources of randomness (stochastic gradients and client sampling). See Section 4.4 for more details on our approach.

## 4.2 Main Results

Let $\hat{\boldsymbol{x}} = \bar{\boldsymbol{x}}_{mP}$, where $m$ is sampled uniformly from $\{0, \ldots, R/P-1\}$, and let $r_0 \in \{0, P, \ldots, R/P\}$. Denote $w_{r_0}^i = \frac{1}{N} \sum_{j=1}^N \frac{\mathbb{1}\{\bar{q}_{r_0}^j > 0\}}{P \bar{q}_{r_0}^j} \sum_{s=r_0}^{r_0+P-1} q_s^i q_s^j$ and $v_{r_0}^i = \bar{q}_{r_0}^i - \frac{1}{N}$. Informally, $w_{r_0}^i$ represents the "non-uniformity" of the client sampling distribution. We also consider a variable $\Lambda_{r_0}^i$ that depends only on the client sampling distribution and characterizes the sample size from which $\boldsymbol{G}_{r_0}^i$ is computed. See Appendix A.1 for further discussion of these quantities. We consider convergence under the following conditions, which are satisfied by several participation patterns of interest.

$$\mathbb{E}[w_{r_0}^i] \leq \frac{P^2}{N} \quad \text{for all } r_0 \text{ and } i, \tag{1}$$

$$\mathbb{E}\left[\sum_{i=1}^N \left(v_{r_0}^i\right)^2 \Lambda_{r_0}^i\right] \leq \rho^2 \quad \text{for all } r_0 \text{ and } i, \tag{2}$$

**Theorem 1.** *Suppose that Assumptions 1 and 2 hold, and that Equation 1 and Equation 2 hold. If $\gamma\eta \leq \frac{p_{sample}}{60LIP}$ and $\eta \leq \frac{\sqrt{p_{sample}}}{60LIP}$, then Algorithm 1 satisfies*

$$\mathbb{E}[\|\nabla f(\hat{\boldsymbol{x}})\|^2] \leq \mathcal{O}\left(\frac{\Delta}{\gamma\eta IR} + \left(\gamma\eta L\rho^2 + \eta^2 L^2 IP\right)\sigma^2\right).$$

**Corollary 1.** *For any $\epsilon > 0$ and $I \geq 1$, there exist choices of $\gamma$ and $\eta$ such that $\mathbb{E}[\|\nabla f(\hat{\boldsymbol{x}})\|^2] \leq \mathcal{O}(\epsilon^2)$ as long as $R \geq \mathcal{O}\left(\frac{\Delta L \rho^2 \sigma^2}{I\epsilon^4} + \frac{\Delta LP}{p_{sample}\epsilon^2}\right)$.*

The complexity of Amplified SCAFFOLD has several important properties:

**Reduced Communication** By choosing $I = \Theta(\Delta\rho^2\sigma^2 p_{sample}P^{-1}\epsilon^{-2})$, Amplified SCAFFOLD has communication complexity $R = \mathcal{O}(LPp_{sample}^{-1}\epsilon^{-2})$, which improves upon the $\epsilon^{-4}$ complexity of parallel SGD. We are not aware of any existing work that achieves this communication reduction for non-convex federated optimization with periodic participation.

**Unaffected by Heterogeneity** The iterations $RI$ and the number of communications $R$ are unaffected by heterogeneity, which is not achieved for periodic participation by any existing work [39, 8].

**Linear Speedup** The number of iterations $RI = \mathcal{O}(\Delta L\rho^2\sigma^2\epsilon^{-4})$ will exhibit linear speedup in the number of clients through the term $\rho^2$, depending on the client participation pattern.

## 4.3 Application to Participation Patterns

The results from Section 4.2 apply under any participation pattern that satisfies Assumption 2, Equation 1, and Equation 2, and below we discuss the participation patterns discussed in Section 3.2. The complexity of Amplified SCAFFOLD for each participation pattern are shown in Table 1, and these results can be obtained by plugging $\rho^2$, $P$, and $p_{sample}$ into Corollary 1, together with a choice of $I$ as described in Section 4.2. The derivations of each result below are given in Appendix B.

**Regularized Participation** Recall that regularized participation satisfies Assumption 2 with $p_{sample} = 1$, and $P$, $\rho^2$ are parameters of the participation pattern. Also, under regularized participation, $w_{r_0}^i = \bar{q}_{r_0}^i = 1/N$ almost surely, so that $\mathbb{E}[w_{r_0}^i] = 1/N \leq P^2/N$ and $v_{r_0}^i = 0$. Therefore Equation 1 and Equation 2 are satisfied. Plugging $p_{sample} = 1$ into Corollary 1 yields

$$R = \mathcal{O}\left(LP\epsilon^{-2}\right), \quad RI = \mathcal{O}\left(\Delta L\rho^2\sigma^2\epsilon^{-4}\right).$$

In this setting, our algorithm exhibits reduced communication and resilience to heterogeneity. To our knowledge, the only existing algorithm with theoretical guarantees for non-convex problems under regularized participation is Amplified FedAvg [39]. However, as seen in Table 1, the communication complexity of Amplified FedAvg has order $\epsilon^{-4}$ in terms of $\epsilon$ and suffers from a $\kappa^2$ dependence.

**Cyclic Participation** Recall that cyclic participation satisfies Assumption 2 with $P = \bar{K}$, $\rho = S^{-1/2}$, and $p_{\text{sample}} = S/N$. Also, $\mathbb{E}[w_{r_0}^i] = S/N^2 \leq P^2/N$ and $\mathbb{E}[(v_{r_0}^i)^2 \Lambda_{r_0}^i] = \rho^2$, so that Equation 1 and Equation 2 are satisfied. Based on the above parameter values, the resulting complexities are

$$R = \mathcal{O}\left(\frac{L\bar{K}}{\epsilon^2}\left(\frac{N}{S}\right)\right), \quad RI = \mathcal{O}\left(\frac{\Delta L \sigma^2}{S\epsilon^4}\right).$$

Again, Amplified SCAFFOLD achieves reduced communication, linear speedup, and resilience to heterogeneity. Amplified FedAvg [39] is the only existing algorithm with theoretical guarantees in this setting, but it fails to achieve resilience to heterogeneity or reduce communication cost outside of the trivial case of full participation ($\bar{K} = 1$, $S = N$). Also, even for the setting of PL-functions, the convergence rate of FedAvg under cyclic participation from [8] does not demonstrate an improvement with respect to the number of local steps. See Appendix F for further discussion of their results.

Recall that i.i.d. participation is a special case of cyclic participation with $\bar{K} = 1$. In this case, Amplified FedAvg fails to recover the reduced communication usually achieved under i.i.d. participation, such as by SCAFFOLD [17]. In fact, Amplified FedAvg fails to recover the communication cost of FedAvg under i.i.d. participation, requiring an additional factor of $LN$. The larger communication cost of Amplified FedAvg is a result of its convergence analysis, which does not leverage the property of unbiased participation (Assumption 2(b)) during the analysis, and requires $P = \mathcal{O}(\epsilon^{-2})$ in order to converge (see Appendix C for more details). In contrast, Amplified SCAFFOLD succeeds in recovering the results of SCAFFOLD under i.i.d. participation, with only a slightly worse dependence of $R$ on $N/S$. This difference in the order of $\frac{N}{S}$ is due to a potential small issue in the analysis of SCAFFOLD, which we intentionally avoided by accepting a slightly worse dependence on $\frac{N}{S}$. We provide a detailed discussion of the $N/S$ dependence in Appendix F.

## 4.4 Proof Sketch

The main challenges for demonstrating convergence are (1) simultaneously handling randomness from stochastic gradients and non-i.i.d. client sampling, and (2) controlling error of control variates under non-i.i.d. client sampling. Previous work [39] subverts (1) by conditioning on $\mathcal{Q}$ throughout the entire analysis. However, this eliminates the possibility of utilizing the condition $\mathbb{E}[\bar{q}_{r_0}^i] = 1/N$, and ultimately incurs a dependence on the data heterogeneity (see the term $\tilde{\delta}^2(P)$ in Theorem 3.1 of [39]). Instead, we take expectation over both sources of randomness throughout the analysis, which requires a careful treatment of each iterate's dependence, and enables communication reduction. For (2), previous analysis of federated algorithms with control variates [17] recursively bounds the error of control variates between consecutive rounds. However, this recursion crucially depends on i.i.d. client participation. We extend this analysis to our setting, establishing a recursion over the control variate error between consecutive windows of $P$ rounds. Establishing this recursion under non-i.i.d. participation involves a non-uniform average of non-uniform averages of error terms, which we handle by invoking the regularity conditions stated in Equation 1 and Equation 2.

Using smoothness of $f$, the objective function decrease $f(\bar{\boldsymbol{x}}_{r_0+P}) - f(\bar{\boldsymbol{x}}_{r_0})$ is upper bounded by $\langle \nabla f(\bar{\boldsymbol{x}}_{r_0}), \bar{\boldsymbol{x}}_{r_0+P} - \bar{\boldsymbol{x}}_{r_0} \rangle + \frac{L}{2}\|\bar{\boldsymbol{x}}_{r_0+P} - \bar{\boldsymbol{x}}_{r_0}\|^2$. Letting $\bar{\boldsymbol{x}}_{r,k} = \sum_{i=1}^N q_r^i \boldsymbol{x}_{r,k}^i$ be a weighted average of local models, the sum of the previous inner product and quadratic terms can be bounded by $-\gamma\eta IP\|\nabla f(\bar{\boldsymbol{x}}_{r_0})\|^2$, plus standard noise terms, the additional "drift" terms

$$\tilde{D}_{r,k} = \sum_{i=1}^N q_r^i \left\|\boldsymbol{x}_{r,k}^i - \bar{\boldsymbol{x}}_{r,k}\right\|^2 \quad \tilde{M}_{r,k} = \left\|\bar{\boldsymbol{x}}_{r,k} - \bar{\boldsymbol{x}}_{r_0}\right\|^2,$$

and control variate errors $C_{r_0}^i = \|\nabla f_i(\bar{\boldsymbol{x}}_{r_0}) - \boldsymbol{G}_{r_0}^i\|^2$. $\tilde{D}_{r,k}$ captures the distance between local client models, while $\tilde{M}_{r,k}$ captures the distance from local models to the previous global model $\bar{\boldsymbol{x}}_{r_0}$.

Taking conditional expectation $D_{r,k} = \mathbb{E}[\tilde{D}_{r,k}|\mathcal{Q}]$ and $M_{r,k} = \mathbb{E}[\tilde{M}_{r,k}|\mathcal{Q}]$, Lemma 1 bounds the drift terms by establishing and unrolling a mutually recurrent relation between $D_{r,k}$ and $M_{r,k}$. The resulting bound involves a non-uniform average over the control variate errors: $\sum_{i=1}^N q_r^i \mathbb{E}[C_{r_0}^i|\mathcal{Q}]$.

Denoting the average control variate error $C_{r_0} = \frac{1}{N} \sum_{i=1}^{N} C_{r_0}^i$, we want to bound $\mathbb{E}[C_{r_0+P}]$ in terms of $\mathbb{E}[C_{r_0}]$. $C_{r_0+P}$ can be decomposed into drift terms, but the result is a non-uniform average:

$$\sum_{s=r_0}^{r_0+P-1} \sum_{k=0}^{I-1} q_s^i (D_{s,k} + M_{s,k}).$$

Since the bound for each $M_{s,k} + D_{s,k}$ from Lemma 1 involves a non-uniform average over $C_{r_0}^i$, the resulting bound of $C_{r_0+P}$ involves a non-uniform average of non-uniform averages of $C_{r_0}^i$, instead of the uniform average $C_{r_0}$. The regularity conditions in Equation 1 and Equation 2 allow us to bound this nested non-uniform average by a uniform average, which finishes the recursion.

Putting everything together, we obtain the descent inequality

$$\mathbb{E}[\tilde{f}_{r_0+P}] \le \mathbb{E}[\tilde{f}_{r_0}] - \gamma\eta IP \mathbb{E}\left[\|\nabla f(\bar{\boldsymbol{x}}_{r_0})\|^2\right] + \gamma\eta IP(\gamma\eta L\rho^2 + \eta^2 L^2 IP)\sigma^2,$$

where $\tilde{f}_{r_0} := f(\bar{\boldsymbol{x}}_{r_0+P}) + \Phi(r_0 + P)$ and $\Phi$ is a potential function that depends on the control variate errors. Theorem 1 is then obtained by averaging over $r_0$ and isolating the gradient.

## 5 Experiments

We experimentally validate our algorithm for non-i.i.d client participation under three settings: minimizing a synthetic function, logistic regression for Fashion-MNIST [1] [45], and training a CNN for CIFAR-10 [20]. We also include an ablation study on Fashion-MNIST, to investigate how each algorithm is affected by changes in data heterogeneity, the number of participating clients, and the number of client groups in cyclic participation.

### 5.1 Setup

All of our experiments utilize a non-i.i.d. client participation pattern similar to cyclic participation (discussed in Section 3.2). We partition the total set of $N$ clients into $\bar{K}$ equally sized subsets, and at each training round only a single client group is available for participation. In our experiments, the available group does not change every round; instead, each group is available for $g$ rounds at a time. Under this pattern, Assumption 2 is satisfied with $P = g\bar{K}$. We refer to $g$ as the availability time.

We evaluate five algorithms: FedAvg [27], FedProx [22], SCAFFOLD [17], Amplified FedAvg [39], and Amplified SCAFFOLD (ours). We tune each algorithm's parameters by grid search, including learning rate $\eta$, amplification rate $\gamma$, and FedProx's $\mu$. The search ranges and tuned values can be found in Appendix D. All experiments were run on a single node with eight NVIDIA A6000 GPUs. Code is available at the following repository: `https://github.com/MingruiLiu-ML-Lab/FL-under-Periodic-Participation`

**Synthetic** We evaluate each algorithm's convergence on a difficult objective based on a lower bound for FedAvg [43]. The objective maps $\mathbb{R}^4$ to $\mathbb{R}$, is convex, and is parameterized by a smoothness $L$, stochastic gradient variance $\sigma^2$, and heterogeneity $\kappa$, so that it satisfies Assumption 1 by construction. The complete definition of the objective can be found in Appendix D. Since there are only two distinct local objectives, we set the number of clients $N = 2$ and the number of sampled clients $S = 1$, and the number of groups $\bar{K} = 2$. All other settings can be found in Appendix D.

**Fashion-MNIST and CIFAR-10** We evaluate each algorithm for training an image classifier, using logistic regression for Fashion-MNIST and a two-layer CNN for CIFAR-10. To simulate heterogeneous data in federated learning, we use a common protocol [17, 39], to partition each dataset into client datasets according to a data similarity parameter $s$. This protocol is detailed in Appendix D. Following [39], we set the number of clients $N = 250$, data similarity $s = 5\%$, and the number of sampled clients per round $S = 10$. For client participation, we set the number of groups $\bar{K} = 5$, so that each group contains clients that have majority label from two different classes. We run all baselines with 5 different random seeds and report the mean results with error bars in Section 5.2 (the radius of each error bar is 1 standard deviation). All other settings can be found in Appendix D.

Additional experimental results are provided in Appendix E, where we compare against extra baselines (FedAdam [32], FedYogi [32], FedAvg-M [5], and Amplified FedAvg with FedProx regularization), and evaluate training under another non-i.i.d. client participation pattern.

---

[1] Fashion-MNIST is licensed under the MIT License.

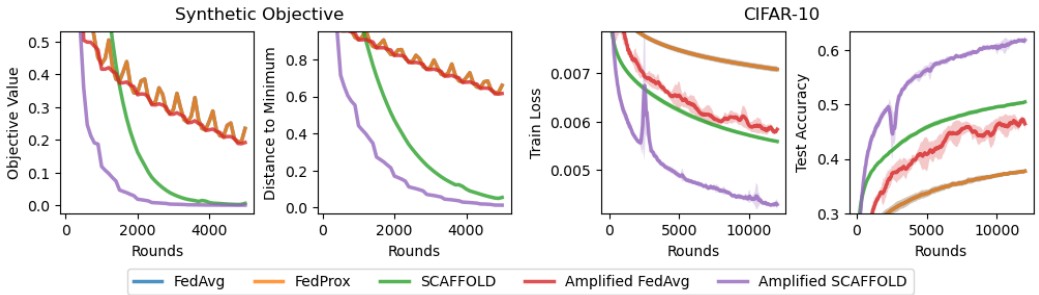

Figure 1: Results for synthetic objective and CIFAR-10. Left: Amplified SCAFFOLD and SCAF-FOLD both converge to the global minimum, but Amplified SCAFFOLD converges significantly faster. Right: Amplified SCAFFOLD converges to the best solution by a significant margin. Note that in both cases, the curves for FedAvg and FedProx are nearly overlapping.

## 5.2 Main Results

Results for the synthetic experiment and CIFAR-10 are shown in Figure 1, and results for Fashion-MNIST are shown in Figure 2. We make the following observations:

**Amplified SCAFFOLD converges the fastest.** In all three settings, Amplified SCAFFOLD reaches the best overall solution among all algorithms (by all metrics) and requires the fewest communication rounds. In the synthetic experiment, Amplified SCAFFOLD requires 800 communication rounds to reach an objective value of 0.2, while SCAFFOLD requires 1900 rounds, and both FedAvg and Amplified FedAvg require 4800 rounds to reach the same objective value.

**Amplified FedAvg is comparable to FedAvg.** Amplified FedAvg shows slight improvement over FedAvg for the synthetic experiment and for Fashion-MNIST. Only for CIFAR-10 is Amplified FedAvg significantly faster than FedAvg, but there it also exhibits a reduction in stability. The underwhelming experimental performance of Amplified FedAvg corroborates our discussion from Section 4.3; Amplified FedAvg requires many communication rounds and suffers from data heterogeneity.

Contrary to our findings, the original evaluation of Amplified FedAvg [39] showed a significant improvement over FedAvg. One explanation is that the original evaluation employed pretraining using FedAvg, so that each algorithm was evaluated only for fine-tuning. Our experiments suggest that Amplified FedAvg may have limited improvement over FedAvg when training from scratch.

**SCAFFOLD beats Amplified FedAvg.** Despite a lack of theoretical guarantees under non-i.i.d. participation, SCAFFOLD outperforms Amplified FedAvg in all settings. This suggests that SCAFFOLD may have reasonable performance under some non-i.i.d. participation patterns. For the synthetic objective and CIFAR-10, SCAFFOLD is still significantly slower than Amplified SCAFFOLD.

## 5.3 Ablation Study

To understand how each algorithm's performance is affected by data heterogeneity, the number of participating clients, and the number of client groups, we perform an ablation study on Fashion-MNIST. First, we fix the data similarity $s = 5\%$ and number of groups $\bar{K} = 5$ while varying the number of participating clients ($S$) over $\{5, 15, 20, 25\}$. Next, we fix $S = 10, \bar{K} = 5$ while varying the similarity $s$ over $\{2.5\%, 10\%, 33\%, 100\%\}$. Lastly, we fix $s = 5\%, S = 10$ while varying the number of client groups $\bar{K} \in \{2, 4, 6, 8\}$. In each of these 12 scenarios, we evaluate all five algorithms using the same settings as detailed in Section 3. We train with three random seeds for each algorithm, and report the average results in Figure 2 (right).

**Amplified SCAFFOLD reaches the best solution in all settings.** Similarly to Section 5.2, Amplified SCAFFOLD consistently reaches the best solution in terms of both training loss and testing accuracy. While our theoretical results provide guarantees for optimization, these experiments show that Amplified SCAFFOLD also exhibits superior generalization in a variety of settings.

**Robustness to data heterogeneity.** When changing from completely homogeneous data ($s = 100\%$) to extremely heterogeneous data ($s = 2.5\%$), the test accuracy of Amplified SCAFFOLD exhibits a

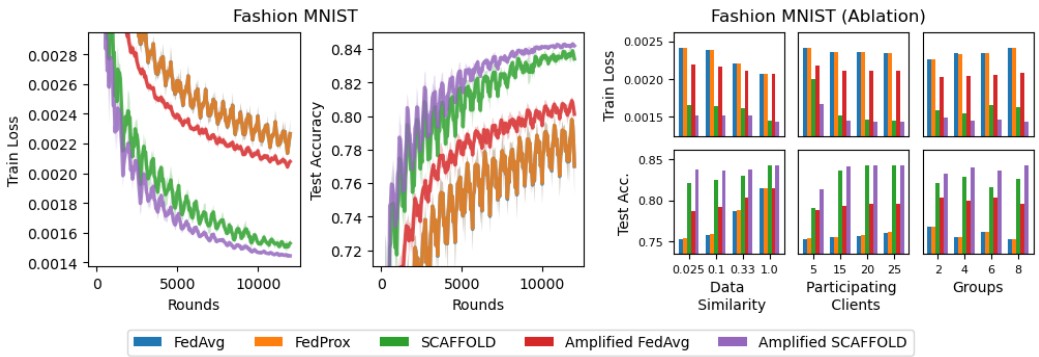

Figure 2: Results for Fashion MNIST and ablation study. Left: Amplified SCAFFOLD reaches the best solution, but SCAFFOLD is competitive. Other baselines are much slower. Right: Amplified SCAFFOLD is robust to changes in data heterogeneity, number of participating clients, and number of client groups.

very small decrease from $84.6\%$ to $84.45\%$, so that our algorithm behaves nearly identically with homogeneous data as with extremely heterogeneous data. All baselines suffer a larger decrease when transitioning from homogeneous data to heterogeneous data.

**Robustness to number of participating clients.** The number of participating clients has a smaller effect on performance than data heterogeneity, but some degradation happens in the extreme case $S = 5$. In particular, SCAFFOLD has competitive performance with large $S \geq 15$, but its test accuracy drops off significantly compared to Amplified SCAFFOLD in the case $S = 5$.

**Robustness to number of client groups.** As $\bar{K}$ increases, FedAvg and Amplified FedAvg get worse, while SCAFFOLD and Amplified SCAFFOLD maintain performance. It makes intuitive sense for an algorithm to degrade as $\bar{K}$ increases, since a larger $\bar{K}$ means that the participation is in some sense "further" from i.i.d. participation. Still, Amplified SCAFFOLD (and SCAFFOLD) are able to maintain performance even as $\bar{K}$ increases. While the worst-case communication complexity of Amplified SCAFFOLD (listed in Table 1) actually increases with $\bar{K}$, these experiments demonstrate that in practice, Amplified SCAFFOLD can maintain performance as $\bar{K}$ increases.

## 6  Conclusion

We propose Amplified SCAFFOLD, an optimization algorithm for federated learning under periodic client participation, and prove that it exhibits reduced communication cost, linear speedup, and is unaffected by data heterogeneity. We also show that Amplified SCAFFOLD experimentally outperforms baselines on standard benchmarks under non-i.i.d. client participation, and that the performance of our algorithm is robust to changes in data heterogeneity and the number of participating clients.

**Limitations** While our analysis covers a general class of participation patterns, it may not cover some participation patterns that appear in practice. Our framework requires that all clients have an equal chance of participation across well-defined windows of time that are known to the algorithm implementer, which may not always hold. One such practical situation is where clients may freely join or leave the federated learning process during training. Extending our algorithm and guarantees for this situation would require a reformulation of the optimization problem, and possibly additional assumptions about the participation structure. We leave such analysis for future work.

## Acknowledgements

We would like to thank the anonymous reviewers for their helpful comments. This work is supported by the Institute for Digital Innovation fellowship from George Mason University, a ORIEI seed funding, an IDIA P3 fellowship from George Mason University, a Cisco Faculty Research Award, and NSF award #2436217, #2425687. Experiments were partially run on Hopper, a research computing cluster provided by the Office of Research Computing at George Mason University (URL: https://orc.gmu.edu).

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

# Contents

# A  Proof of Theorem 1

## A.1  Preliminary Definitions

Let $\mathcal{G}_r$ denote the filtration generated by $\{\xi_{r,k}^i : 0 \le k \le I-1, i \in [N]\}$, that is, by the randomness in the stochastic gradients during round $r$. Also, let $\mathcal{Q}_{r_0}$ denote the filtration generated by $\{q_r^i : r_0 \le r < r_0 + P, i \in [N]\}$, that is, the randomness in client sampling between rounds $r_0$ and $r_0 + P - 1$ (inclusive). Also, let $\mathcal{G}$ denote the filtration generated by $\mathcal{G}_0 \cup \mathcal{G}_1 \cup \ldots \cup \mathcal{G}_{R-1}$ and $\mathcal{Q}$ denote the filtration generated by $\mathcal{Q}_0 \cup \mathcal{Q}_P \cup \ldots \cup \mathcal{Q}_{R-P}$. Similarly, let $\mathcal{G}_{:r}$ denote the filtration generated by $\mathcal{G}_0 \cup \mathcal{G}_1 \cup \ldots \cup \mathcal{G}_{r-1}$ and $\mathcal{Q}_{:r_0}$ denote the filtration generated by $\mathcal{Q}_0 \cup \mathcal{Q}_P \cup \ldots \cup \mathcal{Q}_{r_0-P}$. Lastly, denote $\mathbb{E}_{r_0}[\cdot] = \mathbb{E}[\cdot | \mathcal{Q}_{:r_0}, \mathcal{G}_{:r_0}]$.

In order to analyze the control variate errors (i.e., $\|\nabla f_i(\bar{\boldsymbol{x}}_{r_0}) - \boldsymbol{G}_{r_0}^i\|$), we introduce notation to refer to the iterates whose stochastic gradients were used to construct $\boldsymbol{G}_{r_0}^i$. These iterates are exactly the iterates during the most recent window before $r_0$ in which client $i$ was sampled, i.e., where $\frac{1}{P} \sum_{s=mP}^{(m+1)P-1} q_s^i > 0$. For each $r \in \{r_0, r_0+1, \ldots, r_0+P-1\}$, $k \in \{0, \ldots, I-1\}$, and $i \in [N]$, let

$$t_i(r) = \begin{cases} r & \text{if } \bar{q}_{r_0}^i > 0 \\ t_i(r-P) & \text{if } \bar{q}_{r_0}^i = 0 \end{cases}$$

with the initialization $t_i(r) = 0$ for all $r \in \{-P, \ldots, -1\}$. Then denote

$$\boldsymbol{y}_{r,k}^i = \boldsymbol{x}_{t_i(r),k}^i$$
$$z_r^i = q_{t_i(r)}^i$$
$$\zeta_{r,k}^i = \xi_{t_i(r),k}^i.$$

Also, denote $\bar{z}_{r_0}^i = \frac{1}{P} \sum_{r=r_0}^{r_0+P-1} z_r^i$. Then we can rewrite

$$\boldsymbol{G}_{r_0}^i = \frac{1}{P \bar{z}_{r_0-P}^i I} \sum_{s=r_0-P}^{r_0-1} z_s^i \sum_{k=0}^{I-1} \nabla F_i(\boldsymbol{y}_{s,k}^i; \zeta_{s,k}^i).$$

Define

$$\bar{\boldsymbol{x}}_{r,k} = \sum_{i=1}^{N} q_r^i \boldsymbol{x}_{r,k}^i$$

$$D_{r,k} = \sum_{i=1}^{N} q_r^i \mathbb{E}\left[\left\|\boldsymbol{x}_{r,k}^i - \bar{\boldsymbol{x}}_{r,k}\right\|^2 \Big| \mathcal{Q}\right]$$

$$M_{r,k} = \mathbb{E}\left[\left\|\bar{\boldsymbol{x}}_{r,k} - \bar{\boldsymbol{x}}_{r_0}\right\|^2 \Big| \mathcal{Q}\right]$$

$$S_{r_0}^i = \frac{1}{IP} \sum_{s=r_0-P}^{r_0-1} \sum_{k=0}^{I-1} \frac{z_s^i}{\bar{z}_{r_0-P}^i} \mathbb{E}\left[\left\|\boldsymbol{y}_{s,k}^i - \bar{\boldsymbol{x}}_{r_0}\right\|^2 \Big| \mathcal{Q}\right]$$

$$w_{r_0}^i = \frac{1}{N} \sum_{j=1}^{N} \frac{\mathbb{1}\left\{\bar{q}_{r_0}^j > 0\right\}}{P \bar{q}_{r_0}^j} \sum_{s=r_0}^{r_0+P-1} q_s^i q_s^j$$

$$v_{r_0}^i = \bar{q}_{r_0}^i - \frac{1}{N}$$

$$\Lambda_{r_0}^i = \frac{\frac{1}{P} \sum_{s=r_0-P}^{r_0-1} \left(z_s^i\right)^2}{\left(\frac{1}{P} \sum_{s=r_0-P}^{r_0-1} z_s^i\right)^2}.$$

As discussed in the main body, $\bar{\boldsymbol{x}}_{r,k}$ is a weighted average over local client models, weighted according to client participation. $D_{r,k}$ and $M_{r,k}$ are the drift terms described in the proof sketch of Section 4.4. In the proof sketch, we also informally discuss the control variate error $C_{r_0}^i = \|\nabla f_i(\bar{\boldsymbol{x}}_{r_0}) - \boldsymbol{G}_{r_0}^i\|^2$.

This error is closely related to the term $S_{r_0}^i$ defined above, since $\mathbb{E}\left[\left\|\nabla f_i(\bar{\boldsymbol{x}}_{r_0}) - \mathbb{E}[\boldsymbol{G}_{r_0}^i|\mathcal{Q}]\right\|^2\middle|\mathcal{Q}\right] = L^2 S_{r_0}^i$.

The quantities $w_{r_0}^i, v_{r_0}^i$ and $\Lambda_{r_0}^i$ arise in our proof from the use of control variates under non-i.i.d. client participation. As discussed in the proof sketch (Section 4.4), bounding the errors introduced by control variates involves a non-uniform average of non-uniform averages of error terms. This nested non-uniform average can be bounded by a uniform average as long as the term $w_{r_0}^i$ is not too large, which is exactly the requirement stated in Equation 1. On the other hand, the term $\Lambda_{r_0}^i$ arises while bounding the variance of the update $\|\bar{\boldsymbol{x}}_{r_0+P} - \bar{\boldsymbol{x}}_{r_0}\|$, and it represents the sample size for which the correction $\boldsymbol{G}_{r_0}^i$ is computed. When the client sampling distribution is closer to uniform, then each client may participate frequently during each window of $P$ rounds, and the effective sample size used to compute each control variate is large. In this case, the variance of the update $\|\bar{\boldsymbol{x}}_{r_0+P} - \bar{\boldsymbol{x}}_{r_0}\|$ will be smaller, since larger sample size implies smaller variance. For general client sampling distributions, $\Lambda_{r_0}^i$ describes the reduction of variance of one component of the update (i.e. $\boldsymbol{G}_{r_0}^i$) due to the sampling of stochastic gradients during the previous window of $P$ rounds. In our analysis, the variance of the update $\|\bar{\boldsymbol{x}}_{r_0+P} - \bar{\boldsymbol{x}}_{r_0}\|$ can be bounded in terms of the variance of $\boldsymbol{G}_{r_0}^i$, which in turn depends on $\Lambda_{r_0}^i$. The condition in Equation 2 essentially enforces that clients participate sufficiently uniformly, so that the variance of $\boldsymbol{G}_{r_0}^i$ can be bounded. We note that both Equation 1 and Equation 2 are satisfied by i.i.d. participation, regularized participation, and cyclic participation.

Finally, for ease of exposition, one clarification must be made about the conditional expectation $\mathbb{E}[\cdot|\mathcal{Q}]$ as it appears in, for example, $M_{r,k}$. The filtration $\mathcal{Q}$ contains randomness from every round from 0 to $R-1$, but the expression of which we are taking expectation may only depend on randomness for a smaller subset rounds. For example, in $M_{r,k}$, the expression $\|\bar{\boldsymbol{x}}_{r,k} - \bar{\boldsymbol{x}}_{r_0}\|^2$ only depends on the randomness up to round $r \leq r_0 + P$, so that

$$\mathbb{E}\left[\|\bar{\boldsymbol{x}}_{r,k} - \bar{\boldsymbol{x}}_{r_0}\|^2\middle|\mathcal{Q}\right] = \mathbb{E}\left[\|\bar{\boldsymbol{x}}_{r,k} - \bar{\boldsymbol{x}}_{r_0}\|^2\middle|\mathcal{Q}_{:r_0+P}\right].$$

In several places throughout the proof, we will replace $\mathbb{E}[\cdot|\mathcal{Q}]$ by $\mathbb{E}[\cdot|\mathcal{Q}_{:r_0+P}]$ in similar situations.

### A.2 Proofs

**Lemma 1.** *If* $\eta \leq \frac{1}{60LIP}$ *and* $\mathbb{E}_{r_0}[\bar{q}_{r_0}^i] = \frac{1}{N}$, *then*

$$D_{r,k} + M_{r,k} \leq 108\eta^2 I^2 P^2 \mathbb{E}\left[\|\nabla f(\bar{\boldsymbol{x}}_{r_0})\|^2\middle|\mathcal{Q}\right] + \left(75\eta^2 I + 65\eta^2 IP\rho^2\right)\sigma^2$$

$$+ 109\eta^2 L^2 I^2 P^2 \sum_{i=1}^{N}\left(\bar{q}_{r_0}^i + \frac{1}{N}\right)S_{r_0}^i + 36\eta^2 L^2 I^2 \sum_{i=1}^{N} q_r^i S_{r_0}^i, \tag{3}$$

*and*

$$\mathbb{E}\left[\sum_{r=r_0}^{r_0+P-1}\sum_{k=0}^{I-1}(D_{r,k} + M_{r,k})\right] \leq 108\eta^2 I^3 P^3 \mathbb{E}\left[\|\nabla f(\bar{\boldsymbol{x}}_{r_0})\|^2\right] + \left(75\eta^2 I^2 P + 65\eta^2 I^2 P^2 \rho^2\right)\sigma^2$$

$$+ 254\eta^2 L^2 I^3 P^3 \frac{1}{N}\sum_{i=1}^{N}\mathbb{E}\left[S_{r_0}^i\right]. \tag{4}$$

*Proof.* Denote

$$\bar{\boldsymbol{G}}_{r_0}^i = \frac{1}{P\bar{z}_{r_0-P}^i I}\sum_{s=r_0-P}^{r_0-1} z_s^i \sum_{k=0}^{I-1}\nabla f_i(\boldsymbol{y}_{s,k}^i)$$

$$\bar{\boldsymbol{G}}_{r_0} = \frac{1}{N}\sum_{i=1}^{N}\bar{\boldsymbol{G}}_{r_0}^i.$$

Also, denote $\boldsymbol{g}_{r,k}^i = \nabla F_i(\boldsymbol{x}_{r,k}^i; \xi_{r,k}^i) - \boldsymbol{G}_{r_0}^i + \boldsymbol{G}_{r_0}$ and $\bar{\boldsymbol{g}}_{r,k}^i = \nabla f_i(\boldsymbol{x}_{r,k}^i) - \bar{\boldsymbol{G}}_{r_0}^i + \bar{\boldsymbol{G}}_{r_0}$. $\bar{\boldsymbol{G}}_{r_0}^i$ is the analogue of $\bar{\boldsymbol{G}}_{r_0}$ that depends on deterministic gradients $\nabla f_i(\boldsymbol{y}_{s,k}^i)$ instead of stochastic gradients.

**Variance of updates** We first compute the errors $\mathbb{E}\left[\left\|\boldsymbol{g}_{r,k}^i - \bar{\boldsymbol{g}}_{r,k}^i\right\|^2 \middle| \mathcal{Q}\right]$ and $\mathbb{E}\left[\left\|\sum_{i=1}^N q_r^i \left(\boldsymbol{g}_{r,k}^i - \bar{\boldsymbol{g}}_{r,k}^i\right)\right\|^2 \middle| \mathcal{Q}\right]$, which will be needed in several places.

$$
\begin{aligned}
\mathbb{E}\left[\left\|\boldsymbol{G}_{r_0}^i - \bar{\boldsymbol{G}}_{r_0}^i\right\|^2 \middle| \mathcal{Q}\right] &= \mathbb{E}\left[\left\|\frac{1}{P\bar{z}_{r_0-P}^i I} \sum_{s=r_0-P}^{r_0-1} \sum_{k=0}^{I-1} z_s^i \left(\nabla F_i(\boldsymbol{y}_{s,k}^i; \zeta_{s,k}^i) - \nabla f_i(\boldsymbol{y}_{s,k}^i)\right)\right\|^2 \middle| \mathcal{Q}\right] \\
&\overset{(i)}{\leq} \frac{1}{P\bar{z}_{r_0-P}^i I} \sum_{s=r_0-P}^{r_0-1} \sum_{k=0}^{I-1} z_s^i \mathbb{E}\left[\left\|\nabla F_i(\boldsymbol{y}_{s,k}^i; \zeta_{s,k}^i) - \nabla f_i(\boldsymbol{y}_{s,k}^i)\right\|^2 \middle| \mathcal{Q}\right] \\
&\leq \frac{1}{P\bar{z}_{r_0-P}^i I} \sum_{s=r_0-P}^{r_0-1} \sum_{k=0}^{I-1} z_s^i \sigma^2 \\
&= \sigma^2,
\end{aligned}
\tag{5}
$$

where $(i)$ uses Jensen's inequality. Therefore

$$
\begin{aligned}
&\mathbb{E}\left[\left\|\boldsymbol{g}_{r,k}^i - \bar{\boldsymbol{g}}_{r,k}^i\right\|^2 \middle| \mathcal{Q}\right] \\
&= \mathbb{E}\left[\left\|\left(\nabla F_i(\boldsymbol{x}_{r,k}^i; \xi_{r,k}^i) - \nabla f_i(\boldsymbol{x}_{r,k}^i)\right) - \left(\boldsymbol{G}_{r_0}^i - \bar{\boldsymbol{G}}_{r_0}^i\right) + \left(\boldsymbol{G}_{r_0} - \bar{\boldsymbol{G}}_{r_0}\right)\right\|^2 \middle| \mathcal{Q}\right] \\
&\leq 3\mathbb{E}\left[\left\|\nabla F_i(\boldsymbol{x}_{r,k}^i; \xi_{r,k}^i) - \nabla f_i(\boldsymbol{x}_{r,k}^i)\right\|^2 \middle| \mathcal{Q}\right] + 3\mathbb{E}\left[\left\|\boldsymbol{G}_{r_0}^i - \bar{\boldsymbol{G}}_{r_0}^i\right\|^2 \middle| \mathcal{Q}\right] \\
&\quad + 3\mathbb{E}\left[\left\|\boldsymbol{G}_{r_0} - \bar{\boldsymbol{G}}_{r_0}\right\|^2 \middle| \mathcal{Q}\right] \\
&= 3\mathbb{E}\left[\left\|\nabla F_i(\boldsymbol{x}_{r,k}^i; \xi_{r,k}^i) - \nabla f_i(\boldsymbol{x}_{r,k}^i)\right\|^2 \middle| \mathcal{Q}\right] + 3\mathbb{E}\left[\left\|\boldsymbol{G}_{r_0}^i - \bar{\boldsymbol{G}}_{r_0}^i\right\|^2 \middle| \mathcal{Q}\right] \\
&\quad + 3\mathbb{E}\left[\left\|\frac{1}{N}\sum_{j=1}^N \left(\boldsymbol{G}_{r_0}^j - \bar{\boldsymbol{G}}_{r_0}^j\right)\right\|^2 \middle| \mathcal{Q}\right] \\
&= 3\mathbb{E}\left[\left\|\nabla F_i(\boldsymbol{x}_{r,k}^i; \xi_{r,k}^i) - \nabla f_i(\boldsymbol{x}_{r,k}^i)\right\|^2 \middle| \mathcal{Q}\right] + 3\mathbb{E}\left[\left\|\boldsymbol{G}_{r_0}^i - \bar{\boldsymbol{G}}_{r_0}^i\right\|^2 \middle| \mathcal{Q}\right] \\
&\quad + 3\frac{1}{N}\sum_{j=1}^N \mathbb{E}\left[\left\|\boldsymbol{G}_{r_0}^j - \bar{\boldsymbol{G}}_{r_0}^j\right\|^2 \middle| \mathcal{Q}\right] \\
&\leq 9\sigma^2,
\end{aligned}
\tag{6}
$$

where the last line uses Equation 5. Also

$$\mathbb{E}\left[\left\|\sum_{i=1}^{N} q_r^i \left(\boldsymbol{g}_{r,k}^i - \bar{\boldsymbol{g}}_{r,k}^i\right)\right\|^2 \Big| \mathcal{Q}\right]$$

$$= \mathbb{E}\left[\left\|\sum_{i=1}^{N} q_r^i \left(\left(\nabla F_i(\boldsymbol{x}_{r,k}^i; \xi_{r,k}^i) - \nabla f_i(\boldsymbol{x}_{r,k}^i)\right) - \left(\boldsymbol{G}_{r_0}^i - \bar{\boldsymbol{G}}_{r_0}^i\right) + \left(\boldsymbol{G}_{r_0} - \bar{\boldsymbol{G}}_{r_0}\right)\right)\right\|^2 \Big| \mathcal{Q}\right]$$

$$\leq 3\mathbb{E}\left[\left\|\sum_{i=1}^{N} q_r^i \left(\nabla F_i(\boldsymbol{x}_{r,k}^i; \xi_{r,k}^i) - \nabla f_i(\boldsymbol{x}_{r,k}^i)\right)\right\|^2 \Big| \mathcal{Q}\right] + 3\mathbb{E}\left[\left\|\sum_{i=1}^{N} q_r^i \left(\boldsymbol{G}_{r_0}^i - \bar{\boldsymbol{G}}_{r_0}^i\right)\right\|^2 \Big| \mathcal{Q}\right]$$

$$\quad + 3\mathbb{E}\left[\left\|\sum_{i=1}^{N} q_r^i \left(\boldsymbol{G}_{r_0} - \bar{\boldsymbol{G}}_{r_0}\right)\right\|^2 \Big| \mathcal{Q}\right]$$

$$= 3\mathbb{E}\left[\left\|\sum_{i=1}^{N} q_r^i \left(\nabla F_i(\boldsymbol{x}_{r,k}^i; \xi_{r,k}^i) - \nabla f_i(\boldsymbol{x}_{r,k}^i)\right)\right\|^2 \Big| \mathcal{Q}\right] + 3\mathbb{E}\left[\left\|\sum_{i=1}^{N} q_r^i \left(\boldsymbol{G}_{r_0}^i - \bar{\boldsymbol{G}}_{r_0}^i\right)\right\|^2 \Big| \mathcal{Q}\right]$$

$$\quad + 3\mathbb{E}\left[\left\|\sum_{i=1}^{N} \frac{1}{N} \left(\boldsymbol{G}_{r_0}^i - \bar{\boldsymbol{G}}_{r_0}^i\right)\right\|^2 \Big| \mathcal{Q}\right]$$

$$\overset{(i)}{=} 3\sum_{i=1}^{N} \mathbb{E}\left[\left\|q_r^i \left(\nabla F_i(\boldsymbol{x}_{r,k}^i; \xi_{r,k}^i) - \nabla f_i(\boldsymbol{x}_{r,k}^i)\right)\right\|^2 \Big| \mathcal{Q}\right] + 3\sum_{i=1}^{N} \mathbb{E}\left[\left\|q_r^i \left(\boldsymbol{G}_{r_0}^i - \bar{\boldsymbol{G}}_{r_0}^i\right)\right\|^2 \Big| \mathcal{Q}\right]$$

$$\quad + 3\sum_{i=1}^{N} \mathbb{E}\left[\left\|\frac{1}{N} \left(\boldsymbol{G}_{r_0}^i - \bar{\boldsymbol{G}}_{r_0}^i\right)\right\|^2 \Big| \mathcal{Q}\right]$$

$$= 3\sum_{i=1}^{N} \left(q_r^i\right)^2 \mathbb{E}\left[\left\|\nabla F_i(\boldsymbol{x}_{r,k}^i; \xi_{r,k}^i) - \nabla f_i(\boldsymbol{x}_{r,k}^i)\right\|^2 \Big| \mathcal{Q}\right] + 3\sum_{i=1}^{N} \left(q_r^i\right)^2 \mathbb{E}\left[\left\|\boldsymbol{G}_{r_0}^i - \bar{\boldsymbol{G}}_{r_0}^i\right\|^2 \Big| \mathcal{Q}\right]$$

$$\quad + 3\frac{1}{N^2} \sum_{i=1}^{N} \mathbb{E}\left[\left\|\boldsymbol{G}_{r_0}^i - \bar{\boldsymbol{G}}_{r_0}^i\right\|^2 \Big| \mathcal{Q}\right]$$

$$\leq 6\sigma^2 \sum_{i=1}^{N} \left(q_r^i\right)^2 + 3\frac{\sigma^2}{N}$$

$$\leq \sigma^2 \left(6\rho^2 + 3\frac{1}{N}\right)$$

$$\overset{(ii)}{\leq} 9\rho^2\sigma^2, \tag{7}$$

where $(i)$ uses the fact that stochastic gradient noise is independent across each client, and $(ii)$ uses $\rho^2 \geq \frac{1}{N}$.

**One-step recursive bound for $D_{r,k}$**    For any $k \geq 0$,

$$D_{r,k+1} = \sum_{i=1}^{N} q_r^i \mathbb{E}\left[\left\|\boldsymbol{x}_{r,k+1}^i - \sum_{j=1}^{N} q_r^j \boldsymbol{x}_{r,k+1}^j\right\|^2 \bigg|_{\mathcal{Q}}\right]$$

$$= \sum_{i=1}^{N} q_r^i \mathbb{E}\left[\left\|\boldsymbol{x}_{r,k}^i - \eta \boldsymbol{g}_{r,k}^i - \sum_{j=1}^{N} q_r^j (\boldsymbol{x}_{r,k}^j - \eta \boldsymbol{g}_{r,k}^j)\right\|^2 \bigg|_{\mathcal{Q}}\right]$$

$$= \sum_{i=1}^{N} q_r^i \mathbb{E}\left[\left\|\boldsymbol{x}_{r,k}^i - \sum_{j=1}^{N} q_r^j \boldsymbol{x}_{r,k}^j - \eta \left(\boldsymbol{g}_{r,k}^i - \sum_{j=1}^{N} q_r^j \boldsymbol{g}_{r,k}^j\right)\right\|^2 \bigg|_{\mathcal{Q}}\right]$$

$$= \sum_{i=1}^{N} q_r^i \mathbb{E}\left[\left\|\boldsymbol{x}_{r,k}^i - \sum_{j=1}^{N} q_r^j \boldsymbol{x}_{r,k}^j - \eta \left(\bar{\boldsymbol{g}}_{r,k}^i - \sum_{j=1}^{N} q_r^j \bar{\boldsymbol{g}}_{r,k}^j\right)\right\|^2 \bigg|_{\mathcal{Q}}\right]$$

$$+ \eta^2 \sum_{i=1}^{N} q_r^i \mathbb{E}\left[\left\|(\boldsymbol{g}_{r,k}^i - \bar{\boldsymbol{g}}_{r,k}^i) + \sum_{j=1}^{N} q_r^j (\boldsymbol{g}_{r,k}^j - \bar{\boldsymbol{g}}_{r,k}^j)\right\|^2 \bigg|_{\mathcal{Q}}\right]$$

$$\overset{(i)}{=} \sum_{i=1}^{N} q_r^i \mathbb{E}\left[\left\|\boldsymbol{x}_{r,k}^i - \sum_{j=1}^{N} q_r^j \boldsymbol{x}_{r,k}^j - \eta \left(\bar{\boldsymbol{g}}_{r,k}^i - \sum_{j=1}^{N} q_r^j \bar{\boldsymbol{g}}_{r,k}^j\right)\right\|^2 \bigg|_{\mathcal{Q}}\right] + 36\eta^2 \sigma^2$$

$$\overset{(ii)}{\leq} \left(1 + \frac{1}{\lambda_1}\right) \sum_{i=1}^{N} q_r^i \mathbb{E}\left[\left\|\boldsymbol{x}_{r,k}^i - \sum_{j=1}^{N} q_r^j \boldsymbol{x}_{r,k}^j\right\|^2 \bigg|_{\mathcal{Q}}\right]$$

$$+ \eta^2 (1 + \lambda_1) \sum_{i=1}^{N} q_r^i \mathbb{E}\left[\left\|\bar{\boldsymbol{g}}_{r,k}^i - \sum_{j=1}^{N} q_r^j \bar{\boldsymbol{g}}_{r,k}^j\right\|^2 \bigg|_{\mathcal{Q}}\right] + 36\eta^2 \sigma^2$$

$$= \left(1 + \frac{1}{\lambda_1}\right) D_{r,k} + \eta^2 (1 + \lambda_1) \sum_{i=1}^{N} q_r^i \mathbb{E}\left[\left\|\bar{\boldsymbol{g}}_{r,k}^i - \sum_{j=1}^{N} q_r^j \bar{\boldsymbol{g}}_{r,k}^j\right\|^2 \bigg|_{\mathcal{Q}}\right] + 36\eta^2 \sigma^2, \quad (8)$$

where $(i)$ uses

$$\mathbb{E}\left[\left\|(\boldsymbol{g}_{r,k}^i - \bar{\boldsymbol{g}}_{r,k}^i) + \sum_{j=1}^{N} q_r^j (\boldsymbol{g}_{r,k}^j - \bar{\boldsymbol{g}}_{r,k}^j)\right\|^2 \bigg|_{\mathcal{Q}}\right]$$

$$\leq 2\mathbb{E}\left[\left\|\boldsymbol{g}_{r,k}^i - \bar{\boldsymbol{g}}_{r,k}^i\right\|^2 \bigg|_{\mathcal{Q}}\right] + 2\mathbb{E}\left[\left\|\sum_{j=1}^{N} q_r^j (\boldsymbol{g}_{r,k}^j - \bar{\boldsymbol{g}}_{r,k}^j)\right\|^2 \bigg|_{\mathcal{Q}}\right]$$

$$\leq 2\mathbb{E}\left[\left\|\boldsymbol{g}_{r,k}^i - \bar{\boldsymbol{g}}_{r,k}^i\right\|^2 \bigg|_{\mathcal{Q}}\right] + 2\sum_{j=1}^{N} q_r^j \mathbb{E}\left[\left\|\boldsymbol{g}_{r,k}^j - \bar{\boldsymbol{g}}_{r,k}^j\right\|^2 \bigg|_{\mathcal{Q}}\right]$$

$$\overset{(a)}{\leq} 36\sigma^2,$$

$(a)$ uses Equation 6, and $(ii)$ uses Young's inequality. Focusing on the second term of Equation 8:

$$\sum_{i=1}^{N} q_r^i \mathbb{E}\left[\left\|\bar{\boldsymbol{g}}_{r,k}^i - \sum_{j=1}^{N} q_r^j \bar{\boldsymbol{g}}_{r,k}^j\right\|^2 \middle| \mathcal{Q}\right] = \sum_{i=1}^{N} q_r^i \mathbb{E}\left[\left\|\sum_{j=1}^{N} q_r^j (\bar{\boldsymbol{g}}_{r,k}^i - \bar{\boldsymbol{g}}_{r,k}^j)\right\|^2 \middle| \mathcal{Q}\right]$$

$$\overset{(i)}{\leq} \sum_{i=1}^{N} q_r^i \sum_{j=1}^{N} q_r^j \mathbb{E}\left[\left\|\bar{\boldsymbol{g}}_{r,k}^i - \bar{\boldsymbol{g}}_{r,k}^j\right\|^2 \middle| \mathcal{Q}\right]$$

$$\leq \sum_{i=1}^{N} q_r^i \sum_{j=1}^{N} q_r^j \mathbb{E}\left[\left\|\nabla f_i(\boldsymbol{x}_{r,k}^i) - \bar{\boldsymbol{G}}_{r_0}^i - \nabla f_j(\boldsymbol{x}_{r,k}^j) + \bar{\boldsymbol{G}}_{r_0}^j\right\|^2 \middle| \mathcal{Q}\right]$$

$$\leq \sum_{i=1}^{N} q_r^i \sum_{j=1}^{N} q_r^j \left(2\mathbb{E}\left[\left\|\nabla f_i(\boldsymbol{x}_{r,k}^i) - \bar{\boldsymbol{G}}_{r_0}^i\right\|^2 \middle| \mathcal{Q}\right] + 2\mathbb{E}\left[\left\|\nabla f_j(\boldsymbol{x}_{r,k}^j) - \bar{\boldsymbol{G}}_{r_0}^j\right\|^2 \middle| \mathcal{Q}\right]\right)$$

$$= 4 \sum_{i=1}^{N} q_r^i \mathbb{E}\left[\left\|\nabla f_i(\boldsymbol{x}_{r,k}^i) - \bar{\boldsymbol{G}}_{r_0}^i\right\|^2 \middle| \mathcal{Q}\right],$$

where $(i)$ uses Jensen's inequality. Using the decomposition

$$\nabla f_i(\boldsymbol{x}_{r,k}^i) - \bar{\boldsymbol{G}}_{r_0}^i = (\nabla f_i(\boldsymbol{x}_{r,k}^i) - \nabla f_i(\bar{\boldsymbol{x}}_{r,k})) + (\nabla f_i(\bar{\boldsymbol{x}}_{r,k}) - \nabla f_i(\bar{\boldsymbol{x}}_{r_0})) + (\nabla f_i(\bar{\boldsymbol{x}}_{r_0}) - \bar{\boldsymbol{G}}_{r_0}^i),$$

we have

$$\sum_{i=1}^{N} q_r^i \mathbb{E}\left[\left\|\bar{\boldsymbol{g}}_{r,k}^i - \sum_{j=1}^{N} q_r^j \bar{\boldsymbol{g}}_{r,k}^j\right\|^2 \middle| \mathcal{Q}\right]$$

$$\leq 4 \sum_{i=1}^{N} q_r^i \left(3\mathbb{E}\left[\left\|\nabla f_i(\boldsymbol{x}_{r,k}^i) - \nabla f_i(\bar{\boldsymbol{x}}_{r,k})\right\|^2 \middle| \mathcal{Q}\right] + 3\mathbb{E}\left[\left\|\nabla f_i(\bar{\boldsymbol{x}}_{r,k}) - \nabla f_i(\bar{\boldsymbol{x}}_{r_0})\right\|^2 \middle| \mathcal{Q}\right]\right.$$

$$\left. + 3\mathbb{E}\left[\left\|\nabla f_i(\bar{\boldsymbol{x}}_{r_0}) - \bar{\boldsymbol{G}}_{r_0}^i\right\|^2 \middle| \mathcal{Q}\right]\right)$$

$$\leq 12L^2 \sum_{i=1}^{N} q_r^i \mathbb{E}\left[\left\|\boldsymbol{x}_{r,k}^i - \bar{\boldsymbol{x}}_{r,k}\right\|^2 \middle| \mathcal{Q}\right] + 12L^2 \sum_{i=1}^{N} q_r^i \mathbb{E}\left[\left\|\bar{\boldsymbol{x}}_{r,k} - \bar{\boldsymbol{x}}_{r_0}\right\|^2 \middle| \mathcal{Q}\right]$$

$$+ 12 \sum_{i=1}^{N} q_r^i \mathbb{E}\left[\left\|\nabla f_i(\bar{\boldsymbol{x}}_{r_0}) - \bar{\boldsymbol{G}}_{r_0}^i\right\|^2 \middle| \mathcal{Q}\right]$$

$$= 12L^2 D_{r,k} + 12L^2 M_{r,k} + 12 \sum_{i=1}^{N} q_r^i \mathbb{E}\left[\left\|\nabla f_i(\bar{\boldsymbol{x}}_{r_0}) - \frac{1}{P\bar{z}_{r_0-P}^i I} \sum_{s=r_0-P}^{r_0-1} \sum_{k=0}^{I-1} z_s^i \nabla f_i(\boldsymbol{y}_{s,k}^i)\right\|^2 \middle| \mathcal{Q}\right]$$

$$= 12L^2 D_{r,k} + 12L^2 M_{r,k} + 12 \sum_{i=1}^{N} q_r^i \mathbb{E}\left[\left\|\frac{1}{P\bar{z}_{r_0-P}^i I} \sum_{s=r_0-P}^{r_0-1} \sum_{k=0}^{I-1} z_s^i (\nabla f_i(\bar{\boldsymbol{x}}_{r_0}) - \nabla f_i(\boldsymbol{y}_{s,k}^i))\right\|^2 \middle| \mathcal{Q}\right]$$

$$\leq 12L^2 D_{r,k} + 12L^2 M_{r,k} + 12 \sum_{i=1}^{N} q_r^i \left(\frac{1}{P\bar{z}_{r_0-P}^i I} \sum_{s=r_0-P}^{r_0-1} \sum_{k=0}^{I-1} z_s^i \mathbb{E}\left[\left\|\nabla f_i(\bar{\boldsymbol{x}}_{r_0}) - \nabla f_i(\boldsymbol{y}_{s,k}^i)\right\|^2 \middle| \mathcal{Q}\right]\right)$$

$$\leq 12L^2 D_{r,k} + 12L^2 M_{r,k} + 12L^2 \sum_{i=1}^{N} q_r^i \left(\frac{1}{P\bar{z}_{r_0-P}^i I} \sum_{s=r_0-P}^{r_0-1} \sum_{k=0}^{I-1} z_s^i \mathbb{E}\left[\left\|\bar{\boldsymbol{x}}_{r_0} - \boldsymbol{y}_{s,k}^i\right\|^2 \middle| \mathcal{Q}\right]\right)$$

$$\leq 12L^2 D_{r,k} + 12L^2 M_{r,k} + 12L^2 \sum_{i=1}^{N} q_r^i S_{r_0}^i.$$

Plugging back into Equation 8,

$$D_{r,k+1} \leq \left(1 + \frac{1}{\lambda_1}\right) D_{r,k} + 12\eta^2 L^2(1 + \lambda_1)\left(D_{r,k} + M_{r,k} + \sum_{i=1}^{N} q_r^i S_{r_0}^i\right) + 36\eta^2\sigma^2$$

$$\leq \left(1 + \frac{1}{\lambda_1} + 12\eta^2 L^2(1 + \lambda_1)\right) D_{r,k} + 12\eta^2 L^2(1 + \lambda_1)\left(M_{r,k} + \sum_{i=1}^{N} q_r^i S_{r_0}^i\right) + 36\eta^2\sigma^2.$$

Now choose $\lambda_1 = 2I$, so that $1 \leq \frac{\lambda_1}{2}$ and

$$12\eta^2 L^2(1 + \lambda_1) \leq 18\eta^2 L^2\lambda_1 \leq 36\eta^2 L^2 I \leq \frac{1}{2I},$$

where we used the condition $\eta \leq \frac{1}{60LIP}$. This yields the following recursive bound on $D_{r,k+1}$:

$$D_{r,k+1} \leq \left(1 + \frac{1}{I}\right) D_{r,k} + 18\eta^2 L^2 I M_{r,k} + 18\eta^2 L^2 I \sum_{i=1}^{N} q_r^i S_{r_0}^i + 36\eta^2\sigma^2. \tag{9}$$

**One-step recursive bound for $M_{r,k}$**   For any $k \geq 0$,

$$M_{r,k+1} = \mathbb{E}\left[\left\|\sum_{i=1}^{N} q_r^i \boldsymbol{x}_{r,k+1}^i - \bar{\boldsymbol{x}}_{r_0}\right\|^2 \Big| \mathcal{Q}\right]$$

$$= \mathbb{E}\left[\left\|\sum_{i=1}^{N} q_r^i \boldsymbol{x}_{r,k}^i - \bar{\boldsymbol{x}}_{r_0} - \eta\sum_{i=1}^{N} q_r^i \boldsymbol{g}_{r,k}^i\right\|^2 \Big| \mathcal{Q}\right]$$

$$\leq \mathbb{E}\left[\left\|\sum_{i=1}^{N} q_r^i \boldsymbol{x}_{r,k}^i - \bar{\boldsymbol{x}}_{r_0} - \eta\sum_{i=1}^{N} q_r^i \bar{\boldsymbol{g}}_{r,k}^i\right\|^2 \Big| \mathcal{Q}\right] + \eta^2\mathbb{E}\left[\left\|\sum_{i=1}^{N} q_r^i(\boldsymbol{g}_{r,k}^i - \bar{\boldsymbol{g}}_{r,k}^i)\right\|^2 \Big| \mathcal{Q}\right]$$

$$\overset{(i)}{\leq} \mathbb{E}\left[\left\|\sum_{i=1}^{N} q_r^i \boldsymbol{x}_{r,k}^i - \bar{\boldsymbol{x}}_{r_0} - \eta\sum_{i=1}^{N} q_r^i \bar{\boldsymbol{g}}_{r,k}^i\right\|^2 \Big| \mathcal{Q}\right] + 9\eta^2\rho^2\sigma^2$$

$$\overset{(ii)}{\leq} \left(1 + \frac{1}{\lambda_2}\right)\mathbb{E}\left[\left\|\sum_{i=1}^{N} q_r^i \boldsymbol{x}_{r,k}^i - \bar{\boldsymbol{x}}_{r_0}\right\|^2 \Big| \mathcal{Q}\right] + \eta^2(1 + \lambda_2)\mathbb{E}\left[\left\|\sum_{i=1}^{N} q_r^i \bar{\boldsymbol{g}}_{r,k}^i\right\|^2 \Big| \mathcal{Q}\right] + 9\eta^2\rho^2\sigma^2$$

$$\leq \left(1 + \frac{1}{\lambda_2}\right) M_{r,k} + \eta^2(1 + \lambda_2)\mathbb{E}\left[\left\|\sum_{i=1}^{N} q_r^i \bar{\boldsymbol{g}}_{r,k}^i\right\|^2 \Big| \mathcal{Q}\right] + 9\eta^2\rho^2\sigma^2$$

$$\leq \left(1 + \frac{1}{\lambda_2}\right) M_{r,k} + \eta^2(1 + \lambda_2)\sum_{i=1}^{N} q_r^i\mathbb{E}\left[\left\|\bar{\boldsymbol{g}}_{r,k}^i\right\|^2 \Big| \mathcal{Q}\right] + 9\eta^2\rho^2\sigma^2, \tag{10}$$

where $(i)$ uses Equation 7 and $(ii)$ uses Young's inequality. Focusing on the second term in Equation 10:

$$\sum_{i=1}^{N} q_r^i \mathbb{E}\left[\left\|\bar{\boldsymbol{g}}_{r,k}^i\right\|^2 \middle| \mathcal{Q}\right]$$

$$= \sum_{i=1}^{N} q_r^i \mathbb{E}\left[\left\|\nabla f_i(\boldsymbol{x}_{r,k}^i) - \bar{\boldsymbol{G}}_{r_0}^i + \bar{\boldsymbol{G}}_{r_0}\right\|^2 \middle| \mathcal{Q}\right]$$

$$= \sum_{i=1}^{N} q_r^i \mathbb{E}\Bigg[\Big\|(\nabla f_i(\boldsymbol{x}_{r,k}^i) - \nabla f_i(\bar{\boldsymbol{x}}_{r,k})) + (\nabla f_i(\bar{\boldsymbol{x}}_{r,k}) - \nabla f_i(\bar{\boldsymbol{x}}_{r_0}))$$
$$+ (\nabla f_i(\bar{\boldsymbol{x}}_{r_0}) - \bar{\boldsymbol{G}}_{r_0}^i) + (\bar{\boldsymbol{G}}_{r_0} - \nabla f(\bar{\boldsymbol{x}}_{r_0})) + \nabla f(\bar{\boldsymbol{x}}_{r_0})\Big\|^2 \middle| \mathcal{Q}\Bigg]$$

$$\leq 5\sum_{i=1}^{N} q_r^i \mathbb{E}\left[\left\|\nabla f_i(\boldsymbol{x}_{r,k}^i) - \nabla f_i(\bar{\boldsymbol{x}}_{r,k})\right\|^2 \middle| \mathcal{Q}\right] + 5\sum_{i=1}^{N} q_r^i \mathbb{E}\left[\left\|\nabla f_i(\bar{\boldsymbol{x}}_{r,k}) - \nabla f_i(\bar{\boldsymbol{x}}_{r_0})\right\|^2 \middle| \mathcal{Q}\right]$$
$$+ 5\sum_{i=1}^{N} q_r^i \mathbb{E}\left[\left\|\nabla f_i(\bar{\boldsymbol{x}}_{r_0}) - \bar{\boldsymbol{G}}_{r_0}^i\right\|^2 \middle| \mathcal{Q}\right] + 5\mathbb{E}\left[\left\|\bar{\boldsymbol{G}}_{r_0} - \nabla f(\bar{\boldsymbol{x}}_{r_0})\right\|^2 \middle| \mathcal{Q}\right] + 5\mathbb{E}\left[\left\|\nabla f(\bar{\boldsymbol{x}}_{r_0})\right\|^2 \middle| \mathcal{Q}\right]$$

$$\leq 5L^2 \sum_{i=1}^{N} q_r^i \mathbb{E}\left[\left\|\boldsymbol{x}_{r,k}^i - \bar{\boldsymbol{x}}_{r,k}\right\|^2 \middle| \mathcal{Q}\right] + 5L^2 \sum_{i=1}^{N} q_r^i \mathbb{E}\left[\left\|\bar{\boldsymbol{x}}_{r,k} - \bar{\boldsymbol{x}}_{r_0}\right\|^2 \middle| \mathcal{Q}\right]$$
$$+ 5\sum_{i=1}^{N} \left(q_r^i + \frac{1}{N}\right) \mathbb{E}\left[\left\|\nabla f_i(\bar{\boldsymbol{x}}_{r_0}) - \frac{1}{P\bar{z}_{r_0-P}^i I}\sum_{s=r_0-P}^{r_0-1}\sum_{k=0}^{I-1} z_s^i \nabla f_i(\boldsymbol{y}_{s,k}^i)\right\|^2 \middle| \mathcal{Q}\right] + 5\mathbb{E}\left[\left\|\nabla f(\bar{\boldsymbol{x}}_{r_0})\right\|^2 \middle| \mathcal{Q}\right]$$

$$\leq 5L^2 D_{r,k} + 5L^2 M_{r,k}$$
$$+ 5\sum_{i=1}^{N} \left(q_r^i + \frac{1}{N}\right) \mathbb{E}\left[\left\|\frac{1}{P\bar{z}_{r_0-P}^i I}\sum_{s=r_0-P}^{r_0-1}\sum_{k=0}^{I-1} z_s^i (\nabla f_i(\bar{\boldsymbol{x}}_{r_0}) - \nabla f_i(\boldsymbol{y}_{s,k}^i))\right\|^2 \middle| \mathcal{Q}\right] + 5\mathbb{E}\left[\left\|\nabla f(\bar{\boldsymbol{x}}_{r_0})\right\|^2 \middle| \mathcal{Q}\right]$$

$$\leq 5L^2 D_{r,k} + 5L^2 M_{r,k}$$
$$+ 5\sum_{i=1}^{N} \left(q_r^i + \frac{1}{N}\right) \frac{1}{P\bar{z}_{r_0-P}^i I}\sum_{s=r_0-P}^{r_0-1}\sum_{k=0}^{I-1} z_s^i \mathbb{E}\left[\left\|\nabla f_i(\bar{\boldsymbol{x}}_{r_0}) - \nabla f_i(\boldsymbol{y}_{s,k}^i)\right\|^2 \middle| \mathcal{Q}\right] + 5\mathbb{E}\left[\left\|\nabla f(\bar{\boldsymbol{x}}_{r_0})\right\|^2 \middle| \mathcal{Q}\right]$$

$$\leq 5L^2 D_{r,k} + 5L^2 M_{r,k} + 5L^2 \sum_{i=1}^{N} \left(q_r^i + \frac{1}{N}\right) S_{r_0}^i + 5\mathbb{E}\left[\left\|\nabla f(\bar{\boldsymbol{x}}_{r_0})\right\|^2 \middle| \mathcal{Q}\right].$$

Plugging back into Equation 10:

$$M_{r,k+1} \leq \left(1 + \frac{1}{\lambda_2}\right) M_{r,k} + 5\eta^2 L^2(1 + \lambda_2)\left(D_{r,k} + M_{r,k} + \sum_{i=1}^{N}\left(q_r^i + \frac{1}{N}\right) S_{r_0}^i\right)$$
$$+ 5\eta^2(1 + \lambda_2)\mathbb{E}\left[\left\|\nabla f(\bar{\boldsymbol{x}}_{r_0})\right\|^2 \middle| \mathcal{Q}\right] + 9\eta^2 \rho^2 \sigma^2$$

$$\leq \left(1 + \frac{1}{\lambda_2} + 5\eta^2 L^2(1 + \lambda_2)\right) M_{r,k} + 5\eta^2 L^2(1 + \lambda_2)\left(D_{r,k} + \sum_{i=1}^{N}\left(q_r^i + \frac{1}{N}\right) S_{r_0}^i\right)$$
$$+ 5\eta^2(1 + \lambda_2)\mathbb{E}\left[\left\|\nabla f(\bar{\boldsymbol{x}}_{r_0})\right\|^2 \middle| \mathcal{Q}\right] + 9\eta^2 \rho^2 \sigma^2.$$

Now choose $\lambda_2 = 2IP$, so that $1 \leq \frac{\lambda_2}{2}$ and

$$5\eta^2 L^2(1 + \lambda_2) \leq \frac{15}{2}\eta^2 L^2 \lambda_2 \leq 15\eta^2 L^2 IP \leq 15L^2 IP \frac{1}{72L^2 I^2 P^2} \leq \frac{1}{2IP},$$

where we used the condition $\eta \leq \frac{1}{60LIP}$. This yields the following recursive bound on $M_{r,k+1}$:

$$M_{r,k+1} \leq \left(1 + \frac{1}{IP}\right) M_{r,k} + 15\eta^2 L^2 IP D_{r,k} + 15\eta^2 L^2 IP \sum_{i=1}^{N} \left(q_r^i + \frac{1}{N}\right) S_{r_0}^i$$

$$+ 15\eta^2 IP \mathbb{E}\left[\|\nabla f(\bar{x}_{r_0})\|^2 \Big| \mathcal{Q}\right] + 9\eta^2 \rho^2 \sigma^2. \tag{11}$$

Note that the same argument can be used to bound $M_{r,0}$ in terms of $M_{r-1,I-1}$ with the same recurrence relation.

**Unrolling the recursion** For $0 \leq t \leq IP$, let $r_t = r_0 + \lfloor t/I \rfloor$ and $k_t = t - \lfloor t/I \rfloor * I$. Then let $d_t = D_{r_t,k_t}$ and $m_t = M_{r_t,k_t}$. Also, let

$$q_1 = 18\eta^2 L^2 I$$

$$q_2 = 15\eta^2 L^2 IP$$

$$a_r = 18\eta^2 L^2 I \sum_{i=1}^{N} q_r^i S_{r_0}^i + 36\eta^2 \sigma^2$$

$$b_r = 15\eta^2 L^2 IP \sum_{i=1}^{N} \left(q_r^i + \frac{1}{N}\right) S_{r_0}^i + 15\eta^2 IP \mathbb{E}\left[\|\nabla f(\bar{x}_{r_0})\|^2 \Big| \mathcal{Q}\right] + 9\eta^2 \rho^2 \sigma^2.$$

Then Equation 9 and Equation 11 imply the following mutually recursive relation over $d_t$ and $m_t$:

$$d_t = \begin{cases} 0 & k_t = 0 \\ \left(1 + \frac{1}{I}\right) d_{t-1} + q_1 m_{t-1} + a_{r_{t-1}} & \text{otherwise} \end{cases}$$

$$m_t = \left(1 + \frac{1}{IP}\right) m_{t-1} + q_2 d_{t-1} + b_{r_{t-1}}, \tag{12}$$

where $d_0 = m_0 = 0$. We will unroll this recurrence, showing a bound for $m_t$ and $d_t$ in terms of the following quantities. For any $0 \leq t \leq IP - 1$ and $0 \leq s \leq r_t$, let $j(s,t) = \min\{I, t - sI\}$ and $\ell(s,t) = \max\{0, t - (s+1)I\}$. For any $s, t$, if $k_{t+1} = 0$, define

$$\alpha_{s,t+1} = 0.$$

Otherwise, define

$$\alpha_{s,t+1} = \alpha_{s,t}\left(1 + \frac{1}{I}\right) + q_1 P \left(\left(1 + \frac{1}{IP}\right)^{j(s,t)} - 1\right)\left(1 + \frac{1}{IP}\right)^{\ell(s,t)}$$

$$+ q_1 q_2 \sum_{i=0}^{t-sI-2} \left(1 + \frac{1}{IP}\right)^{i} \alpha_{s,t-1-i}.$$

Also, define

$$\beta_{s,t+1} = \beta_{s,t}\left(1 + \frac{1}{IP}\right) + \mathbb{1}\left\{s = r_t\right\}\frac{q_2}{P}\left(\left(1 + \frac{1}{I}\right)^{k_t} - 1\right) + q_1 q_2 \sum_{i=0}^{k_t-2} \left(1 + \frac{1}{I}\right)^{i} \beta_{s,t-1-i}$$

$$\psi_{s,t+1} = \frac{1}{P} \sum_{i=0}^{t-sI-1} \left(1 + \frac{1}{IP}\right)^{i} \alpha_{s,t-i}$$

$$\phi_{s,t+1} = P \sum_{i=0}^{k_t-1} \left(1 + \frac{1}{I}\right)^{i} \beta_{s,t-i},$$

under the initial conditions

$$\phi_{s,kI} = 0 \text{ for all } s \leq P \text{ and } s \leq k \leq P$$

$$\alpha_{s,kI} = 0 \text{ for all } s \leq P \text{ and } s \leq k \leq P$$

$$\psi_{s,sI} = 0 \text{ for all } s \leq P$$

$$\beta_{s,sI} = 0 \text{ for all } s \leq P.$$

Now we can unroll the recurrence in Equation 12, by proving the following statements by induction on $t$:

$$d_t \leq \left(\left(1 + \frac{1}{I}\right)^{k_t} - 1\right) a_{r_t} I + q_1 I \sum_{s=0}^{r_t} \phi_{s,t} a_s + I \sum_{s=0}^{r_t} \alpha_{s,t} b_s \tag{13}$$

$$m_t \leq IP \sum_{s=0}^{r_t} \left(\left(\left(1 + \frac{1}{IP}\right)^{j(s,t)} - 1\right)\left(1 + \frac{1}{IP}\right)^{\ell(s,t)} + q_2 \psi_{s,t}\right) b_s + IP \sum_{s=0}^{r_t} \beta_{s,t} a_s. \tag{14}$$

Equation 13 and Equation 14 hold for the base case $t = 0$, since $d_0 = m_0 = 0$. Now suppose that Equation 13 and Equation 14 hold for some $t \leq IP$, and we will show that they hold for $t+1$. We consider two cases: $k_{t+1} \neq 0$ and $k_{t+1} = 0$.

In the first case, $r_{t+1} = r_t$, i.e., step $t+1$ is in the same round as step $t$, and $k_{t+1} = k_t + 1$. Using Equation 12 together with the inductive hypothesis:

$$d_{t+1} \leq \left(1 + \frac{1}{I}\right)\left(\left(\left(1 + \frac{1}{I}\right)^{k_t} - 1\right) a_{r_t} I + q_1 I \sum_{s=0}^{r_t} \phi_{s,t} a_s + I \sum_{s=0}^{r_t} \alpha_{s,t} b_s\right)$$

$$+ q_1\left(IP \sum_{s=0}^{r_t} \left(\left(\left(1 + \frac{1}{IP}\right)^{j(s,t)} - 1\right)\left(1 + \frac{1}{IP}\right)^{\ell(s,t)} + q_2 \psi_{s,t}\right) b_s + IP \sum_{s=0}^{r_t} \beta_{s,t} a_s\right) + a_{r_t}$$

$$\leq \left(\left(1 + \frac{1}{I}\right)^{k_t + 1} - \left(1 + \frac{1}{I}\right) + \frac{1}{I}\right) a_{r_t} I + q_1 I \sum_{s=0}^{r_t} \left(\left(1 + \frac{1}{I}\right)\phi_{s,t} + P\beta_{s,t}\right) a_s$$

$$+ I \sum_{s=0}^{r_t} \left(\left(1 + \frac{1}{I}\right)\alpha_{s,t} + q_1 P\left(\left(1 + \frac{1}{IP}\right)^{j(s,t)} - 1\right)\left(1 + \frac{1}{IP}\right)^{\ell(s,t)} + q_1 q_2 P\psi_{s,t}\right) b_s$$

$$\overset{(i)}{\leq} \left(\left(1 + \frac{1}{I}\right)^{k_t + 1} - 1\right) a_{r_t} I + q_1 I \sum_{s=0}^{r_t} \phi_{s,t+1} a_s + I \sum_{s=0}^{r_t} \alpha_{s,t+1} b_s$$

$$\overset{(ii)}{\leq} \left(\left(1 + \frac{1}{I}\right)^{k_{t+1}} - 1\right) a_{r_{t+1}} I + q_1 I \sum_{s=0}^{r_{t+1}} \phi_{s,t+1} a_s + I \sum_{s=0}^{r_{t+1}} \alpha_{s,t+1} b_s,$$

where $(i)$ uses the fact that $\phi_{s,t+1} = \left(1 + \frac{1}{I}\right)\phi_{s,t} + P\beta_{s,t}$ and the definitions of $\alpha_{s,t+1}$, $\psi_{s,t}$, and $(ii)$ uses $k_{t+1} = k_t + 1$ and $r_{t+1} = r_t$. Similarly,

$$m_{t+1} \le \left(1 + \frac{1}{IP}\right)\left(IP\sum_{s=0}^{r_t}\left(\left(\left(1 + \frac{1}{IP}\right)^{j(s,t)} - 1\right)\left(1 + \frac{1}{IP}\right)^{\ell(s,t)} + q_2\psi_{s,t}\right)b_s + IP\sum_{s=0}^{r_t}\beta_{s,t}a_s\right)$$

$$+ q_2\left(\left(\left(1 + \frac{1}{I}\right)^{k_t} - 1\right)a_{r_t}I + q_1 I\sum_{s=0}^{r_t}\phi_{s,t}a_s + I\sum_{s=0}^{r_t}\alpha_{s,t}b_s\right) + b_{r_t}$$

$$\le IP\sum_{s=0}^{r_t}\left(\left(\left(1 + \frac{1}{IP}\right)^{j(s,t)} - 1\right)\left(1 + \frac{1}{IP}\right)^{\ell(s,t)+1}\right.$$

$$+ \mathbb{1}\left\{s = r_t\right\}\frac{1}{IP} + q_2\left(\left(1 + \frac{1}{IP}\right)\psi_{s,t} + \frac{1}{P}\alpha_{s,t}\right)\bigg)b_s$$

$$+ IP\sum_{s=0}^{r_t}\left(\left(1 + \frac{1}{IP}\right)\beta_{s,t} + \mathbb{1}\left\{s = r_t\right\}\frac{q_2}{P}\left(\left(1 + \frac{1}{I}\right)^{k_t} - 1\right) + \frac{q_1 q_2}{P}\phi_{s,t}\right)a_s$$

$$\overset{(i)}{\le} IP\sum_{s=0}^{r_t}\left(\left(\left(1 + \frac{1}{IP}\right)^{j(s,t+1)} - 1\right)\left(1 + \frac{1}{IP}\right)^{\ell(s,t+1)} + q_2\left(\left(1 + \frac{1}{IP}\right)\psi_{s,t} + \frac{1}{P}\alpha_{s,t}\right)\right)b_s$$

$$+ IP\sum_{s=0}^{r_t}\left(\left(1 + \frac{1}{IP}\right)\beta_{s,t} + \mathbb{1}\left\{s = r_t\right\}\frac{q_2}{P}\left(\left(1 + \frac{1}{I}\right)^{k_t} - 1\right) + \frac{q_1 q_2}{P}\phi_{s,t}\right)a_s$$

$$\overset{(ii)}{\le} IP\sum_{s=0}^{r_t}\left(\left(\left(1 + \frac{1}{IP}\right)^{j(s,t+1)} - 1\right)\left(1 + \frac{1}{IP}\right)^{\ell(s,t+1)} + q_2\psi_{s,t+1}\right)b_s + IP\sum_{s=0}^{r_t}\beta_{s,t+1}a_s$$

$$\overset{(iii)}{\le} IP\sum_{s=0}^{r_{t+1}}\left(\left(\left(1 + \frac{1}{IP}\right)^{j(s,t+1)} - 1\right)\left(1 + \frac{1}{IP}\right)^{\ell(s,t+1)} + q_2\psi_{s,t+1}\right)b_s + IP\sum_{s=0}^{r_{t+1}}\beta_{s,t+1}a_s,$$

where $(i)$ uses the fact that for $s < r_t$:

$$\left(\left(1 + \frac{1}{IP}\right)^{j(s,t)} - 1\right)\left(1 + \frac{1}{IP}\right)^{\ell(s,t)+1} + \mathbb{1}\left\{s = r_t\right\}\frac{1}{IP}$$

$$= \left(\left(1 + \frac{1}{IP}\right)^{I} - 1\right)\left(1 + \frac{1}{IP}\right)^{t+1-(s+1)I}$$

$$= \left(\left(1 + \frac{1}{IP}\right)^{j(s,t+1)} - 1\right)\left(1 + \frac{1}{IP}\right)^{\ell(s,t+1)}$$

and for $s = r_t$:

$$\left(\left(1 + \frac{1}{IP}\right)^{j(s,t)} - 1\right)\left(1 + \frac{1}{IP}\right)^{\ell(s,t)+1} + \mathbb{1}\left\{s = r_t\right\}\frac{1}{IP}$$

$$= \left(\left(1 + \frac{1}{IP}\right)^{t-sI} - 1\right)\left(1 + \frac{1}{IP}\right) + \frac{1}{IP}$$

$$= \left(\left(1 + \frac{1}{IP}\right)^{t+1-sI} - 1\right)$$

$$= \left(\left(1 + \frac{1}{IP}\right)^{j(s,t+1)} - 1\right)\left(1 + \frac{1}{IP}\right)^{\ell(s,t+1)},$$

$(ii)$ uses the fact that $\psi_{s,t+1} = \left(1 + \frac{1}{IP}\right)\psi_{s,t} + \frac{1}{P}\alpha_{s,t}$ and the definitions of $\beta_{s,t+1}$, $\phi_{s,t}$, and $(iii)$ uses $k_{t+1} = k_t + 1$ and $r_{t+1} = r_t$. This completes the inductive step for the first case ($k_{t+1} \ne 0$).

In the second case (i.e., $k_{t+1} = 0$), Equation 13 must hold for $t + 1$, since $d_{t+1} = 0$. So it only remains to show Equation 14 holds for $t + 1$. Note that $r_t = r_{t+1} - 1$. As in the first case, we can

use Equation 12 together with the inductive hypothesis to obtain:

$$m_{t+1} \leq IP \sum_{s=0}^{r_t} \left( \left( \left( 1 + \frac{1}{IP} \right)^{j(s,t+1)} - 1 \right) \left( 1 + \frac{1}{IP} \right)^{\ell(s,t+1)} + q_2 \psi_{s,t+1} \right) b_s + IP \sum_{s=0}^{r_t} \phi_{s,t+1} a_s$$

$$\leq IP \sum_{s=0}^{r_{t+1}} \left( \left( \left( 1 + \frac{1}{IP} \right)^{j(s,t+1)} - 1 \right) \left( 1 + \frac{1}{IP} \right)^{\ell(s,t+1)} + q_2 \psi_{s,t+1} \right) b_s + IP \sum_{s=0}^{r_{t+1}} \phi_{s,t+1} a_s,$$

where the second line uses the fact that the $r_{t+1}$-st element of both sums is 0, since $j(r_{t+1}, t+1) = 0$, $\psi_{r_{t+1},t+1} = 0$, and $\phi_{r_{t+1},t+1} = 0$. This completes the inductive step for both cases, and proves Equation 13 and Equation 14.

We can now bound $\alpha_{s,t}$ and $\beta_{s,t}$ separately by induction on $t$. First, for any $s \leq P-1$ and $t$ with $sI \leq t \leq IP$, we claim that

$$\alpha_{s,t} \leq 36 q_1 IP, \tag{15}$$

which we will show by induction on $t$. Let $s \leq P-1$ be given. For the base case, Equation 15 holds when $t = sI$ since $\alpha_{s,sI} = 0$. Now suppose that it holds for all $t' \leq t$. Then

$$\left( \left( 1 + \frac{1}{IP} \right)^{m_1} - 1 \right) \left( 1 + \frac{1}{IP} \right)^{m_2}$$

$$\leq \left( \left( 1 + \frac{1}{IP} \right)^{t-sI} - 1 \right) \left( 1 + \frac{1}{IP} \right)^{0}$$

$$\leq \left( 1 + \frac{1}{IP} \right)^{t-sI} - 1, \tag{16}$$

and

$$q_1 q_2 \sum_{i=0}^{t-sI-2} \left( 1 + \frac{1}{IP} \right)^{i} \alpha_{s,t-1-i} \leq 36 q_1^2 q_2 IP \sum_{i=0}^{t-sI-2} \left( 1 + \frac{1}{IP} \right)^{i}$$

$$= 36 q_1^2 q_2 I^2 P^2 \left( \left( 1 + \frac{1}{IP} \right)^{t-sI-1} - 1 \right)$$

$$\leq q_1 P \left( \left( 1 + \frac{1}{IP} \right)^{t-sI} - 1 \right), \tag{17}$$

where the last line uses the definition of $q_1$ and $q_2$ together with the condition $\eta \leq \frac{1}{60 LIP}$:

$$36 q_1 q_2 I^2 P = 36 \cdot 270 \eta^4 L^4 I^4 P^2 \leq \frac{36 \cdot 270}{60^4} L^4 I^4 P^2 \frac{1}{L^4 I^4 P^4} \leq \frac{1}{P^2} \leq 1.$$

Plugging Equation 16 and Equation 17 into the definition of $\alpha_{s,t+1}$ yields

$$\alpha_{s,t+1} \leq \alpha_{s,t} \left( 1 + \frac{1}{I} \right) + 2 q_1 P \left( \left( 1 + \frac{1}{IP} \right)^{t-sI} - 1 \right)$$

$$\overset{(i)}{\leq} 2 q_1 P \sum_{i=0}^{k_t} \left( 1 + \frac{1}{I} \right)^{i} \left( \left( 1 + \frac{1}{IP} \right)^{t-sI-i} - 1 \right)$$

$$\leq 2 q_1 P \left( 1 + \frac{1}{I} \right)^{I} \sum_{i=0}^{k_t} \left( \left( 1 + \frac{1}{IP} \right)^{t-sI-i} - 1 \right)$$

$$\leq 6 q_1 P \sum_{i=0}^{k_t} \left( 1 + \frac{1}{IP} \right)^{t-sI-i} \leq 6 q_1 P \left( 1 + \frac{1}{IP} \right)^{(r_t-s)I} \sum_{i=0}^{k_t} \left( 1 + \frac{1}{IP} \right)^{i}$$

$$\leq 18 q_1 P \sum_{i=0}^{I-1} \left( 1 + \frac{1}{IP} \right)^{i} \overset{(ii)}{\leq} 18 q_1 \sum_{i=0}^{IP-1} \left( 1 + \frac{1}{IP} \right)^{i}$$

$$\leq 18 q_1 IP \left( \left( 1 + \frac{1}{IP} \right)^{IP} - 1 \right) \leq 36 q_1 IP,$$

where $(i)$ unrolls the recurrence on the first line until $\alpha_{s,r_t I} = 0$, $(ii)$ uses $P \sum_{i=0}^{I-1} \left(1 + \frac{1}{IP}\right)^i \leq \sum_{i=0}^{IP-1} \left(1 + \frac{1}{IP}\right)^i$ since $\left(1 + \frac{1}{IP}\right)^i$ is increasing with $i$, and we repeatedly used $\left(1 + \frac{1}{x}\right)^x < e \leq 3$. This completes the induction and proves Equation 15.

We will similarly prove the statement

$$\beta_{s,t} \leq 12 \frac{q_2 I}{P}. \tag{18}$$

for all $s \leq P - 1$ and $t$ with $sI \leq t \leq IP$ by induction on $t$. Let $s \leq P - 1$ be given. For the base case, Equation 15 holds when $t = sI$ since $\beta_{s,sI} = 0$. Now suppose that it holds for all $t' \leq t$. Then

$$q_1 q_2 \sum_{i=0}^{k_t - 2} \left(1 + \frac{1}{I}\right)^i \beta_{s,t-1-i} \leq 12 \frac{q_1 q_2^2 I}{P} \sum_{i=0}^{k_t - 2} \left(1 + \frac{1}{I}\right)^i$$

$$= 12 \frac{q_1 q_2^2 I^2}{P} \left(\left(1 + \frac{1}{I}\right)^{k_t - 1} - 1\right)$$

$$\leq 12 \frac{q_1 q_2^2 I^2}{P} \left(\left(1 + \frac{1}{I}\right)^{k_t} - 1\right) \tag{19}$$

$$\leq \frac{q_2}{P} \left(\left(1 + \frac{1}{I}\right)^{k_t} - 1\right), \tag{20}$$

where the last line uses the definition of $q_1$ and $q_2$ together with the condition $\eta \leq \frac{1}{60LIP}$:

$$12 q_1 q_2 I^2 \leq 12 \cdot 270 \eta^4 L^4 I^4 P \leq \frac{12 \cdot 270}{60^4} L^4 I^4 P \frac{1}{L^4 I^4 P^4} \leq \frac{1}{P^3} \leq 1.$$

We now consider two cases: $t < (s+1)I$ and $t \geq (s+1)I$. In the first case, we have $s = r_t$, and plugging Equation 20 into the definition of $\beta_{s,t+1}$ yields

$$\beta_{s,t+1} \leq \beta_{s,t} \left(1 + \frac{1}{IP}\right) + 2 \frac{q_2}{P} \left(\left(1 + \frac{1}{I}\right)^{k_t} - 1\right)$$

$$\overset{(i)}{\leq} 2 \frac{q_2}{P} \sum_{i=0}^{k_t} \left(1 + \frac{1}{IP}\right)^i \left(\left(1 + \frac{1}{I}\right)^{k_t - i} - 1\right)$$

$$\leq 2 \frac{q_2}{P} \left(1 + \frac{1}{IP}\right)^{k_t} \sum_{i=0}^{k_t} \left(\left(1 + \frac{1}{I}\right)^{k_t - i} - 1\right)$$

$$\leq 2 \frac{q_2}{P} \left(1 + \frac{1}{IP}\right)^I \sum_{i=0}^{k_t} \left(1 + \frac{1}{I}\right)^i$$

$$\leq 6 \frac{q_2 I}{P} \left(\left(1 + \frac{1}{I}\right)^{k_t + 1} - 1\right) \leq 6 \frac{q_2 I}{P} \left(\left(1 + \frac{1}{I}\right)^I - 1\right)$$

$$\leq 12 \frac{q_2 I}{P},$$

where $(i)$ unrolls the recurrence on the first line until $\beta_{s,sI} = 0$ and we repeatedly used $\left(1 + \frac{1}{x}\right)^x < e \leq 3$. In the second case, we have $s > r_t$, so plugging Equation 19 into the definition of $\beta_{s,t+1}$

yields

$$\beta_{s,t+1} \leq \beta_{s,t}\left(1 + \frac{1}{IP}\right) + 12\frac{q_1 q_2^2 I^2}{P}\left(\left(1 + \frac{1}{I}\right)^{k_t} - 1\right)$$

$$\leq \beta_{s,t}\left(1 + \frac{1}{IP}\right) + 12\frac{q_1 q_2^2 I^2}{P}\left(\left(1 + \frac{1}{I}\right)^{I} - 1\right)$$

$$\leq \beta_{s,t}\left(1 + \frac{1}{IP}\right) + 24\frac{q_1 q_2^2 I^2}{P}$$

$$\overset{(i)}{\leq} 24\frac{q_1 q_2^2 I^2}{P}\sum_{i=0}^{t-sI}\left(1 + \frac{1}{IP}\right)^{i}$$

$$= 24 q_1 q_2^2 I^3\left(\left(1 + \frac{1}{IP}\right)^{t-sI+1} - 1\right) \leq 24 q_1 q_2^2 I^3\left(\left(1 + \frac{1}{IP}\right)^{IP} - 1\right)$$

$$\leq 48 q_1 q_2^2 I^3 \leq 12\frac{q_2 I}{P},$$

where $(i)$ unrolls the recurrence on the previous line until $\beta_{s,sI} = 0$ and the last inequality uses the definition of $q_1$ and $q_2$ together with the condition $\eta \leq \frac{1}{60LIP}$:

$$48 q_1 q_2 I^2 = 48 \cdot 270 \eta^4 L^4 I^4 P \leq \frac{48 \cdot 270}{60^4}L^4 I^4 P\frac{1}{L^4 I^4 P^4} \leq \frac{12}{P^3} \leq \frac{12}{P}.$$

This completes the induction in both cases and proves Equation 18.

We can then use Equation 15 and Equation 18 to yield bounds for the remaining terms of Equation 13 and Equation 14 as follows:

$$q_2\psi_{s,t} = \frac{q_2}{P}\sum_{i=0}^{t-sI-1}\left(1 + \frac{1}{IP}\right)^{i}\alpha_{s,t-1-i} \leq 36 q_1 q_2 I\sum_{i=0}^{t-sI-1}\left(1 + \frac{1}{IP}\right)^{i}$$

$$\leq 36 q_1 q_2 I\sum_{i=0}^{IP-1}\left(1 + \frac{1}{IP}\right)^{i} \leq 36 q_1 q_2 I^2 P\left(\left(1 + \frac{1}{IP}\right)^{IP} - 1\right)$$

$$\leq 72 q_1 q_2 I^2 P \leq 72 \cdot 270 \eta^4 L^4 I^4 P^2 \leq \frac{72 \cdot 270}{60^4}L^4 I^4 P^2\frac{1}{L^4 I^4 P^4} \leq \frac{1}{P^2},$$

and

$$q_1\phi_{s,t} = q_1 P\sum_{i=0}^{k_t-1}\left(1 + \frac{1}{I}\right)^{i}\beta_{s,t-i} \leq 12 q_1 q_2 I\sum_{i=0}^{k_t-1}\left(1 + \frac{1}{I}\right)^{i}$$

$$= 12 q_1 q_2 I^2\left(\left(1 + \frac{1}{I}\right)^{k_t} - 1\right) \leq 12 q_1 q_2 I^2\left(\left(1 + \frac{1}{I}\right)^{I} - 1\right) \leq 24 q_1 q_2 I^2.$$

Finally, we can plug these into Equation 13 to yield

$$d_t \leq \left(\left(1 + \frac{1}{I}\right)^{k_t} - 1\right)a_{r_t}I + 24 q_1 q_2 I^3\sum_{s=0}^{r_t}a_s + 36 q_1 I^2 P\sum_{s=0}^{r_t}b_s$$

$$\leq 2a_{r_t}I + 24 q_1 q_2 I^3\sum_{s=0}^{r_t}a_s + 36 q_1 I^2 P\sum_{s=0}^{r_t}b_s,$$

and into Equation 14

$$m_t \le IP \sum_{s=0}^{r_t} \left( \left( \left( 1 + \frac{1}{IP} \right)^{m_1} - 1 \right) \left( 1 + \frac{1}{IP} \right)^{m_2} + \frac{1}{P^2} \right) b_s + 12q_2 I^2 \sum_{s=0}^{r_t} a_s$$

$$\overset{(i)}{\le} IP \sum_{s=0}^{r_t} \left( \frac{6}{P} + \frac{1}{P^2} \right) b_s + 12q_2 I^2 \sum_{s=0}^{r_t} a_s$$

$$\le 7I \sum_{s=0}^{r_t} b_s + 12q_2 I^2 \sum_{s=0}^{r_t} a_s,$$

where $(i)$ uses

$$\left( 1 + \frac{1}{IP} \right)^{m_1} - 1 \le \left( 1 + \frac{1}{IP} \right)^{I} - 1 = IP \sum_{i=0}^{I-1} \left( 1 + \frac{1}{IP} \right)^i$$

$$\le I \sum_{i=0}^{IP-1} \left( 1 + \frac{1}{IP} \right)^i = \frac{1}{P} \left( \left( 1 + \frac{1}{IP} \right)^{IP} - 1 \right) \le \frac{2}{P},$$

and

$$\left( 1 + \frac{1}{IP} \right)^{m_2} \le \left( 1 + \frac{1}{IP} \right)^{IP} \le 3.$$

Or, in the original notation:

$$D_{r,k} \le 2I \left( 18\eta^2 L^2 I \sum_{i=1}^{N} q_{r_t}^i S_{r_0}^i + 36\eta^2 \sigma^2 \right) + 6480\eta^4 L^4 I^5 P \sum_{s=r_0}^{r} \left( 18\eta^2 L^2 I \sum_{i=1}^{N} q_s^i S_{r_0}^i + 36\eta^2 \sigma^2 \right)$$

$$+ 648\eta^2 L^2 I^3 P \sum_{s=r_0}^{r} \left( 15\eta^2 L^2 IP \sum_{i=1}^{N} \left( q_s^i + \frac{1}{N} \right) S_{r_0}^i + 15\eta^2 IP \mathbb{E}\left[ \|\nabla f(\bar{\boldsymbol{x}}_{r_0})\|^2 \Big| \mathcal{Q} \right] + 9\eta^2 \rho^2 \sigma^2 \right)$$

$$\le 36\eta^2 L^2 I^2 \sum_{i=1}^{N} q_r^i S_{r_0}^i + 18 \cdot 6480\eta^6 L^6 I^6 P \sum_{s=r_0}^{r} \sum_{i=1}^{N} q_s^i S_{r_0}^i$$

$$+ 9720\eta^4 L^4 I^4 P^2 \sum_{s=r_0}^{r} \sum_{i=1}^{N} \left( q_s^i + \frac{1}{N} \right) S_{r_0}^i + 9720(r - r_0)\eta^4 L^2 I^4 P^3 \mathbb{E}\left[ \|\nabla f(\bar{\boldsymbol{x}}_{r_0})\|^2 \Big| \mathcal{Q} \right]$$

$$+ \left( 72\eta^2 I + 36 \cdot 6480(r - r_0)\eta^6 L^4 I^5 P + 5832(r - r_0)\eta^4 L^2 I^3 P \rho^2 \right) \sigma^2$$

$$\overset{(i)}{\le} 36\eta^2 L^2 I^2 \sum_{i=1}^{N} q_r^i S_{r_0}^i + 18 \cdot 6480\eta^6 L^6 I^6 P \sum_{s=r_0}^{r_0+P-1} \sum_{i=1}^{N} q_s^i S_{r_0}^i$$

$$+ 9720\eta^4 L^4 I^4 P^2 \sum_{s=r_0}^{r_0+P-1} \sum_{i=1}^{N} \left( q_s^i + \frac{1}{N} \right) S_{r_0}^i$$

$$+ \left( 72\eta^2 I + 36 \cdot 6480\eta^6 L^4 I^5 P^2 + 5832\eta^4 L^2 I^3 P^2 \rho^2 \right) \sigma^2 + 9720\eta^4 L^2 I^4 P^3 \mathbb{E}\left[ \|\nabla f(\bar{\boldsymbol{x}}_{r_0})\|^2 \Big| \mathcal{Q} \right]$$

$$= 36\eta^2 L^2 I^2 \sum_{i=1}^{N} q_r^i S_{r_0}^i + 9720\eta^4 L^4 I^4 P^3 \frac{1}{N} \sum_{i=1}^{N} S_{r_0}^i$$

$$+ \left( 18 \cdot 6480\eta^6 L^6 I^6 P^2 + 9720\eta^4 L^4 I^4 P^3 \right) \sum_{i=1}^{N} \bar{q}_{r_0}^i S_{r_0}^i$$

$$+ \left( 72\eta^2 I + 36 \cdot 6480\eta^6 L^4 I^5 P^2 + 5832\eta^4 L^2 I^3 P^2 \rho^2 \right) \sigma^2 + 9720\eta^4 L^2 I^4 P^3 \mathbb{E}\left[ \|\nabla f(\bar{\boldsymbol{x}}_{r_0})\|^2 \Big| \mathcal{Q} \right]$$

$$\overset{(ii)}{\le} 36\eta^2 L^2 I^2 \sum_{i=1}^{N} q_r^i S_{r_0}^i + 9720\eta^4 L^4 I^4 P^3 \frac{1}{N} \sum_{i=1}^{N} S_{r_0}^i + 9753\eta^4 L^4 I^4 P^3 \sum_{i=1}^{N} \bar{q}_{r_0}^i S_{r_0}^i$$

$$+ \left( 73\eta^2 I + 5832\eta^4 L^2 I^3 P^2 \rho^2 \right) \sigma^2 + 9720\eta^4 L^2 I^4 P^3 \mathbb{E}\left[ \|\nabla f(\bar{\boldsymbol{x}}_{r_0})\|^2 \Big| \mathcal{Q} \right], \tag{21}$$

where $(i)$ uses the fact that the interval $\{r_0, \ldots, r\}$ is contained in the interval $\{r_0, \ldots, r_0 + P - 1\}$ and $(ii)$ uses the condition $\eta \leq \frac{1}{60LIP}$ to simplify non-dominating terms. Also

$$
M_{r,k} \leq 7I \sum_{r=r_0}^{r} \left( 15\eta^2 L^2 IP \sum_{i=1}^{N} \left( q_s^i + \frac{1}{N} \right) S_{r_0}^i + 15\eta^2 IP\mathbb{E} \left[ \|\nabla f(\bar{\boldsymbol{x}}_{r_0})\|^2 \Big| \mathcal{Q} \right] + 9\eta^2 \rho^2 \sigma^2 \right)
$$

$$
+ 180\eta^2 L^2 I^3 P \sum_{r=r_0}^{r} \left( 18\eta^2 L^2 I \sum_{i=1}^{N} q_s^i S_{r_0}^i + 36\eta^2 \sigma^2 \right)
$$

$$
\leq 105\eta^2 L^2 I^2 P \sum_{r=r_0}^{r} \sum_{i=1}^{N} \left( q_s^i + \frac{1}{N} \right) S_{r_0}^i + 3240\eta^4 L^4 I^4 P \sum_{r=r_0}^{r} \sum_{i=1}^{N} q_s^i S_{r_0}^i
$$

$$
+ \left( 63(r - r_0)\eta^2 I\rho^2 + 6480(r - r_0)\eta^4 L^2 I^3 P \right) \sigma^2 + 105(r - r_0)\eta^2 I^2 P\mathbb{E} \left[ \|\nabla f(\bar{\boldsymbol{x}}_{r_0})\|^2 \Big| \mathcal{Q} \right]
$$

$$
\overset{(i)}{\leq} 105\eta^2 L^2 I^2 P \sum_{r=r_0}^{r_0+P-1} \sum_{i=1}^{N} \left( q_s^i + \frac{1}{N} \right) S_{r_0}^i + 3240\eta^4 L^4 I^4 P \sum_{r=r_0}^{r_0+P-1} \sum_{i=1}^{N} q_s^i S_{r_0}^i
$$

$$
+ \left( 63\eta^2 IP\rho^2 + 6480\eta^4 L^2 I^3 P^2 \right) \sigma^2 + 105\eta^2 I^2 P^2 \mathbb{E} \left[ \|\nabla f(\bar{\boldsymbol{x}}_{r_0})\|^2 \Big| \mathcal{Q} \right]
$$

$$
\leq 105\eta^2 L^2 I^2 P^2 \frac{1}{N} \sum_{i=1}^{N} S_{r_0}^i + \left( 105\eta^2 L^2 I^2 P^2 + 3240\eta^4 L^4 I^4 P^2 \right) \sum_{i=1}^{N} \bar{q}_{r_0}^i S_{r_0}^i
$$

$$
+ \left( 63\eta^2 IP\rho^2 + 6480\eta^4 L^2 I^3 P^2 \right) \sigma^2 + 105\eta^2 I^2 P^2 \mathbb{E} \left[ \|\nabla f(\bar{\boldsymbol{x}}_{r_0})\|^2 \Big| \mathcal{Q} \right]
$$

$$
\overset{(ii)}{\leq} 105\eta^2 L^2 I^2 P^2 \frac{1}{N} \sum_{i=1}^{N} S_{r_0}^i + 106\eta^2 L^2 I^2 P^2 \sum_{i=1}^{N} \bar{q}_{r_0}^i S_{r_0}^i
$$

$$
+ \left( 63\eta^2 IP\rho^2 + 6480\eta^4 L^2 I^3 P^2 \right) \sigma^2 + 105\eta^2 I^2 P^2 \mathbb{E} \left[ \|\nabla f(\bar{\boldsymbol{x}}_{r_0})\|^2 \Big| \mathcal{Q} \right], \tag{22}
$$

where $(i)$ and $(ii)$ use the same operations as in Equation 21. Summing Equation 22 and Equation 21,

$$
D_{r,k} + M_{r,k} \leq 36\eta^2 L^2 I^2 \sum_{i=1}^{N} q_r^i S_{r_0}^i + \left( 9720\eta^4 L^4 I^4 P^3 + 105\eta^2 L^2 I^2 P^2 \right) \frac{1}{N} \sum_{i=1}^{N} S_{r_0}^i
$$

$$
+ \left( 9753\eta^4 L^4 I^4 P^3 + 106\eta^2 L^2 I^2 P^2 \right) \sum_{i=1}^{N} \bar{q}_{r_0}^i S_{r_0}^i
$$

$$
+ \left( 73\eta^2 I + 5832\eta^4 L^2 I^3 P^2 \rho^2 + 63\eta^2 IP\rho^2 + 6480\eta^4 L^2 I^3 P^2 \right) \sigma^2
$$

$$
+ \left( 9720\eta^4 L^2 I^4 P^3 + 105\eta^2 I^2 P^2 \right) \mathbb{E} \left[ \|\nabla f(\bar{\boldsymbol{x}}_{r_0})\|^2 \Big| \mathcal{Q} \right]
$$

$$
\leq 36\eta^2 L^2 I^2 \sum_{i=1}^{N} q_r^i S_{r_0}^i + 109\eta^2 L^2 I^2 P^2 \sum_{i=1}^{N} \left( \bar{q}_{r_0}^i + \frac{1}{N} \right) S_{r_0}^i
$$

$$
+ \left( 75\eta^2 I + 65\eta^2 IP\rho^2 \right) \sigma^2 + 108\eta^2 I^2 P^2 \mathbb{E} \left[ \|\nabla f(\bar{\boldsymbol{x}}_{r_0})\|^2 \Big| \mathcal{Q} \right],
$$

where the last inequality uses $\eta \leq \frac{1}{60LIP}$. This proves Equation 3. Equation 4 follows by summing over $r \in \{r_0, \ldots, r_0 + P - 1\}$ and $k \in \{0, \ldots, I - 1\}$, taking total expectation, and applying the condition $\mathbb{E}_{r_0}[\bar{q}_{r_0}^i] = \frac{1}{N}$. $\square$

**Lemma 2.** *If* $\eta \leq \frac{1}{60LIP}$, $\gamma\eta \leq \frac{1}{60LIP}$, $\mathbb{E}_{r_0}[\bar{q}_{r_0}^i] = \frac{1}{N}$, *and* $\mathbb{E} \left[ \sum_{i=1}^{N} \left( v_{r_0}^i \right)^2 \Lambda_{r_0}^i \right] \leq \rho^2$, *then*

$$
\mathbb{E} \left[ \|\bar{\boldsymbol{x}}_{r_0+P} - \bar{\boldsymbol{x}}_{r_0}\|^2 \right] \leq 6\gamma^2 \eta^2 I^2 P^2 \mathbb{E} \left[ \|\nabla f(\bar{\boldsymbol{x}}_{r_0})\|^2 \right] + \gamma\eta IP \left( 5\gamma\eta\rho^2 + 7\eta^2 LI \right) \sigma^2
$$

$$
+ 11\gamma^2 \eta^2 L^2 I^2 P^2 \frac{1}{N} \sum_{i=1}^{N} \mathbb{E} \left[ S_{r_0}^i \right]. \tag{23}
$$

*Proof.* By the algorithm definition,

$$\bar{\boldsymbol{x}}_{r_0+P}-\bar{\boldsymbol{x}}_{r_0} = -\gamma\eta \sum_{r=r_0}^{r_0+P-1} \sum_{i=1}^{N} \sum_{k=0}^{I-1} q_r^i (\nabla F_i(\boldsymbol{x}_{r,k}^i; \xi_{r,k}^i)-\boldsymbol{G}_{r_0}^i+\boldsymbol{G}_{r_0}) = -\gamma\eta \sum_{r=r_0}^{r_0+P-1} \sum_{i=1}^{N} \sum_{k=0}^{I-1} q_r^i \boldsymbol{g}_{r,k}^i.$$

To obtain the variance of the update $\bar{\boldsymbol{x}}_{r_0+P} - \bar{\boldsymbol{x}}_{r_0}$,

$$\mathbb{E}\left[\left\|\sum_{r=r_0}^{r_0+P} \sum_{i=1}^{N} \sum_{k=0}^{I-1} q_r^i \left(\boldsymbol{g}_{r,k}^i - \bar{\boldsymbol{g}}_{r,k}^i\right)\right\|^2\right]$$

$$= \mathbb{E}\left[\left\|\sum_{r=r_0}^{r_0+P} \sum_{i=1}^{N} \sum_{k=0}^{I-1} q_r^i \left((\nabla F_i(\boldsymbol{x}_{r,k}^i; \xi_{r,k}^i) - \nabla f_i(\boldsymbol{x}_{r,k}^i)) - (\boldsymbol{G}_{r_0}^i - \bar{\boldsymbol{G}}_{r_0}^i) + (\boldsymbol{G}_{r_0} - \bar{\boldsymbol{G}}_{r_0})\right)\right\|^2\right]$$

$$\leq \underbrace{2\,\mathbb{E}\left[\left\|\sum_{r=r_0}^{r_0+P} \sum_{i=1}^{N} \sum_{k=0}^{I-1} q_r^i \left(\nabla F_i(\boldsymbol{x}_{r,k}^i; \xi_{r,k}^i) - \nabla f_i(\boldsymbol{x}_{r,k}^i)\right)\right\|^2\right]}_{A_1}$$

$$\underbrace{+ 2\,\mathbb{E}\left[\left\|\sum_{r=r_0}^{r_0+P} \sum_{i=1}^{N} \sum_{k=0}^{I-1} q_r^i \left(-\boldsymbol{G}_{r_0}^i + \bar{\boldsymbol{G}}_{r_0}^i + \boldsymbol{G}_{r_0} - \bar{\boldsymbol{G}}_{r_0}\right)\right\|^2\right]}_{A_2}. \tag{24}$$

We can bound the two terms $A_1$ and $A_2$ separately as follows. For $A_1$,

$$A_1 = 2\mathbb{E}\left[\left\|\sum_{r=r_0}^{r_0+P} \sum_{i=1}^{N} \sum_{k=0}^{I-1} q_r^i \left(\nabla F_i(\boldsymbol{x}_{r,k}^i; \xi_{r,k}^i) - \nabla f_i(\boldsymbol{x}_{r,k}^i)\right)\right\|^2\right]$$

$$= 2\mathbb{E}\left[\mathbb{E}\left[\left\|\sum_{r=r_0}^{r_0+P} \sum_{i=1}^{N} \sum_{k=0}^{I-1} q_r^i \left(\nabla F_i(\boldsymbol{x}_{r,k}^i; \xi_{r,k}^i) - \nabla f_i(\boldsymbol{x}_{r,k}^i)\right)\right\|^2 \Big| \mathcal{Q}\right]\right]$$

$$\overset{(i)}{=} 2\mathbb{E}\left[\sum_{r=r_0}^{r_0+P} \sum_{i=1}^{N} \sum_{k=0}^{I-1} \mathbb{E}\left[\left\|q_r^i \left(\nabla F_i(\boldsymbol{x}_{r,k}^i; \xi_{r,k}^i) - \nabla f_i(\boldsymbol{x}_{r,k}^i)\right)\right\|^2 \Big| \mathcal{Q}\right]\right]$$

$$= 2\sum_{r=r_0}^{r_0+P} \sum_{i=1}^{N} \sum_{k=0}^{I-1} \mathbb{E}\left[(q_r^i)^2\right] \mathbb{E}\left[\left\|\nabla F_i(\boldsymbol{x}_{r,k}^i; \xi_{r,k}^i) - \nabla f_i(\boldsymbol{x}_{r,k}^i)\right\|^2 \Big| \mathcal{Q}\right]$$

$$\leq 2\sigma^2 \sum_{r=r_0}^{r_0+P} \sum_{i=1}^{N} \sum_{k=0}^{I-1} \mathbb{E}\left[(q_r^i)^2\right]$$

$$\leq 2IP\rho^2\sigma^2, \tag{25}$$

where $(i)$ uses the fact that for each $i$, $\left\{q_r^i \left(\nabla F_i(\boldsymbol{x}_{r,k}^i) - \nabla f_i(\boldsymbol{x}_{r,k}^i)\right)\right\}_{r,k}$ is a martingale difference sequence with respect to $\mathcal{G}$ (when conditioned on $\mathcal{Q}$) and that stochastic gradient noise is independent

across clients. For $A_2$,

$$\sum_{r=r_0}^{r_0+P} \sum_{i=1}^{N} \sum_{k=0}^{I-1} q_r^i \left( -\boldsymbol{G}_{r_0}^i - \bar{\boldsymbol{G}}_{r_0}^i + \boldsymbol{G}_{r_0} - \bar{\boldsymbol{G}}_{r_0} \right)$$

$$= -\sum_{r=r_0}^{r_0+P} \sum_{i=1}^{N} \sum_{k=0}^{I-1} q_r^i \left( \boldsymbol{G}_{r_0}^i - \bar{\boldsymbol{G}}_{r_0}^i \right) + \sum_{r=r_0}^{r_0+P} \sum_{i=1}^{N} \sum_{k=0}^{I-1} q_r^i \left( \boldsymbol{G}_{r_0} - \bar{\boldsymbol{G}}_{r_0} \right)$$

$$= -I \sum_{r=r_0}^{r_0+P} \sum_{i=1}^{N} q_r^i \left( \boldsymbol{G}_{r_0}^i - \bar{\boldsymbol{G}}_{r_0}^i \right) + IP \left( \boldsymbol{G}_{r_0} - \bar{\boldsymbol{G}}_{r_0} \right)$$

$$= -IP \sum_{i=1}^{N} \left( \frac{1}{P} \sum_{r=r_0}^{r_0+P} q_r^i \right) \left( \boldsymbol{G}_{r_0}^i - \bar{\boldsymbol{G}}_{r_0}^i \right) + IP \sum_{i=1}^{N} \frac{1}{N} \left( \boldsymbol{G}_{r_0}^i - \bar{\boldsymbol{G}}_{r_0}^i \right)$$

$$= -IP \sum_{i=1}^{N} \left( \bar{q}_{r_0}^i - \frac{1}{N} \right) \left( \boldsymbol{G}_{r_0}^i - \bar{\boldsymbol{G}}_{r_0}^i \right),$$

so

$$A_2 = 2I^2 P^2 \mathbb{E} \left[ \left\| \sum_{i=1}^{N} \left( \bar{q}_{r_0}^i - \frac{1}{N} \right) \left( \boldsymbol{G}_{r_0}^i - \bar{\boldsymbol{G}}_{r_0}^i \right) \right\|^2 \right]$$

$$= 2I^2 P^2 \mathbb{E} \left[ \mathbb{E} \left[ \left\| \sum_{i=1}^{N} \left( \bar{q}_{r_0}^i - \frac{1}{N} \right) \left( \boldsymbol{G}_{r_0}^i - \bar{\boldsymbol{G}}_{r_0}^i \right) \right\|^2 \bigg| \mathcal{Q} \right] \right]$$

$$\overset{(i)}{=} 2I^2 P^2 \mathbb{E} \left[ \sum_{i=1}^{N} \mathbb{E} \left[ \left\| \left( \bar{q}_{r_0}^i - \frac{1}{N} \right) \left( \boldsymbol{G}_{r_0}^i - \bar{\boldsymbol{G}}_{r_0}^i \right) \right\|^2 \bigg| \mathcal{Q} \right] \right]$$

$$= 2I^2 P^2 \mathbb{E} \left[ \sum_{i=1}^{N} \left( \bar{q}_{r_0}^i - \frac{1}{N} \right)^2 \mathbb{E} \left[ \left\| \boldsymbol{G}_{r_0}^i - \bar{\boldsymbol{G}}_{r_0}^i \right\|^2 \bigg| \mathcal{Q} \right] \right]$$

$$= 2I^2 P^2 \mathbb{E} \left[ \sum_{i=1}^{N} \left( \bar{q}_{r_0}^i - \frac{1}{N} \right)^2 \mathbb{E} \left[ \left\| \frac{1}{P \bar{z}_{r_0-P}^i I} \sum_{s=r_0-P}^{r_0-1} \sum_{k=0}^{I-1} z_s^i \left( \nabla F_i(\boldsymbol{y}_{s,k}^i; \zeta_{s,k}^i) - \nabla f_i(\boldsymbol{y}_{s,k}^i) \right) \right\|^2 \bigg| \mathcal{Q} \right] \right]$$

$$= 2I^2 P^2 \mathbb{E} \left[ \sum_{i=1}^{N} \left( \bar{q}_{r_0}^i - \frac{1}{N} \right)^2 \sum_{s=r_0-P}^{r_0-1} \sum_{k=0}^{I-1} \mathbb{E} \left[ \left\| \frac{1}{P \bar{z}_{r_0-P}^i I} z_s^i \left( \nabla F_i(\boldsymbol{y}_{s,k}^i; \zeta_{s,k}^i) - \nabla f_i(\boldsymbol{y}_{s,k}^i) \right) \right\|^2 \bigg| \mathcal{Q} \right] \right]$$

$$= 2I^2 P^2 \mathbb{E} \left[ \sum_{i=1}^{N} \left( \bar{q}_{r_0}^i - \frac{1}{N} \right)^2 \sum_{s=r_0-P}^{r_0-1} \sum_{k=0}^{I-1} \frac{1}{P^2 \left( \bar{z}_{r_0-P}^i \right)^2 I^2} \left( z_s^i \right)^2 \mathbb{E} \left[ \left\| \nabla F_i(\boldsymbol{y}_{s,k}^i; \zeta_{s,k}^i) - \nabla f_i(\boldsymbol{y}_{s,k}^i) \right\|^2 \bigg| \mathcal{Q} \right] \right]$$

$$= 2I^2 P^2 \frac{\sigma^2}{I} \mathbb{E} \left[ \sum_{i=1}^{N} \left( \bar{q}_{r_0}^i - \frac{1}{N} \right)^2 \sum_{s=r_0-P}^{r_0-1} \frac{\left( z_s^i \right)^2}{P^2 \left( \bar{z}_{r_0-P}^i \right)^2} \right]$$

$$= 2IP\sigma^2 \mathbb{E} \left[ \sum_{i=1}^{N} \left( \bar{q}_{r_0}^i - \frac{1}{N} \right)^2 \frac{\frac{1}{P} \sum_{s=r_0-P}^{r_0-1} \left( z_s^i \right)^2}{\left( \frac{1}{P} \sum_{s=r_0-P}^{r_0-1} z_s^i \right)^2} \right] \leq 2IP\rho^2\sigma^2, \tag{26}$$

where $(i)$ uses the fact that, conditioned on $\mathcal{Q}$, the variables $\boldsymbol{G}_{r_0}^i - \bar{\boldsymbol{G}}_{r_0}^i$ depend only on stochastic gradient noise, which is independent across clients, and the last inequality uses the condition $\mathbb{E} \left[ \sum_{i=1}^{N} \left( v_{r_0}^i \right)^2 \Lambda_{r_0}^i \right] \leq \rho^2$. Plugging Equation 25 and Equation 26 into Equation 24 yields

$$\mathbb{E} \left[ \left\| \sum_{r=r_0}^{r_0+P} \sum_{i=1}^{N} \sum_{k=0}^{I-1} q_r^i \left( \boldsymbol{g}_{r,k}^i - \bar{\boldsymbol{g}}_{r,k}^i \right) \right\|^2 \right] \leq 4IP\rho^2\sigma^2. \tag{27}$$

Therefore

$$\mathbb{E}\left[\|\bar{\boldsymbol{x}}_{r_0+P} - \bar{\boldsymbol{x}}_{r_0}\|^2\right]$$

$$= \gamma^2\eta^2\mathbb{E}\left[\left\|\sum_{r=r_0}^{r_0+P-1}\sum_{i=1}^{N}\sum_{k=0}^{I-1} q_r^i \boldsymbol{g}_{r,k}^i\right\|^2\right]$$

$$\leq \gamma^2\eta^2\mathbb{E}\left[\left\|\sum_{r=r_0}^{r_0+P-1}\sum_{i=1}^{N}\sum_{k=0}^{I-1} q_r^i \bar{\boldsymbol{g}}_{r,k}^i\right\|^2\right] + \gamma^2\eta^2\mathbb{E}\left[\left\|\sum_{r=r_0}^{r_0+P-1}\sum_{i=1}^{N}\sum_{k=0}^{I-1} q_r^i \left(\boldsymbol{g}_{r,k}^i - \bar{\boldsymbol{g}}_{r,k}^i\right)\right\|^2\right]$$

$$\overset{(i)}{\leq} \gamma^2\eta^2\mathbb{E}\left[\left\|\sum_{r=r_0}^{r_0+P-1}\sum_{i=1}^{N}\sum_{k=0}^{I-1} q_r^i \bar{\boldsymbol{g}}_{r,k}^i\right\|^2\right] + 4\gamma^2\eta^2 IP\rho^2\sigma^2$$

$$\leq \gamma^2\eta^2 IP \sum_{r=r_0}^{r_0+P-1}\sum_{i=1}^{N}\sum_{k=0}^{I-1}\mathbb{E}\left[q_r^i \left\|\nabla f_i(\boldsymbol{x}_{r,k}^i) - \bar{\boldsymbol{G}}_{r_0}^i + \bar{\boldsymbol{G}}_{r_0}\right\|^2\right] + 4\gamma^2\eta^2 IP\rho^2\sigma^2$$

$$\overset{(ii)}{\leq} 5\gamma^2\eta^2 IP \sum_{r=r_0}^{r_0+P-1}\sum_{i=1}^{N}\sum_{k=0}^{I-1}\mathbb{E}\left[q_r^i\left(\|\nabla f(\bar{\boldsymbol{x}}_{r_0})\|^2 + \|\nabla f_i(\boldsymbol{x}_{r,k}^i) - \nabla f_i(\bar{\boldsymbol{x}}_{r,k})\|^2\right.\right.$$

$$\left.\left. + \|\nabla f_i(\bar{\boldsymbol{x}}_{r,k}) - \nabla f_i(\bar{\boldsymbol{x}}_{r_0})\|^2 + \|\nabla f_i(\bar{\boldsymbol{x}}_{r_0}) - \bar{\boldsymbol{G}}_{r_0}^i\|^2 + \|\nabla f(\bar{\boldsymbol{x}}_{r_0}) - \bar{\boldsymbol{G}}_{r_0}\|^2\right)\right] + 4\gamma^2\eta^2 IP\rho^2\sigma^2$$

$$\leq 5\gamma^2\eta^2 I^2 P^2 \mathbb{E}\left[\|\nabla f(\bar{\boldsymbol{x}}_{r_0})\|^2\right]$$

$$+ 5\gamma^2\eta^2 L^2 IP \sum_{r=r_0}^{r_0+P-1}\sum_{k=0}^{I-1}\sum_{i=1}^{N}\left(\mathbb{E}\left[q_r^i\|\boldsymbol{x}_{r,k}^i - \bar{\boldsymbol{x}}_{r,k}\|^2\right] + \mathbb{E}\left[q_r^i\|\bar{\boldsymbol{x}}_{r,k} - \bar{\boldsymbol{x}}_{r_0}\|^2\right]\right)$$

$$+ 5\gamma^2\eta^2 I^2 P^2 \mathbb{E}\left[\|\nabla f(\bar{\boldsymbol{x}}_{r_0}) - \bar{\boldsymbol{G}}_{r_0}\|^2 + \sum_{i=1}^{N}\left(\frac{1}{P}\sum_{r=r_0}^{r_0+P-1} q_r^i\right)\|\nabla f_i(\bar{\boldsymbol{x}}_{r_0}) - \bar{\boldsymbol{G}}_{r_0}^i\|^2\right]$$

$$+ 4\gamma^2\eta^2 IP\rho^2\sigma^2$$

$$\overset{(iii)}{\leq} 5\gamma^2\eta^2 I^2 P^2 \mathbb{E}\left[\|\nabla f(\bar{\boldsymbol{x}}_{r_0})\|^2\right] + 5\gamma^2\eta^2 L^2 IP\mathbb{E}\left[\sum_{r=r_0}^{r_0+P-1}\sum_{k=0}^{I-1} D_{r,k} + M_{r,k}\right]$$

$$+ 10\gamma^2\eta^2 I^2 P^2 \frac{1}{N}\sum_{i=1}^{N}\mathbb{E}\left[\|\nabla f_i(\bar{\boldsymbol{x}}_{r_0}) - \bar{\boldsymbol{G}}_{r_0}^i\|^2\right] + 4\gamma^2\eta^2 IP\rho^2\sigma^2, \tag{28}$$

where $(i)$ uses Equation 27, $(ii)$ uses the decomposition

$$\nabla f_i(\boldsymbol{x}_{r,k}^i) - \bar{\boldsymbol{G}}_{r_0}^i + \bar{\boldsymbol{G}}_{r_0} = \nabla f(\bar{\boldsymbol{x}}_{r_0}) + (\nabla f_i(\boldsymbol{x}_{r,k}^i) - \nabla f_i(\bar{\boldsymbol{x}}_{r,k})) + (\nabla f_i(\bar{\boldsymbol{x}}_{r,k}) - \nabla f_i(\bar{\boldsymbol{x}}_{r_0}))$$
$$+ (\nabla f_i(\bar{\boldsymbol{x}}_{r_0}) - \bar{\boldsymbol{G}}_{r_0}^i) + (\nabla f(\bar{\boldsymbol{x}}_{r_0}) - \bar{\boldsymbol{G}}_{r_0}),$$

and $(iii)$ uses $\boldsymbol{G}_{r_0} = \frac{1}{N} \sum_{i=1}^{N} \boldsymbol{G}_{r_0}^i$ and $\mathbb{E}_{r_0} \left[ \bar{q}_{r_0}^i \right] = \frac{1}{N}$ to obtain

$$
\mathbb{E} \left[ \|\nabla f(\bar{\boldsymbol{x}}_{r_0}) - \bar{\boldsymbol{G}}_{r_0}\|^2 + \sum_{i=1}^{N} \left( \frac{1}{P} \sum_{r=r_0}^{r_0+P-1} q_r^i \right) \|\nabla f_i(\bar{\boldsymbol{x}}_{r_0}) - \bar{\boldsymbol{G}}_{r_0}^i\|^2 \right]
$$

$$
= \mathbb{E} \left[ \mathbb{E}_{r_0} \left[ \|\nabla f(\bar{\boldsymbol{x}}_{r_0}) - \bar{\boldsymbol{G}}_{r_0}\|^2 + \sum_{i=1}^{N} \left( \frac{1}{P} \sum_{r=r_0}^{r_0+P-1} q_r^i \right) \|\nabla f_i(\bar{\boldsymbol{x}}_{r_0}) - \bar{\boldsymbol{G}}_{r_0}^i\|^2 \right] \right]
$$

$$
= \mathbb{E} \left[ \|\nabla f(\bar{\boldsymbol{x}}_{r_0}) - \bar{\boldsymbol{G}}_{r_0}\|^2 + \sum_{i=1}^{N} \mathbb{E}_{r_0} \left[ \frac{1}{P} \sum_{r=r_0}^{r_0+P-1} q_r^i \right] \|\nabla f_i(\bar{\boldsymbol{x}}_{r_0}) - \bar{\boldsymbol{G}}_{r_0}^i\|^2 \right]
$$

$$
= \mathbb{E} \left[ \|\nabla f(\bar{\boldsymbol{x}}_{r_0}) - \bar{\boldsymbol{G}}_{r_0}\|^2 + \frac{1}{N} \sum_{i=1}^{N} \|\nabla f_i(\bar{\boldsymbol{x}}_{r_0}) - \bar{\boldsymbol{G}}_{r_0}^i\|^2 \right]
$$

$$
= \mathbb{E} \left[ \left\| \frac{1}{N} \sum_{i=1}^{N} \left( \nabla f_i(\bar{\boldsymbol{x}}_{r_0}) - \bar{\boldsymbol{G}}_{r_0}^i \right) \right\|^2 + \frac{1}{N} \sum_{i=1}^{N} \|\nabla f_i(\bar{\boldsymbol{x}}_{r_0}) - \bar{\boldsymbol{G}}_{r_0}^i\|^2 \right]
$$

$$
\leq \mathbb{E} \left[ \frac{1}{N} \sum_{i=1}^{N} \|\nabla f_i(\bar{\boldsymbol{x}}_{r_0}) - \bar{\boldsymbol{G}}_{r_0}^i\|^2 + \frac{1}{N} \sum_{i=1}^{N} \|\nabla f_i(\bar{\boldsymbol{x}}_{r_0}) - \bar{\boldsymbol{G}}_{r_0}^i\|^2 \right].
$$

The remaining term in Equation 28 can be simplified as:

$$
\mathbb{E} \left[ \|\nabla f_i(\bar{\boldsymbol{x}}_{r_0}) - \bar{\boldsymbol{G}}_{r_0}^i\|^2 \right] \leq \mathbb{E} \left[ \left\| \nabla f_i(\bar{\boldsymbol{x}}_{r_0}) - \frac{1}{IP} \sum_{r=r_0-P}^{r_0-1} \sum_{k=0}^{I-1} \frac{z_r^i}{\bar{z}_{r_0-P}^i} \nabla f_i(\boldsymbol{y}_{r,k}^i) \right\|^2 \right]
$$

$$
\leq \mathbb{E} \left[ \left\| \frac{1}{IP} \sum_{r=r_0-P}^{r_0-1} \sum_{k=0}^{I-1} \frac{z_r^i}{\bar{z}_{r_0-P}^i} \left( \nabla f_i(\bar{\boldsymbol{x}}_{r_0}) - \nabla f_i(\boldsymbol{y}_{r,k}^i) \right) \right\|^2 \right]
$$

$$
\leq \frac{1}{IP} \sum_{r=r_0-P}^{r_0-1} \sum_{k=0}^{I-1} \mathbb{E} \left[ \frac{z_r^i}{\bar{z}_{r_0-P}^i} \|\nabla f_i(\bar{\boldsymbol{x}}_{r_0}) - \nabla f_i(\boldsymbol{y}_{r,k}^i)\|^2 \right]
$$

$$
\leq \frac{L^2}{IP} \sum_{r=r_0-P}^{r_0-1} \sum_{k=0}^{I-1} \mathbb{E} \left[ \frac{z_r^i}{\bar{z}_{r_0-P}^i} \|\bar{\boldsymbol{x}}_{r_0} - \boldsymbol{y}_{r,k}^i\|^2 \right]
$$

$$
= L^2 \mathbb{E} \left[ \frac{1}{IP} \sum_{r=r_0-P}^{r_0-1} \sum_{k=0}^{I-1} \frac{z_r^i}{\bar{z}_{r_0-P}^i} \mathbb{E} \left[ \|\bar{\boldsymbol{x}}_{r_0} - \boldsymbol{y}_{r,k}^i\|^2 \Big| \mathcal{Q} \right] \right]
$$

$$
= L^2 \mathbb{E} \left[ S_{r_0}^i \right].
$$

Plugging back to Equation 28 yields

$$\mathbb{E}\left[\|\bar{\boldsymbol{x}}_{r_0+P} - \bar{\boldsymbol{x}}_{r_0}\|^2\right]$$

$$\leq 5\gamma^2\eta^2 I^2 P^2 \mathbb{E}\left[\|\nabla f(\bar{\boldsymbol{x}}_{r_0})\|^2\right] + 4\gamma^2\eta^2 IP\rho^2\sigma^2$$

$$+ 5\gamma^2\eta^2 L^2 IP\mathbb{E}\left[\sum_{r=r_0}^{r_0+P-1}\sum_{k=0}^{I-1}(D_{r,k} + M_{r,k})\right] + 10\gamma^2\eta^2 L^2 I^2 P^2 \frac{1}{N}\sum_{i=1}^{N}\mathbb{E}\left[S_{r_0}^i\right]$$

$$\overset{(i)}{\leq} 5\gamma^2\eta^2 I^2 P^2 \mathbb{E}\left[\|\nabla f(\bar{\boldsymbol{x}}_{r_0})\|^2\right] + 4\gamma^2\eta^2 IP\rho^2\sigma^2$$

$$+ 5\gamma^2\eta^2 L^2 IP\left(108\eta^2 I^3 P^3 \mathbb{E}\left[\|\nabla f(\bar{\boldsymbol{x}}_{r_0})\|^2\right] + \left(75\eta^2 I^2 P + 65\eta^2 I^2 P^2\rho^2\right)\sigma^2\right.$$

$$\left.+ 254\eta^2 L^2 I^3 P^3 \frac{1}{N}\sum_{i=1}^{N}\mathbb{E}\left[S_{r_0}^i\right]\right) + 10\gamma^2\eta^2 L^2 I^2 P^2 \frac{1}{N}\sum_{i=1}^{N}\mathbb{E}\left[S_{r_0}^i\right]$$

$$\leq 5\gamma^2\eta^2 I^2 P^2 \left(1 + 108\eta^2 L^2 I^2 P^2\right)\mathbb{E}\left[\|\nabla f(\bar{\boldsymbol{x}}_{r_0})\|^2\right]$$

$$+ \gamma\eta IP\left(4\gamma\eta\rho^2 + 375\gamma\eta^3 L^2 I^2 P + 325\gamma\eta^3 L^2 I^2 P^2\rho^2\right)\sigma^2$$

$$+ 10\gamma^2\eta^2 L^2 I^2 P^2 \left(1 + 127\eta^2 L^2 I^2 P^2\right)\frac{1}{N}\sum_{i=1}^{N}\mathbb{E}\left[S_{r_0}^i\right]$$

$$\overset{(ii)}{\leq} 6\gamma^2\eta^2 I^2 P^2 \mathbb{E}\left[\|\nabla f(\bar{\boldsymbol{x}}_{r_0})\|^2\right] + \gamma\eta IP\left(5\gamma\eta\rho^2 + 7\eta^2 LI\right)\sigma^2 + 11\gamma^2\eta^2 L^2 I^2 P^2 \frac{1}{N}\sum_{i=1}^{N}\mathbb{E}\left[S_{r_0}^i\right].$$

where $(i)$ uses Equation 4 from Lemma 1 and $(ii)$ uses $\eta \leq \frac{1}{60LIP}$ and $\gamma\eta \leq \frac{1}{60LIP}$. This proves Equation 23. $\qquad\square$

**Lemma 3.** *Suppose that* $P_{\mathcal{Q}_{r_0}}(\bar{q}_{r_0}^i > 0) \geq p_{sample}$, $\mathbb{E}\left[w_{r_0}^i \big| \mathcal{Q}_{:r_0}\right] \leq \frac{P^2}{N}$, $\mathbb{E}_{r_0}[\bar{q}_{r_0}^i] = \frac{1}{N}$, *and* $\mathbb{E}\left[\sum_{i=1}^{N}\left(v_{r_0}^i\right)^2 \Lambda_{r_0}^i\right] \leq \rho^2$. *If* $\eta \leq \frac{\sqrt{p_{sample}}}{60LIP}$ *and* $\gamma\eta \leq \frac{p_{sample}}{60LIP}$, *then*

$$\frac{1}{N}\sum_{i=1}^{N}\mathbb{E}\left[S_{r_0+P}^i\right] \leq \left(324\eta^2 I^2 P^2 + \frac{20}{p_{sample}}\gamma^2\eta^2 I^2 P^2\right)\mathbb{E}\left[\|\nabla f(\bar{\boldsymbol{x}}_{r_0})\|^2\right]$$

$$+ \left(226\eta^2 I + 195\eta^2 IP\rho^2 + \frac{17}{p_{sample}}\gamma^2\eta^2 IP\rho^2\right)\sigma^2$$

$$+ \left(1 - \frac{1}{2}p_{sample}\right)\frac{1}{N}\sum_{j=1}^{N}\mathbb{E}\left[S_{r_0}^j\right].$$

*Proof.* Let $1 \leq i \leq N$. We can consider the value $S_{r_0+P}^i$ under two cases: $\bar{q}_{r_0}^i > 0$ and the complement. Let $A_{r_0}^i = \{\bar{q}_{r_0}^i > 0\}$. Denote

$$B_1^i = \mathbb{1}\left\{A_{r_0}^i\right\}\frac{1}{IP\bar{q}_{r_0}^i}\sum_{s=r_0}^{r_0+P-1}\sum_{k=0}^{I-1}q_s^i\mathbb{E}\left[\|\boldsymbol{x}_{s,k}^i - \bar{\boldsymbol{x}}_{r_0+P}\|^2 \Big| \mathcal{Q}\right]$$

$$B_2^i = \mathbb{1}\left\{\bar{A}_{r_0}^i\right\}\frac{1}{IP\bar{z}_{r_0}^i}\sum_{s=r_0}^{r_0+P-1}\sum_{k=0}^{I-1}z_s^i\mathbb{E}\left[\|\boldsymbol{y}_{s,k}^i - \bar{\boldsymbol{x}}_{r_0+P}\|^2 \Big| \mathcal{Q}\right].$$

Then $S_{r_0+P}^i = B_1^i + B_2^i$, and we can consider the two cases separately.

Notice that

$$\|\boldsymbol{x}_{s,k}^i - \bar{\boldsymbol{x}}_{r_0+P}\|^2 \leq 3\|\boldsymbol{x}_{s,k}^i - \bar{\boldsymbol{x}}_{s,k}\|^2 + 3\|\bar{\boldsymbol{x}}_{s,k} - \bar{\boldsymbol{x}}_{r_0}\|^2 + 3\|\bar{\boldsymbol{x}}_{r_0+P} - \bar{\boldsymbol{x}}_{r_0}\|^2.$$

Therefore

$$
B_1^i = \mathbb{1}\left\{A_{r_0}^i\right\} \frac{1}{IP\bar{q}_{r_0}^i} \sum_{s=r_0}^{r_0+P-1} \sum_{k=0}^{I-1} q_s^i \mathbb{E}\left[\left\|\boldsymbol{x}_{s,k}^i - \bar{\boldsymbol{x}}_{r_0+P}\right\|^2 \Big| \mathcal{Q}\right]
$$

$$
\leq \mathbb{1}\left\{A_{r_0}^i\right\} \frac{3}{IP\bar{q}_{r_0}^i} \sum_{s=r_0}^{r_0+P-1} \sum_{k=0}^{I-1} q_s^i \Bigg( \mathbb{E}\left[\|\boldsymbol{x}_{s,k}^i - \bar{\boldsymbol{x}}_{s,k}\|^2 \Big| \mathcal{Q}\right] + \mathbb{E}\left[\|\bar{\boldsymbol{x}}_{s,k} - \bar{\boldsymbol{x}}_{r_0}\|^2 \Big| \mathcal{Q}\right]
$$

$$
+ \mathbb{E}\left[\|\bar{\boldsymbol{x}}_{r_0+P} - \bar{\boldsymbol{x}}_{r_0}\|^2 \Big| \mathcal{Q}\right] \Bigg)
$$

$$
\leq \mathbb{1}\left\{A_{r_0}^i\right\} \frac{3}{IP\bar{q}_{r_0}^i} \sum_{s=r_0}^{r_0+P-1} \sum_{k=0}^{I-1} q_s^i \left(D_{s,k} + M_{s,k}\right) + 3\mathbb{1}\left\{A_{r_0}^i\right\} \mathbb{E}\left[\|\bar{\boldsymbol{x}}_{r_0+P} - \bar{\boldsymbol{x}}_{r_0}\|^2 \Big| \mathcal{Q}\right].
$$

Using Equation 3 from Lemma 1,

$$
\mathbb{1}\left\{A_{r_0}^i\right\} \frac{3}{IP\bar{q}_{r_0}^i} \sum_{s=r_0}^{r_0+P-1} \sum_{k=0}^{I-1} q_s^i \left(D_{s,k} + M_{s,k}\right)
$$

$$
\leq \mathbb{1}\left\{A_{r_0}^i\right\} \frac{3}{IP\bar{q}_{r_0}^i} \sum_{s=r_0}^{r_0+P-1} \sum_{k=0}^{I-1} q_s^i \Bigg( 108\eta^2 I^2 P^2 \mathbb{E}\left[\|\nabla f(\bar{\boldsymbol{x}}_{r_0})\|^2 \Big| \mathcal{Q}\right] + \left(75\eta^2 I + 65\eta^2 IP\rho^2\right)\sigma^2
$$

$$
+ 109\eta^2 L^2 I^2 P^2 \sum_{j=1}^N \left(\bar{q}_{r_0}^j + \frac{1}{N}\right) S_{r_0}^j + 36\eta^2 L^2 I^2 \sum_{j=1}^N q_s^j S_{r_0}^j \Bigg)
$$

$$
\leq 324\eta^2 I^2 P^2 \mathbb{E}\left[\|\nabla f(\bar{\boldsymbol{x}}_{r_0})\|^2 \Big| \mathcal{Q}\right] + \left(225\eta^2 I + 195\eta^2 IP\rho^2\right)\sigma^2
$$

$$
+ 327\eta^2 L^2 I^2 P^2 \sum_{j=1}^N \left(\bar{q}_{r_0}^j + \frac{1}{N}\right) S_{r_0}^j + \mathbb{1}\left\{A_{r_0}^i\right\} \frac{3}{P\bar{q}_{r_0}^i} \sum_{s=r_0}^{r_0+P-1} q_s^i \left( 36\eta^2 L^2 I^2 \sum_{j=1}^N q_s^j S_{r_0}^j \right)
$$

$$
= 324\eta^2 I^2 P^2 \mathbb{E}\left[\|\nabla f(\bar{\boldsymbol{x}}_{r_0})\|^2 \Big| \mathcal{Q}\right] + \left(225\eta^2 I + 195\eta^2 IP\rho^2\right)\sigma^2
$$

$$
+ 327\eta^2 L^2 I^2 P^2 \sum_{j=1}^N \left(\bar{q}_{r_0}^j + \frac{1}{N}\right) S_{r_0}^j + 108\eta^2 L^2 I^2 \sum_{j=1}^N \left( \frac{\mathbb{1}\left\{A_{r_0}^i\right\}}{P\bar{q}_{r_0}^i} \sum_{s=r_0}^{r_0+P-1} q_s^i q_s^j \right) S_{r_0}^j.
$$

Denote $w_{r_0}^{i,j} = \frac{\mathbb{1}\left\{A_{r_0}^i\right\}}{P\bar{q}_{r_0}^i} \sum_{s=r_0}^{r_0+P-1} q_s^i q_s^j$, so that $w_{r_0}^i = \frac{1}{N}\sum_{j=1}^N w_{r_0}^{i,j}$. Then

$$
B_1^i \leq 324\eta^2 I^2 P^2 \mathbb{E}\left[\|\nabla f(\bar{\boldsymbol{x}}_{r_0})\|^2 \Big| \mathcal{Q}\right] + \left(225\eta^2 I + 195\eta^2 IP\rho^2\right)\sigma^2
$$

$$
+ 327\eta^2 L^2 I^2 P^2 \sum_{j=1}^N \left(\bar{q}_{r_0}^j + \frac{1}{N}\right) S_{r_0}^j + 108\eta^2 L^2 I^2 \sum_{j=1}^N \left( \frac{\mathbb{1}\left\{A_{r_0}^i\right\}}{P\bar{q}_{r_0}^i} \sum_{s=r_0}^{r_0+P-1} q_s^i q_s^j \right) S_{r_0}^j
$$

$$
+ 3\mathbb{1}\left\{A_{r_0}^i\right\} \mathbb{E}\left[\|\bar{\boldsymbol{x}}_{r_0+P} - \bar{\boldsymbol{x}}_{r_0}\|^2 \Big| \mathcal{Q}\right]
$$

$$
\mathbb{E}\left[B_1^i \big| \mathcal{Q}_{:r_0}\right] \overset{(i)}{\leq} 324\eta^2 I^2 P^2 \mathbb{E}\left[\|\nabla f(\bar{\boldsymbol{x}}_{r_0})\|^2 \Big| \mathcal{Q}_{:r_0}\right] + \left(225\eta^2 I + 195\eta^2 IP\rho^2\right)\sigma^2
$$

$$
+ 654\eta^2 L^2 I^2 P^2 \frac{1}{N}\sum_{j=1}^N S_{r_0}^j + 108\eta^2 L^2 I^2 \sum_{j=1}^N \mathbb{E}\left[ \frac{\mathbb{1}\left\{A_{r_0}^i\right\}}{P\bar{q}_{r_0}^i} \sum_{s=r_0}^{r_0+P-1} q_s^i q_s^j \Bigg| \mathcal{Q}_{:r_0}\right] S_{r_0}^j
$$

$$
+ 3\mathbb{E}\left[\mathbb{1}\left\{A_{r_0}^i\right\} \|\bar{\boldsymbol{x}}_{r_0+P} - \bar{\boldsymbol{x}}_{r_0}\|^2 \big| \mathcal{Q}_{:r_0}\right]
$$

$$
\mathbb{E}\left[B_1^i\right] \overset{(ii)}{\leq} 324\eta^2 I^2 P^2 \mathbb{E}\left[\|\nabla f(\bar{\boldsymbol{x}}_{r_0})\|^2\right] + \left(225\eta^2 I + 195\eta^2 IP\rho^2\right)\sigma^2
$$

$$
+ 654\eta^2 L^2 I^2 P^2 \frac{1}{N}\sum_{j=1}^N \mathbb{E}\left[S_{r_0}^j\right] + 108\eta^2 L^2 I^2 \sum_{j=1}^N \mathbb{E}\left[\mathbb{E}\left[w_{r_0}^{i,j} \big| \mathcal{Q}_{:r_0}\right] S_{r_0}^j\right]
$$

$$
+ 3\mathbb{E}\left[\mathbb{1}\left\{A_{r_0}^i\right\} \|\bar{\boldsymbol{x}}_{r_0+P} - \bar{\boldsymbol{x}}_{r_0}\|^2\right], \tag{29}
$$

where $(i)$ uses $\mathbb{E}_{r_0}[\bar{q}_{r_0}^i] = \frac{1}{N}$ and the tower property $\mathbb{E}\left[\mathbb{E}\left[\cdot|\mathcal{Q}\right]|\mathcal{Q}_{:r_0}\right] = \mathbb{E}\left[\cdot|\mathcal{Q}_{:r_0}\right]$, and $(ii)$ uses the tower property $\mathbb{E}\left[\mathbb{E}\left[\cdot|\mathcal{Q}_{:r_0}\right]\right] = \mathbb{E}\left[\cdot\right]$.

Now consider $B_2^i$. Under $\bar{A}_{r_0}^i$, $z_r^i = z_{r-P}^i$ and $\boldsymbol{y}_{r,k}^i = \boldsymbol{y}_{r-P,k}^i$ for all $r \in \{r_0, \ldots, r_0 + P - 1\}$. Therefore

$$
\begin{aligned}
B_2^i &= \mathbb{1}\left\{\bar{A}_{r_0}^i\right\} \frac{1}{IP\bar{z}_{r_0}^i} \sum_{s=r_0}^{r_0+P-1} \sum_{k=0}^{I-1} z_s^i \mathbb{E}\left[\left\|\boldsymbol{y}_{s,k}^i - \bar{\boldsymbol{x}}_{r_0+P}\right\|^2 \Big| \mathcal{Q}\right] \\
&= \mathbb{1}\left\{\bar{A}_{r_0}^i\right\} \frac{1}{IP\bar{z}_{r_0-P}^i} \sum_{s=r_0-P}^{r_0-1} \sum_{k=0}^{I-1} z_s^i \mathbb{E}\left[\left\|\boldsymbol{y}_{s,k}^i - \bar{\boldsymbol{x}}_{r_0+P}\right\|^2 \Big| \mathcal{Q}\right] \\
&\overset{(i)}{\leq} \mathbb{1}\left\{\bar{A}_{r_0}^i\right\} \frac{1}{IP\bar{z}_{r_0-P}^i} \sum_{s=r_0-P}^{r_0-1} \sum_{k=0}^{I-1} z_s^i \Bigg((1+\lambda)\mathbb{E}\left[\left\|\boldsymbol{y}_{s,k}^i - \bar{\boldsymbol{x}}_{r_0}\right\|^2 \Big| \mathcal{Q}\right] \\
&\quad + \left(1+\frac{1}{\lambda}\right)\mathbb{E}\left[\left\|\bar{\boldsymbol{x}}_{r_0+P} - \bar{\boldsymbol{x}}_{r_0}\right\|^2 | \mathcal{Q}\right]\Bigg) \\
&= \mathbb{1}\left\{\bar{A}_{r_0}^i\right\}(1+\lambda)\frac{1}{IP\bar{z}_{r_0-P}^i} \sum_{s=r_0-P}^{r_0-1} \sum_{k=0}^{I-1} z_s^i \mathbb{E}\left[\left\|\boldsymbol{y}_{s,k}^i - \bar{\boldsymbol{x}}_{r_0}\right\|^2 \Big| \mathcal{Q}\right] \\
&\quad + \mathbb{1}\left\{\bar{A}_{r_0}^i\right\}\left(1+\frac{1}{\lambda}\right)\mathbb{E}\left[\left\|\bar{\boldsymbol{x}}_{r_0+P} - \bar{\boldsymbol{x}}_{r_0}\right\|^2 \Big| \mathcal{Q}\right] \\
&= \mathbb{1}\left\{\bar{A}_{r_0}^i\right\}(1+\lambda)S_{r_0}^i + \mathbb{1}\left\{\bar{A}_{r_0}^i\right\}\left(1+\frac{1}{\lambda}\right)\mathbb{E}\left[\left\|\bar{\boldsymbol{x}}_{r_0+P} - \bar{\boldsymbol{x}}_{r_0}\right\|^2 \Big| \mathcal{Q}\right]
\end{aligned}
$$

where $(i)$ uses Young's inequality with an arbitrary $\lambda > 0$. Taking conditional expectation $\mathbb{E}[\cdot|\mathcal{Q}_{:r_0}]$ followed by total expectation yields

$$
\mathbb{E}[B_2^i|\mathcal{Q}_{:r_0}] \leq (1 - p_{\text{sample}})(1+\lambda)S_{r_0}^i + \left(1+\frac{1}{\lambda}\right)\mathbb{E}\left[\mathbb{1}\left\{\bar{A}_{r_0}^i\right\}\left\|\bar{\boldsymbol{x}}_{r_0+P} - \bar{\boldsymbol{x}}_{r_0}\right\|^2 \Big|\mathcal{Q}_{:r_0}\right]
$$

$$
\mathbb{E}[B_2^i] \leq (1 - p_{\text{sample}})(1+\lambda)\mathbb{E}\left[S_{r_0}^i\right] + \left(1+\frac{1}{\lambda}\right)\mathbb{E}\left[\mathbb{1}\left\{\bar{A}_{r_0}^i\right\}\left\|\bar{\boldsymbol{x}}_{r_0+P} - \bar{\boldsymbol{x}}_{r_0}\right\|^2\right]. \quad (30)
$$

Adding Equation 29 and Equation 30 yields

$$
\begin{aligned}
\mathbb{E}\left[S_{r_0+P}^i\right] &= \mathbb{E}[B_1^i] + \mathbb{E}[B_2^i] \\
&\leq 324\eta^2 I^2 P^2 \mathbb{E}\left[\left\|\nabla f(\bar{\boldsymbol{x}}_{r_0})\right\|^2\right] + \left(225\eta^2 I + 195\eta^2 IP\rho^2\right)\sigma^2 \\
&\quad + 654\eta^2 L^2 I^2 P^2 \frac{1}{N}\sum_{j=1}^{N} \mathbb{E}\left[S_{r_0}^j\right] + 108\eta^2 L^2 I^2 \sum_{j=1}^{N} \mathbb{E}\left[\mathbb{E}\left[w_{r_0}^{i,j}|\mathcal{Q}_{:r_0}\right]S_{r_0}^j\right] \\
&\quad + (1 - p_{\text{sample}})(1+\lambda)\mathbb{E}\left[S_{r_0}^i\right] + \left(3+\frac{1}{\lambda}\right)\mathbb{E}\left[\left\|\bar{\boldsymbol{x}}_{r_0+P} - \bar{\boldsymbol{x}}_{r_0}\right\|^2\right].
\end{aligned}
$$

Averaging over $i$,

$$\frac{1}{N} \sum_{i=1}^{N} \mathbb{E}\left[S_{r_0+P}^i\right]$$

$$\leq 324\eta^2 I^2 P^2 \mathbb{E}\left[\|\nabla f(\bar{\boldsymbol{x}}_{r_0})\|^2\right] + \left(225\eta^2 I + 195\eta^2 IP\rho^2\right)\sigma^2 + 654\eta^2 L^2 I^2 P^2 \frac{1}{N}\sum_{j=1}^{N} \mathbb{E}\left[S_{r_0}^j\right]$$

$$+ 108\eta^2 L^2 I^2 \sum_{j=1}^{N} \mathbb{E}\left[\mathbb{E}\left[\frac{1}{N}\sum_{i=1}^{N} w_{r_0}^{i,j}\middle|\mathcal{Q}_{:r_0}\right] S_{r_0}^j\right]$$

$$+ \left(1 - p_{\text{sample}}\right)\left(1 + \lambda\right)\frac{1}{N}\sum_{i=1}^{N} \mathbb{E}\left[S_{r_0}^i\right] + \left(3 + \frac{1}{\lambda}\right)\mathbb{E}\left[\|\bar{\boldsymbol{x}}_{r_0+P} - \bar{\boldsymbol{x}}_{r_0}\|^2\right]$$

$$\leq 324\eta^2 I^2 P^2 \mathbb{E}\left[\|\nabla f(\bar{\boldsymbol{x}}_{r_0})\|^2\right] + \left(225\eta^2 I + 195\eta^2 IP\rho^2\right)\sigma^2$$

$$+ \left(\left(1 - p_{\text{sample}}\right)\left(1 + \lambda\right) + 654\eta^2 L^2 I^2 P^2\right)\frac{1}{N}\sum_{j=1}^{N} \mathbb{E}\left[S_{r_0}^j\right]$$

$$+ 108\eta^2 L^2 I^2 \sum_{j=1}^{N} \mathbb{E}\left[\mathbb{E}\left[w_{r_0}^j\middle|\mathcal{Q}_{:r_0}\right] S_{r_0}^j\right] + \left(3 + \frac{1}{\lambda}\right)\mathbb{E}\left[\|\bar{\boldsymbol{x}}_{r_0+P} - \bar{\boldsymbol{x}}_{r_0}\|^2\right]$$

$$\leq 324\eta^2 I^2 P^2 \mathbb{E}\left[\|\nabla f(\bar{\boldsymbol{x}}_{r_0})\|^2\right] + \left(225\eta^2 I + 195\eta^2 IP\rho^2\right)\sigma^2$$

$$+ \left(\left(1 - p_{\text{sample}}\right)\left(1 + \lambda\right) + 762\eta^2 L^2 I^2 P^2\right)\frac{1}{N}\sum_{j=1}^{N} \mathbb{E}\left[S_{r_0}^j\right] + \left(3 + \frac{1}{\lambda}\right)\mathbb{E}\left[\|\bar{\boldsymbol{x}}_{r_0+P} - \bar{\boldsymbol{x}}_{r_0}\|^2\right],$$

where the last inequality uses the condition $\mathbb{E}\left[w_{r_0}^i\middle|\mathcal{Q}_{:r_0}\right] \leq \frac{P^2}{N}$. We can then apply Lemma 2 and choose $\lambda = \frac{3p_{\text{sample}}}{10(1-p_{\text{sample}})}$ to obtain

$$\frac{1}{N} \sum_{i=1}^{N} \mathbb{E}\left[S_{r_0+P}^i\right]$$

$$\leq \left(324\eta^2 I^2 P^2 + \left(3 + \frac{1}{\lambda}\right)6\gamma^2\eta^2 I^2 P^2\right)\mathbb{E}\left[\|\nabla f(\bar{\boldsymbol{x}}_{r_0})\|^2\right]$$

$$+ \left(225\eta^2 I + 195\eta^2 IP\rho^2 + \left(3 + \frac{1}{\lambda}\right)5\gamma^2\eta^2 IP\rho^2 + \left(3 + \frac{1}{\lambda}\right)7\gamma\eta^3 LI^2 P\right)\sigma^2$$

$$+ \left(\left(1 - p_{\text{sample}}\right)\left(1 + \lambda\right) + 762\eta^2 L^2 I^2 P^2 + \left(3 + \frac{1}{\lambda}\right)11\gamma^2\eta^2 L^2 I^2 P^2\right)\frac{1}{N}\sum_{j=1}^{N} \mathbb{E}\left[S_{r_0}^j\right]$$

$$\leq \left(324\eta^2 I^2 P^2 + \frac{20}{p_{\text{sample}}}\gamma^2\eta^2 I^2 P^2\right)\mathbb{E}\left[\|\nabla f(\bar{\boldsymbol{x}}_{r_0})\|^2\right]$$

$$+ \left(225\eta^2 I + 195\eta^2 IP\rho^2 + \frac{17}{p_{\text{sample}}}\gamma^2\eta^2 IP\rho^2 + \frac{23}{p_{\text{sample}}}\gamma\eta^3 LI^2 P\right)\sigma^2$$

$$+ \left(1 - \frac{7}{10}p_{\text{sample}} + 762\eta^2 L^2 I^2 P^2 + \frac{110}{3p_{\text{sample}}}\gamma^2\eta^2 L^2 I^2 P^2\right)\frac{1}{N}\sum_{j=1}^{N} \mathbb{E}\left[S_{r_0}^j\right]$$

$$\leq \left(324\eta^2 I^2 P^2 + \frac{20}{p_{\text{sample}}}\gamma^2\eta^2 I^2 P^2\right)\mathbb{E}\left[\|\nabla f(\bar{\boldsymbol{x}}_{r_0})\|^2\right]$$

$$+ \left(226\eta^2 I + 195\eta^2 IP\rho^2 + \frac{17}{p_{\text{sample}}}\gamma^2\eta^2 IP\rho^2\right)\sigma^2 + \left(1 - \frac{1}{2}p_{\text{sample}}\right)\frac{1}{N}\sum_{j=1}^{N} \mathbb{E}\left[S_{r_0}^j\right],$$

where the last inequality uses the conditions $\eta \leq \frac{\sqrt{p_{\text{sample}}}}{60LIP}$ and $\gamma\eta \leq \frac{p_{\text{sample}}}{60LIP}$, and we used that $3 + \frac{1}{\lambda} \leq \frac{10}{3p_{\text{sample}}}$. $\qquad\qquad\square$

**Theorem 2** (Theorem 1 restated). *Suppose Assumptions 1 and 2 hold, and $\mathbb{E}[w_{r_0}^i | \mathcal{Q}_{:r_0}] \leq \frac{P^2}{N}$, and $\mathbb{E}\left[\sum_{i=1}^N \left(v_{r_0}^i\right)^2 \Lambda_{r_0}^i\right] \leq \rho^2$. If*

$$\gamma\eta \leq \frac{p_{\text{sample}}}{60LIP}, \quad \eta \leq \frac{\sqrt{p_{\text{sample}}}}{60LIP},$$

*then Algorithm 1 satisfies*

$$\frac{P}{R} \sum_{r_0 \in \{0, P, \ldots, R-P\}} \mathbb{E}[\|\nabla f(\bar{\boldsymbol{x}}_{r_0})\|^2] \leq \frac{5\Delta}{\gamma\eta IR} + \left(20\gamma\eta L\rho^2 + 5785\eta^2 L^2 IP\right)\sigma^2.$$

*Proof.* Recall that

$$\bar{\boldsymbol{x}}_{r_0+P} - \bar{\boldsymbol{x}}_{r_0} = -\gamma\eta \sum_{r=r_0}^{r_0+P-1} \sum_{i=1}^N q_r^i \sum_{k=0}^{I-1} \nabla F_i(\boldsymbol{x}_{r,k}^i; \xi_{r,k}^i) + \gamma\eta IP \sum_{i=1}^N \left(\bar{q}_{r_0}^i - \frac{1}{N}\right)\boldsymbol{G}_{r_0}^i \qquad (31)$$

Using the quadratic upper bound for smooth functions and taking total expectation,

$$
\begin{aligned}
\mathbb{E}[f(\bar{\boldsymbol{x}}_{r_0+P}) - f(\bar{\boldsymbol{x}}_{r_0})] &\leq -\gamma\eta\mathbb{E}\left[\left\langle \nabla f(\bar{\boldsymbol{x}}_{r_0}), \sum_{r=r_0}^{r_0+P-1} \sum_{i=1}^N q_r^i \sum_{k=0}^{I-1} \nabla F_i(\boldsymbol{x}_{r,k}^i; \xi_{r,k}^i)\right\rangle\right] \\
&\quad + \gamma\eta IP\mathbb{E}\left[\left\langle \nabla f(\bar{\boldsymbol{x}}_{r_0}), \sum_{i=1}^N \left(\bar{q}_{r_0}^i - \frac{1}{N}\right)\boldsymbol{G}_{r_0}^i\right\rangle\right] \\
&\quad + \frac{L}{2}\mathbb{E}\left[\|\bar{\boldsymbol{x}}_{r_0+P} - \bar{\boldsymbol{x}}_{r_0}\|^2\right] \\
&\overset{(i)}{\leq} -\gamma\eta\mathbb{E}\left[\left\langle \nabla f(\bar{\boldsymbol{x}}_{r_0}), \mathbb{E}_{r_0}\left[\sum_{r=r_0}^{r_0+P-1} \sum_{i=1}^N q_r^i \sum_{k=0}^{I-1} \nabla F_i(\boldsymbol{x}_{r,k}^i; \xi_{r,k}^i)\right]\right\rangle\right] \\
&\quad + \gamma\eta IP\mathbb{E}\left[\left\langle \nabla f(\bar{\boldsymbol{x}}_{r_0}), \sum_{i=1}^N \left(\mathbb{E}_{r_0}\left[\bar{q}_{r_0}^i\right] - \frac{1}{N}\right)\boldsymbol{G}_{r_0}^i\right\rangle\right] \\
&\quad + \frac{L}{2}\mathbb{E}\left[\|\bar{\boldsymbol{x}}_{r_0+P} - \bar{\boldsymbol{x}}_{r_0}\|^2\right] \\
&\overset{(ii)}{\leq} -\gamma\eta\mathbb{E}\left[\left\langle \nabla f(\bar{\boldsymbol{x}}_{r_0}), \mathbb{E}_{r_0}\left[\sum_{r=r_0}^{r_0+P-1} \sum_{i=1}^N q_r^i \sum_{k=0}^{I-1} \nabla f_i(\boldsymbol{x}_{r,k}^i)\right]\right\rangle\right] \\
&\quad + \frac{L}{2}\mathbb{E}\left[\|\bar{\boldsymbol{x}}_{r_0+P} - \bar{\boldsymbol{x}}_{r_0}\|^2\right], \qquad\qquad\qquad (32)
\end{aligned}
$$

where $(i)$ uses the law of total expectation $\mathbb{E}[\cdot] = \mathbb{E}[\mathbb{E}_{r_0}[\cdot]]$, and $(ii)$ uses the condition $\mathbb{E}_{r_0}[\bar{q}^i_{r_0}] = \frac{1}{N}$ together with

$$
\mathbb{E}_{r_0}\left[\sum_{r=r_0}^{r_0+P-1}\sum_{i=1}^{N} q_r^i \sum_{k=0}^{I-1} \nabla F_i(\boldsymbol{x}_{r,k}^i; \xi_{r,k}^i)\right] = \sum_{r=r_0}^{r_0+P-1}\sum_{i=1}^{N}\sum_{k=0}^{I-1}\mathbb{E}_{r_0}\left[q_r^i \nabla F_i(\boldsymbol{x}_{r,k}^i; \xi_{r,k}^i)\right]
$$

$$
= \sum_{r=r_0}^{r_0+P-1}\sum_{i=1}^{N}\sum_{k=0}^{I-1}\mathbb{E}_{r_0}\left[\mathbb{E}\left[q_r^i \nabla F_i(\boldsymbol{x}_{r,k}^i; \xi_{r,k}^i)\big|\boldsymbol{x}_{r,k}^i\right]\right]
$$

$$
= \sum_{r=r_0}^{r_0+P-1}\sum_{i=1}^{N}\sum_{k=0}^{I-1}\mathbb{E}_{r_0}\left[q_r^i\mathbb{E}\left[\nabla F_i(\boldsymbol{x}_{r,k}^i; \xi_{r,k}^i)\big|\boldsymbol{x}_{r,k}^i\right]\right]
$$

$$
= \sum_{r=r_0}^{r_0+P-1}\sum_{i=1}^{N}\sum_{k=0}^{I-1}\mathbb{E}_{r_0}\left[q_r^i \nabla f_i(\boldsymbol{x}_{r,k}^i)\right]
$$

$$
= \mathbb{E}_{r_0}\left[\sum_{r=r_0}^{r_0+P-1}\sum_{i=1}^{N} q_r^i \sum_{k=0}^{I-1} \nabla f_i(\boldsymbol{x}_{r,k}^i)\right].
$$

Focusing on the inner product term of Equation 32:

$$
-\gamma\eta\mathbb{E}\left[\left\langle \nabla f(\bar{\boldsymbol{x}}_{r_0}), \mathbb{E}_{r_0}\left[\sum_{r=r_0}^{r_0+P-1}\sum_{i=1}^{N} q_r^i \sum_{k=0}^{I-1} \nabla f_i(\boldsymbol{x}_{r,k}^i)\right]\right\rangle\right]
$$

$$
= -\frac{\gamma\eta}{IP}\mathbb{E}\left[\left\langle IP\nabla f(\bar{\boldsymbol{x}}_{r_0}), \mathbb{E}_{r_0}\left[\sum_{r=r_0}^{r_0+P-1}\sum_{i=1}^{N} q_r^i \sum_{k=0}^{I-1} \nabla f_i(\boldsymbol{x}_{r,k}^i)\right]\right\rangle\right]
$$

$$
\leq -\frac{\gamma\eta IP}{2}\mathbb{E}\left[\|\nabla f(\bar{\boldsymbol{x}}_{r_0})\|^2\right] + \frac{\gamma\eta}{2IP}\mathbb{E}\left[\left\|\mathbb{E}_{r_0}\left[\sum_{r=r_0}^{r_0+P-1}\sum_{i=1}^{N} q_r^i \sum_{k=0}^{I-1} \nabla f_i(\boldsymbol{x}_{r,k}^i)\right] - IP\nabla f(\bar{\boldsymbol{x}}_{r_0})\right\|^2\right],
$$

$$
\tag{33}
$$

where we used $-\langle a, b\rangle = \frac{1}{2}\|b-a\|^2 - \frac{1}{2}\|a\|^2 - \frac{1}{2}\|b\|^2 \leq \frac{1}{2}\|b-a\|^2 - \frac{1}{2}\|a\|^2$. Also

$$
\mathbb{E}\left[\left\|\mathbb{E}_{r_0}\left[\sum_{r=r_0}^{r_0+P-1}\sum_{i=1}^{N} q_r^i \sum_{k=0}^{I-1}\nabla f_i(\boldsymbol{x}_{r,k}^i)\right] - IP\nabla f(\bar{\boldsymbol{x}}_{r_0})\right\|^2\right]
$$

$$
= \mathbb{E}\left[\left\|\mathbb{E}_{r_0}\left[\sum_{r=r_0}^{r_0+P-1}\sum_{i=1}^{N} q_r^i \sum_{k=0}^{I-1}\left(\nabla f_i(\boldsymbol{x}_{r,k}^i) - \nabla f(\bar{\boldsymbol{x}}_{r_0})\right)\right]\right\|^2\right]
$$

$$
\overset{(i)}{\leq} 3\mathbb{E}\left[\left\|\mathbb{E}_{r_0}\left[\sum_{r=r_0}^{r_0+P-1}\sum_{i=1}^{N} q_r^i \sum_{k=0}^{I-1}\left(\nabla f_i(\boldsymbol{x}_{r,k}^i) - \nabla f_i(\bar{\boldsymbol{x}}_{r,k})\right)\right]\right\|^2\right]
$$

$$
+ 3\mathbb{E}\left[\left\|\mathbb{E}_{r_0}\left[\sum_{r=r_0}^{r_0+P-1}\sum_{i=1}^{N} q_r^i \sum_{k=0}^{I-1}(\nabla f_i(\bar{\boldsymbol{x}}_{r,k}) - \nabla f_i(\bar{\boldsymbol{x}}_{r_0}))\right]\right\|^2\right]
$$

$$
+ 3\mathbb{E}\left[\left\|\mathbb{E}_{r_0}\left[\sum_{r=r_0}^{r_0+P-1}\sum_{i=1}^{N} q_r^i \sum_{k=0}^{I-1}(\nabla f_i(\bar{\boldsymbol{x}}_{r_0}) - \nabla f(\bar{\boldsymbol{x}}_{r_0}))\right]\right\|^2\right]
$$

$$
\overset{(ii)}{\leq} 3\mathbb{E}\left[\left\|\sum_{r=r_0}^{r_0+P-1}\sum_{i=1}^{N} q_r^i \sum_{k=0}^{I-1}\left(\nabla f_i(\boldsymbol{x}_{r,k}^i) - \nabla f_i(\bar{\boldsymbol{x}}_{r,k})\right)\right\|^2\right]
$$

$$
+ 3\mathbb{E}\left[\left\|\sum_{r=r_0}^{r_0+P-1}\sum_{i=1}^{N} q_r^i \sum_{k=0}^{I-1}(\nabla f_i(\bar{\boldsymbol{x}}_{r,k}) - \nabla f_i(\bar{\boldsymbol{x}}_{r_0}))\right\|^2\right]
$$

$$
+ 3\mathbb{E}\left[\left\|I\sum_{i=1}^{N}\mathbb{E}_{r_0}\left[\sum_{r=r_0}^{r_0+P-1} q_r^i\right](\nabla f_i(\bar{\boldsymbol{x}}_{r_0}) - \nabla f(\bar{\boldsymbol{x}}_{r_0}))\right\|^2\right]
$$

$$
\overset{(iii)}{\leq} 3IP\sum_{r=r_0}^{r_0+P-1}\sum_{i=1}^{N}\sum_{k=0}^{I-1}\mathbb{E}\left[q_r^i\left\|\nabla f_i(\boldsymbol{x}_{r,k}^i) - \nabla f_i(\bar{\boldsymbol{x}}_{r,k})\right\|^2\right]
$$

$$
+ 3IP\sum_{r=r_0}^{r_0+P-1}\sum_{i=1}^{N}\sum_{k=0}^{I-1}\mathbb{E}\left[q_r^i\left\|\nabla f_i(\bar{\boldsymbol{x}}_{r,k}) - \nabla f_i(\bar{\boldsymbol{x}}_{r_0})\right\|^2\right]
$$

$$
+ 3I^2P^2\mathbb{E}\left[\left\|\sum_{i=1}^{N}\mathbb{E}_{r_0}\left[\frac{1}{P}\sum_{r=r_0}^{r_0+P-1} q_r^i\right](\nabla f_i(\bar{\boldsymbol{x}}_{r_0}) - \nabla f(\bar{\boldsymbol{x}}_{r_0}))\right\|^2\right]
$$

$$
\overset{(iv)}{\leq} 3L^2IP\sum_{r=r_0}^{r_0+P-1}\sum_{i=1}^{N}\sum_{k=0}^{I-1}\mathbb{E}\left[q_r^i\left\|\boldsymbol{x}_{r,k}^i - \bar{\boldsymbol{x}}_{r,k}\right\|^2\right] + 3L^2IP\sum_{r=r_0}^{r_0+P-1}\sum_{i=1}^{N}\sum_{k=0}^{I-1}\mathbb{E}\left[q_r^i\left\|\bar{\boldsymbol{x}}_{r,k} - \bar{\boldsymbol{x}}_{r_0}\right\|^2\right]
$$

$$
= 3L^2IP\sum_{r=r_0}^{r_0+P-1}\sum_{k=0}^{I-1}\mathbb{E}\left[\sum_{i=1}^{N} q_r^i\mathbb{E}\left[\left\|\boldsymbol{x}_{r,k}^i - \bar{\boldsymbol{x}}_{r,k}\right\|^2\Big|\mathcal{Q}\right]\right]
$$

$$
+ 3L^2IP\sum_{r=r_0}^{r_0+P-1}\sum_{k=0}^{I-1}\mathbb{E}\left[\sum_{i=1}^{N} q_r^i\mathbb{E}\left[\left\|\bar{\boldsymbol{x}}_{r,k} - \bar{\boldsymbol{x}}_{r_0}\right\|^2\Big|\mathcal{Q}\right]\right]
$$

$$
\leq 3L^2IP\mathbb{E}\left[\sum_{r=r_0}^{r_0+P-1}\sum_{k=0}^{I-1}(D_{r,k} + M_{r,k})\right],
$$

where $(i)$ uses the decomposition

$$
\nabla f_i(\boldsymbol{x}_{r,k}^i) - \nabla f(\bar{\boldsymbol{x}}_{r_0}) = (\nabla f_i(\boldsymbol{x}_{r,k}^i) - \nabla f_i(\bar{\boldsymbol{x}}_{r,k})) + (\nabla f_i(\bar{\boldsymbol{x}}_{r,k}) - \nabla f_i(\bar{\boldsymbol{x}}_{r_0})) + (\nabla f_i(\bar{\boldsymbol{x}}_{r_0}) - \nabla f(\bar{\boldsymbol{x}}_{r_0})),
$$

$(ii)$ and $(iii)$ use Jensen's inequality, and $(iv)$ uses smoothness of each $f_i$ along with the condition $\mathbb{E}_{r_0}\left[\bar{q}_{r_0}^i\right] = \frac{1}{N}$. Plugging into Equation 33 yields

$$
-\gamma\eta\mathbb{E}\left[\left\langle \nabla f(\bar{\boldsymbol{x}}_{r_0}), \mathbb{E}_{r_0}\left[\sum_{r=r_0}^{r_0+P-1}\sum_{i=1}^{N}q_r^i\sum_{k=0}^{I-1}\nabla f_i(\boldsymbol{x}_{r,k}^i)\right]\right\rangle\right]
$$

$$
\leq -\frac{\gamma\eta IP}{2}\mathbb{E}\left[\|\nabla f(\bar{\boldsymbol{x}}_{r_0})\|^2\right] + \frac{3}{2}\gamma\eta L^2\mathbb{E}\left[\sum_{r=r_0}^{r_0+P-1}\sum_{k=0}^{I-1}(D_{r,k}+M_{r,k})\right]
$$

$$
\overset{(i)}{\leq} -\frac{\gamma\eta IP}{2}\mathbb{E}\left[\|\nabla f(\bar{\boldsymbol{x}}_{r_0})\|^2\right] + \frac{3}{2}\gamma\eta L^2\Big(108\eta^2 I^3 P^3\mathbb{E}\left[\|\nabla f(\bar{\boldsymbol{x}}_{r_0})\|^2\right]
$$

$$
+ \left(75\eta^2 I^2 P + 65\eta^2 I^2 P^2\rho^2\right)\sigma^2 + 254\eta^2 L^2 I^3 P^3\frac{1}{N}\sum_{i=1}^{N}\mathbb{E}\left[S_{r_0}^i\right]\Big)
$$

$$
\leq \gamma\eta IP\left(-\frac{1}{2}+162\eta^2 L^2 I^2 P^2\right)\mathbb{E}\left[\|\nabla f(\bar{\boldsymbol{x}}_{r_0})\|^2\right] + \gamma\eta IP\left(113\eta^2 L^2 I + 98\eta^2 L^2 IP\rho^2\right)\sigma^2
$$

$$
+ 381\gamma\eta^3 L^4 I^3 P^3\frac{1}{N}\sum_{i=1}^{N}\mathbb{E}\left[S_{r_0}^i\right],
$$

where $(i)$ uses Equation 4 from Lemma 1.

Plugging into Equation 32 and applying Lemma 2 yields

$$
\mathbb{E}[f(\bar{\boldsymbol{x}}_{r_0+P}) - f(\bar{\boldsymbol{x}}_{r_0})]
$$

$$
\leq \gamma\eta IP\left(-\frac{1}{2}+162\eta^2 L^2 I^2 P^2\right)\mathbb{E}\left[\|\nabla f(\bar{\boldsymbol{x}}_{r_0})\|^2\right] + \gamma\eta IP\left(113\eta^2 L^2 I + 98\eta^2 L^2 IP\rho^2\right)\sigma^2
$$

$$
+ 381\gamma\eta^3 L^4 I^3 P^3\frac{1}{N}\sum_{i=1}^{N}\mathbb{E}\left[S_{r_0}^i\right] + \frac{1}{2}L\mathbb{E}\left[\|\bar{\boldsymbol{x}}_{r_0+P} - \bar{\boldsymbol{x}}_{r_0}\|^2\right]
$$

$$
\leq \gamma\eta IP\left(-\frac{1}{2}+162\eta^2 L^2 I^2 P^2\right)\mathbb{E}\left[\|\nabla f(\bar{\boldsymbol{x}}_{r_0})\|^2\right] + \gamma\eta IP\left(113\eta^2 L^2 I + 98\eta^2 L^2 IP\rho^2\right)\sigma^2
$$

$$
+ 381\gamma\eta^3 L^4 I^3 P^3\frac{1}{N}\sum_{i=1}^{N}\mathbb{E}\left[S_{r_0}^i\right] + \frac{1}{2}L\Big(6\gamma^2\eta^2 I^2 P^2\mathbb{E}\left[\|\nabla f(\bar{\boldsymbol{x}}_{r_0})\|^2\right] \tag{34}
$$

$$
+ \gamma\eta IP\left(5\gamma\eta\rho^2 + 7\eta^2 LI\right)\sigma^2 + 11\gamma^2\eta^2 L^2 I^2 P^2\frac{1}{N}\sum_{i=1}^{N}\mathbb{E}\left[S_{r_0}^i\right]\Big)
$$

$$
\leq \gamma\eta IP\left(-\frac{1}{2}+162\eta^2 L^2 I^2 P^2 + 3\gamma\eta LIP\right)\mathbb{E}\left[\|\nabla f(\bar{\boldsymbol{x}}_{r_0})\|^2\right]
$$

$$
+ \gamma\eta IP\left(\frac{5}{2}\gamma\eta L\rho^2 + 117\eta^2 L^2 I + 98\eta^2 L^2 IP\rho^2\right)\sigma^2
$$

$$
+ \left(381\gamma\eta^3 L^4 I^3 P^3 + 6\gamma^2\eta^2 L^3 I^2 P^2\right)\frac{1}{N}\sum_{i=1}^{N}\mathbb{E}\left[S_{r_0}^i\right]
$$

$$
\overset{(i)}{\leq} \gamma\eta IP\left(-\frac{1}{2}+162\eta^2 L^2 I^2 P^2 + 3\gamma\eta LIP\right)\mathbb{E}\left[\|\nabla f(\bar{\boldsymbol{x}}_{r_0})\|^2\right]
$$

$$
+ \gamma\eta IP\left(\frac{5}{2}\gamma\eta L\rho^2 + 117\eta^2 L^2 I + 98\eta^2 L^2 IP\rho^2\right)\sigma^2 + p_{\text{sample}}\gamma\eta L^2 IP\frac{1}{N}\sum_{i=1}^{N}\mathbb{E}\left[S_{r_0}^i\right],
$$

$$
\tag{35}
$$

where $(i)$ uses $\eta \leq \frac{\sqrt{p_{\text{sample}}}}{60LIP}$ and $\gamma\eta \leq \frac{p_{\text{sample}}}{60LIP}$.

Using Lemma 3,

$$2\gamma\eta L^2 IP\frac{1}{N}\sum_{i=1}^{N}\mathbb{E}\left[S_{r_0+P}^i\right]$$

$$\leq 2\gamma\eta L^2 IP\Bigg(\left(324\eta^2 I^2 P^2 + \frac{20}{p_{\text{sample}}}\gamma^2\eta^2 I^2 P^2\right)\mathbb{E}\left[\|\nabla f(\bar{\boldsymbol{x}}_{r_0})\|^2\right]$$

$$+\left(226\eta^2 I + 195\eta^2 IP\rho^2 + \frac{17}{p_{\text{sample}}}\gamma^2\eta^2 IP\rho^2\right)\sigma^2 + \left(1 - \frac{1}{2}p_{\text{sample}}\right)\frac{1}{N}\sum_{j=1}^{N}\mathbb{E}\left[S_{r_0}^j\right]\Bigg)$$

$$\leq \gamma\eta IP\left(648\eta^2 L^2 I^2 P^2 + \frac{40}{p_{\text{sample}}}\gamma^2\eta^2 L^2 I^2 P^2\right)\mathbb{E}\left[\|\nabla f(\bar{\boldsymbol{x}}_{r_0})\|^2\right]$$

$$+\gamma\eta IP\left(552\eta^2 L^2 I + 390\eta^2 L^2 IP\rho^2 + \frac{34}{p_{\text{sample}}}\gamma^2\eta^2 L^2 IP\rho^2\right)\sigma^2$$

$$+2\gamma\eta L^2 IP\frac{1}{N}\sum_{j=1}^{N}\mathbb{E}\left[S_{r_0}^j\right] - p_{\text{sample}}\gamma\eta L^2 IP\frac{1}{N}\sum_{j=1}^{N}\mathbb{E}\left[S_{r_0}^j\right]$$

$$\leq \gamma\eta IP\left(648\eta^2 L^2 I^2 P^2 + \frac{40}{p_{\text{sample}}}\gamma^2\eta^2 L^2 I^2 P^2\right)\mathbb{E}\left[\|\nabla f(\bar{\boldsymbol{x}}_{r_0})\|^2\right]$$

$$+\gamma\eta IP\left(552\eta^2 L^2 I + 390\eta^2 L^2 IP\rho^2 + \gamma\eta L\rho^2\right)\sigma^2$$

$$+2\gamma\eta L^2 IP\frac{1}{N}\sum_{j=1}^{N}\mathbb{E}\left[S_{r_0}^j\right] - p_{\text{sample}}\gamma\eta L^2 IP\frac{1}{N}\sum_{j=1}^{N}\mathbb{E}\left[S_{r_0}^j\right],$$

where the last inequality uses $\gamma\eta \leq \frac{p_{\text{sample}}}{60LIP}$. Adding this to Equation 35 and rearranging,

$$\mathbb{E}[f(\bar{\boldsymbol{x}}_{r_0+P})] + 2\gamma\eta L^2 IP\frac{1}{N}\sum_{i=1}^{N}\mathbb{E}\left[S_{r_0+P}^i\right]$$

$$\leq \left(\mathbb{E}[f(\bar{\boldsymbol{x}}_{r_0})] + 2\gamma\eta L^2 IP\frac{1}{N}\sum_{i=1}^{N}\mathbb{E}\left[S_{r_0}^i\right]\right)$$

$$\gamma\eta IP\left(-\frac{1}{2} + 810\eta^2 L^2 I^2 P^2 + 3\gamma\eta LIP + \frac{40}{p_{\text{sample}}}\eta^2 L^2 I^2 P^2\right)\mathbb{E}\left[\|\nabla f(\bar{\boldsymbol{x}}_{r_0})\|^2\right]$$

$$+\gamma\eta IP\left(\frac{7}{2}\gamma\eta L\rho^2 + 669\eta^2 L^2 I + 488\eta^2 L^2 IP\rho^2\right)\sigma^2$$

$$\leq \left(\mathbb{E}[f(\bar{\boldsymbol{x}}_{r_0})] + 2\gamma\eta L^2 IP\frac{1}{N}\sum_{i=1}^{N}\mathbb{E}\left[S_{r_0}^i\right]\right)$$

$$-\frac{1}{5}\gamma\eta IP\mathbb{E}\left[\|\nabla f(\bar{\boldsymbol{x}}_{r_0})\|^2\right] + \gamma\eta IP\left(4\gamma\eta L\rho^2 + 669\eta^2 L^2 I + 488\eta^2 L^2 IP\rho^2\right)\sigma^2,$$

where we used $\gamma\eta \leq \frac{1}{60LIP}$ and $\eta \leq \frac{1}{60LIP}$. Finally, we can average over $r_0 \in \{0, P, 2P, \ldots, R - P\}$ and rearrange to obtain

$$\frac{P}{R}\sum_{r_0\in\{0,P,\ldots,R-P\}}\mathbb{E}[\|\nabla f(\bar{\boldsymbol{x}}_{r_0})\|^2] \leq 5\frac{\mathbb{E}[f(\bar{\boldsymbol{x}}_0) - f(\bar{\boldsymbol{x}}_R)]}{\gamma\eta IR} + 10\frac{L^2 P}{R}\frac{1}{N}\sum_{i=1}^{N}\mathbb{E}\left[S_0^i - S_R^i\right]$$

$$+ \left(20\gamma\eta L\rho^2 + 3345\eta^2 L^2 I + 2440\eta^2 L^2 IP\rho^2\right)\sigma^2$$

$$\overset{(i)}{\leq} \frac{5\Delta}{\gamma\eta IR} + \left(20\gamma\eta L\rho^2 + 3345\eta^2 L^2 I + 2440\eta^2 L^2 IP\rho^2\right)\sigma^2$$

$$\overset{(ii)}{\leq} \frac{5\Delta}{\gamma\eta IR} + \left(20\gamma\eta L\rho^2 + 5785\eta^2 L^2 IP\right)\sigma^2,$$

where $(i)$ uses $\frac{1}{N}\sum_{i=1}^{N}\mathbb{E}[S_R^i] \geq 0$ and $\frac{1}{N}\sum_{i=1}^{N}\mathbb{E}[S_0^i] = 0$ by initialization, and $(ii)$ uses $P \geq 1$ and $\rho \leq 1$. $\qquad\square$

**Corollary 2** (Corollary 1 restated). *Let $\epsilon > 0$ and $I \geq 1$. Suppose Assumptions 1 and 2 hold, and that $\mathbb{E}[w_{r_0}^i | \mathcal{Q}_{:r_0}] \leq \frac{P^2}{N}$, and $\mathbb{E}\left[\sum_{i=1}^{N} \left(v_{r_0}^i\right)^2 \Lambda_{r_0}^i\right] \leq \rho^2$. If*

$$R \geq \frac{\Delta L \rho^2 \sigma^2}{I\epsilon^4} + \frac{300\Delta LP}{p_{sample}\epsilon^2}$$

$$\eta \leq \frac{1}{264} \min\left\{ \frac{\Delta^{1/4}\rho^{1/2}}{L^{3/4}I^{3/4}R^{1/4}P^{1/2}\sigma^{1/2}}, \frac{\rho\sqrt{p_{sample}}}{LIP} \right\}$$

$$\gamma = \frac{1}{\eta} \min\left\{ \frac{\sqrt{\Delta}}{60\sqrt{LIR}\rho\sigma}, \frac{p_{sample}}{60LIP} \right\},$$

*then Algorithm 1 satisfies*

$$\frac{P}{R} \sum_{r_0 \in \{0, P, \ldots, R-P\}} \mathbb{E}[\|\nabla f(\bar{\boldsymbol{x}}_{r_0})\|^2] \leq 302\epsilon^2.$$

*Proof.* First, $\eta \leq \frac{\rho\sqrt{p_{sample}}}{264LIP} \leq \frac{\sqrt{p_{sample}}}{60LIP}$ and $\gamma\eta \leq \frac{p_{sample}}{60LIP}$ together with Assumptions 1 and 2 imply that the conditions of Theorem 2 are satisfied. Therefore

$$\frac{P}{R} \sum_{r_0 \in \{0, P, \ldots, R-P\}} \mathbb{E}[\|\nabla f(\bar{\boldsymbol{x}}_{r_0})\|^2] \leq \frac{5\Delta}{\gamma\eta IR} + \left(20\gamma\eta L\rho^2 + 5785\eta^2 L^2 IP\right)\sigma^2.$$

From our choice of $\eta$,

$$5785\eta^2 L^2 IP \leq \frac{5785}{264^2} L^2 IP \min\left\{ \frac{\Delta^{1/2}\rho}{L^{3/2}I^{3/2}R^{1/2}P\sigma}, \frac{\rho^2 p_{sample}}{L^2 I^2 P^2} \right\}$$

$$\leq 5L\rho^2 \min\left\{ \frac{\Delta^{1/2}}{60L^{1/2}I^{1/2}R^{1/2}\rho\sigma}, \frac{p_{sample}}{60LIP} \right\}$$

$$= 5\gamma\eta L\rho^2.$$

Therefore

$$\frac{P}{R} \sum_{r_0 \in \{0, P, \ldots, R-P\}} \mathbb{E}[\|\nabla f(\bar{\boldsymbol{x}}_{r_0})\|^2] \leq \frac{5\Delta}{\gamma\eta IR} + 25\gamma\eta L\rho^2\sigma^2$$

$$\leq \frac{5\Delta}{IR}\left(\frac{60\sqrt{LIR}\rho\sigma}{\sqrt{\Delta}} + \frac{60LIP}{p_{sample}}\right) + 25L\rho^2\sigma^2\frac{\sqrt{\Delta}}{60\sqrt{LIR}\rho\sigma}$$

$$\leq \frac{301\sqrt{\Delta L}\rho\sigma}{\sqrt{IR}} + \frac{300\Delta LP}{p_{sample}R},$$

where we used our choice of $\gamma\eta$. Finally, from our choice of $R$,

$$\frac{P}{R} \sum_{r_0 \in \{0, P, \ldots, R-P\}} \mathbb{E}[\|\nabla f(\bar{\boldsymbol{x}}_{r_0})\|^2] \leq \frac{301\sqrt{\Delta L}\rho\sigma}{\sqrt{I}} \frac{\sqrt{I}\epsilon^2}{\sqrt{\Delta L}\rho\sigma} + \frac{300\Delta LP}{p_{sample}} \frac{p_{sample}\epsilon^2}{300\Delta LP}$$

$$\leq 302\epsilon^2.$$

$\square$

## B  Proofs for Specific Participation Patterns

Here we derive the convergence rates for the specific participation patterns discussed in Section 4.3. In order to apply Corollary 1, the conditions in Assumption 2, Equation 1, and Equation 2 must be satisfied. Since we already showed that Assumption 2 is satisfied by regularized participation and cyclic participation (in Section 3.2), it only remains to show that Equation 1 and Equation 2 is satisfied, and plug the appropriate values of $\rho^2$, $P$, and $p_{sample}$ into Corollary 1.

**Corollary 3** (Regularized Participation). *Under regularized participation, for any $\epsilon > 0$, there exist choices of $\eta, \gamma, I$, and $R$ such that $\mathbb{E}[\|\nabla f(\hat{\boldsymbol{x}})\|^2] \leq O(\epsilon^2)$ and*

$$R = \mathcal{O}\left(\frac{LP}{\epsilon^2}\right), \quad RI = \mathcal{O}\left(\frac{\Delta L \rho^2 \sigma^2}{\epsilon^4}\right).$$

*Proof.* We first show that Equation 1 and Equation 2 are satisfied under regularized participation. Let $r_0 \in \{0, \ldots, R - P\}$. Using the fact that $\bar{q}_{r_0}^i = \frac{1}{N}$ almost surely:

$$\mathbb{E}\left[w_{r_0}^i \middle| \mathcal{Q}_{:r_0}\right] = \mathbb{E}\left[\frac{1}{N}\sum_{j=1}^N \frac{\mathbb{1}\{\bar{q}_{r_0}^j > 0\}}{P\bar{q}_{r_0}^j}\sum_{s=r_0}^{r_0+P-1} q_s^i q_s^j \middle| \mathcal{Q}_{:r_0}\right] = \frac{1}{P}\mathbb{E}\left[\sum_{j=1}^N \sum_{s=r_0}^{r_0+P-1} q_s^i q_s^j \middle| \mathcal{Q}_{:r_0}\right]$$

$$= \frac{1}{P}\mathbb{E}\left[\sum_{s=r_0}^{r_0+P-1} q_s^i \sum_{j=1}^N q_s^j \middle| \mathcal{Q}_{:r_0}\right] = \frac{1}{P}\mathbb{E}\left[\sum_{s=r_0}^{r_0+P-1} q_s^i \middle| \mathcal{Q}_{:r_0}\right]$$

$$= \mathbb{E}\left[q_{r_0}^i \middle| \mathcal{Q}_{:r_0}\right] = \frac{1}{N} \leq \frac{P^2}{N},$$

where we used the fact that $\sum_{i=1}^N q_r^i = 1$. This shows that Equation 1 is satisfied. Also, $\bar{q}_{r_0}^i = \frac{1}{N}$ means that $v_{r_0}^i = 0$, so that $\mathbb{E}\left[(v_{r_0}^i)^2 \Lambda_{r_0}^i\right] = 0 \leq \rho^2$, and Equation 2 is satisfied. Therefore we can apply Corollary 1. The complexity result follows by plugging in $p_{\text{sample}} = 1$ and choosing $I = \mathcal{O}\left(\frac{\Delta \rho^2 \sigma^2}{P\epsilon^2}\right)$. $\qquad\square$

**Corollary 4** (Cyclic Participation). *Under cyclic participation with $\bar{K}$ groups and $S$ participating clients in each round, for any $\epsilon > 0$, there exist choices of $\eta, \gamma, P, I$, and $R$ such that $\mathbb{E}[\|\nabla f(\hat{\boldsymbol{x}})\|^2] \leq O(\epsilon^2)$ and*

$$R = \mathcal{O}\left(\frac{L\bar{K}}{\epsilon^2}\left(\frac{N}{S}\right)\right), \quad RI = \mathcal{O}\left(\frac{\Delta L \sigma^2}{S\epsilon^4}\right).$$

*Proof.* Again, we show that Equation 1 and Equation 2 are satisfied under cyclic participation. Let $r_0 \in \{0, \ldots, R - P\}$. Denoting $g(i) = \lfloor \frac{\bar{K}}{N}(i - 1)\rfloor$ and $r(i) = r_0 + g(i)$, it holds that, over the rounds $r \in \{r_0, \ldots, r_0 + P - 1\}$, client $i$ belongs to group $g(i)$ and is available only during round $r(i)$. Also, let $\Pi(i)$ denote the set of indices of clients in group $g(i)$. To simplify $w_{r_0}^i$, recall that for client $i$ in group $(r \mod \bar{K})$:

$$q_r^i = \begin{cases} \frac{1}{S} & \text{with probability } \frac{S}{N} \\ 0 & \text{with probability } 1 - \frac{S}{N} \end{cases},$$

and $q_r^i = 0$ for all clients $i$ not in group $(r \mod \bar{K})$. So

$$\mathbb{E}\left[w_{r_0}^i \big| \mathcal{Q}_{:r_0}\right] = \mathbb{E}\left[\frac{1}{N}\sum_{j=1}^N \frac{\mathbb{1}\left\{\bar{q}_{r_0}^j > 0\right\}}{P\bar{q}_{r_0}^j} \sum_{s=r_0}^{r_0+P-1} q_s^i q_s^j \Bigg| \mathcal{Q}_{:r_0}\right]$$

$$= \frac{1}{NP}\mathbb{E}\left[\sum_{s=r_0}^{r_0+P-1} q_s^i \sum_{j=1}^N \frac{\mathbb{1}\left\{\bar{q}_{r_0}^j > 0\right\}}{\bar{q}_{r_0}^j} q_s^j \Bigg| \mathcal{Q}_{:r_0}\right]$$

$$\overset{(i)}{=} \frac{1}{NP}\mathbb{E}\left[q_{r(i)}^i \sum_{j=1}^N \frac{\mathbb{1}\left\{\bar{q}_{r_0}^j > 0\right\}}{\bar{q}_{r_0}^j} q_{r(i)}^j \Bigg| \mathcal{Q}_{:r_0}\right]$$

$$\overset{(ii)}{=} \frac{1}{NP}\mathbb{E}\left[q_{r(i)}^i \sum_{j\in\Pi(i)} \frac{\mathbb{1}\left\{\bar{q}_{r_0}^j > 0\right\}}{\bar{q}_{r_0}^j} q_{r(i)}^j \Bigg| \mathcal{Q}_{:r_0}\right]$$

$$\overset{(iii)}{=} \frac{1}{NP}\mathbb{E}\left[q_{r(i)}^i \sum_{j\in\Pi(i)} \frac{\mathbb{1}\left\{\bar{q}_{r_0}^j > 0\right\}}{1/SP} \frac{1}{S} \Bigg| \mathcal{Q}_{:r_0}\right]$$

$$= \frac{1}{N}\mathbb{E}\left[q_{r(i)}^i \sum_{j\in\Pi(i)} \mathbb{1}\left\{\bar{q}_{r_0}^j > 0\right\} \Bigg| \mathcal{Q}_{:r_0}\right]$$

$$\overset{(iv)}{=} \frac{S}{N}\mathbb{E}\left[q_{r(i)}^i \big| \mathcal{Q}_{:r_0}\right]$$

$$= \frac{S}{N}\left(\frac{S}{N}\frac{1}{S} + \left(1 - \frac{S}{N}\right)\cdot 0\right)$$

$$= \frac{S}{N^2} \le \frac{P2}{N},$$

where $(i)$ uses $q_r^i = 0$ for all $r \ne r(i)$, $(ii)$ uses $q_{r(i)}^j = 0$ for all $j \notin \Pi(i)$, $(iii)$ uses the fact that $\bar{q}_{r_0}^j > 0$ implies $q_{r(i)}^j = \frac{1}{S}$ and $\bar{q}_{r_0}^j = \frac{1}{SP}$, and $(iv)$ uses the fact that $S$ clients are sampled in each round. This shows that Equation 1 is satisfied.

To see that Equation 2 is satisfied:

$$\mathbb{E}\left[\sum_{i=1}^N (v_{r_0}^i)^2 \Lambda_{r_0}^i\right] \overset{(i)}{=} \mathbb{E}\left[\sum_{i=1}^N \mathbb{E}\left[\left(\bar{q}_{r_0}^i - \frac{1}{N}\right)^2 \Bigg| \mathcal{Q}_{:r_0}\right] \Lambda_{r_0}^i\right]$$

$$\overset{(ii)}{=} \frac{1}{SNP}\left(1 - \frac{S\bar{K}}{N}\right)\sum_{i=1}^N \mathbb{E}\left[\Lambda_{r_0}^i\right]$$

$$= \frac{1}{SNP}\left(1 - \frac{S\bar{K}}{N}\right)\sum_{i=1}^N \left(\frac{1}{P}\sum_{s=r_0-P}^{r_0-1} (z_s^i)^2 \Bigg/ \left(\frac{1}{P}\sum_{s=r_0-P}^{r_0-1} z_s^i\right)^2\right)$$

$$\overset{(iii)}{=} \frac{1}{SP}\left(1 - \frac{S\bar{K}}{N}\right)\frac{1}{P}\frac{1}{S^2} \Bigg/ \left(\frac{1}{P}\frac{1}{S}\right)^2$$

$$= \frac{1}{S}\left(1 - \frac{S\bar{K}}{N}\right) \le \frac{1}{S} = \rho^2,$$

where $(i)$ uses the tower property, $(ii)$ uses the variance of $\bar{q}_{r_0}^i$ as computed in Appendix C, and $(iii)$ uses the fact that $\bar{z}_{r_0}^i > 0$ by construction of $\bar{z}_{r_0}^i$, and in this case $z_r^i = \frac{1}{S}$ for exactly one $r \in \{r_0 - P, \ldots, r_0 - 1\}$ and $z_r^i = 0$ for all other $r$. Therefore Equation 2 is satisfied.

This shows we can apply Corollary 1. The complexity result follows by plugging in $p_{\text{sample}} = \frac{S}{N}$, $P = \bar{K}$, $\rho = \frac{1}{\sqrt{S}}$, and choosing $I = \mathcal{O}\left(\frac{\Delta\sigma^2}{\bar{K}N\epsilon^2}\right)$. $\qquad\square$

## C   Complexity of Amplified FedAvg

In this section we derive the computation and communication complexity required for Amplified FedAvg [39] to find an $\epsilon$-stationary point under various client participation patterns, which we listed in Table 1 of the main paper. Table 1 compares the complexity under i.i.d. participation, regularized participation, and cyclic participation. Since i.i.d. participation is a special case of cyclic participation with $\bar{K} = 1$ groups, here we only consider regularized and cyclic participation, and the result for i.i.d. participation follows.

Many works in federated learning characterize data heterogeneity by assuming that there exists a constant $\kappa$ such that

$$\|\nabla f_i(\boldsymbol{x}) - \nabla f(\boldsymbol{x})\| \leq \kappa,$$

for all $\boldsymbol{x}$. The previous analysis of Amplified FedAvg [39] instead assumes an upper bound $\tilde{\delta}(P)$ on a weighted heterogeneity that depends on client sampling. Specifically, they assume that there exists $\tilde{\delta}(P)$ such that

$$\left\| \frac{1}{P} \sum_{r=r_0}^{r_0+P-1} \sum_{i=1}^{N} q_r^i (\nabla f_i(\boldsymbol{x}) - \nabla f(\boldsymbol{x})) \right\|^2 \leq \tilde{\delta}^2(P),$$

for all $\boldsymbol{x}$ and $r_0$. We restate their Corollary 3.2 for convenience:

**Corollary 5** (Corollary 3.2 [39] informally restated)**.** *There exist parameter choices such that Amplified FedAvg satisfies*

$$\min_r \mathbb{E}\left[\|\nabla f(\bar{\boldsymbol{x}}_r)\|^2\right] \leq \mathcal{O}\left( \frac{\sqrt{\Delta L}\rho\sigma}{\sqrt{RI}} + \frac{\Delta LP + \kappa^2}{R} + \frac{\sigma^2}{RIP} + \mathbb{E}\left[\tilde{\delta}^2(P)\right] \right). \qquad (36)$$

As pointed out in their Section 4.2, we can interpret $\tilde{\delta}(P)$ in terms of the conventional heterogeneity constant $\kappa$ as:

$$\tilde{\delta}^2(P) \leq N\kappa^2 \sum_{i=1}^{N} \left( \bar{q}_{r_0}^i - \frac{1}{N} \right)^2.$$

Our Assumption 2(b) implies that $\mathbb{E}_{\mathcal{Q}_{r_0}}[\bar{q}_{r_0}^i] = \frac{1}{N}$. If we choose $v \geq 0$ such that $\text{Var}[\bar{q}_{r_0}^i] \leq v^2$, then we can take expecation of the above to obtain

$$\mathbb{E}\left[\tilde{\delta}^2(P)\right] \leq N^2\kappa^2 v^2.$$

This "conversion" of $\tilde{\delta}(P)$ to $\kappa$ and $v$ will allow us to compare their complexity to that of algorithms that use the conventional heterogeneity assumption by computing $v$ for each participation pattern.

**Regularized Participation**   In this case, $\bar{q}_{r_0}^i = \frac{1}{N}$ almost surely, so that $v = 0$ and accordingly $\tilde{\delta}^2(P) = 0$. Therefore

$$\min_r \mathbb{E}\left[\|\nabla f(\bar{\boldsymbol{x}}_r)\|^2\right] \leq \mathcal{O}\left( \frac{\sqrt{\Delta L}\rho\sigma}{\sqrt{RI}} + \frac{\Delta LP + \kappa^2}{R} + \frac{\sigma^2}{RIP} \right).$$

Therefore the choices

$$R \geq \mathcal{O}\left( \frac{\Delta LP + \kappa^2}{\epsilon^2} \right), \quad RI \geq \mathcal{O}\left( \frac{\Delta L\rho^2\sigma^2}{\epsilon^4} + \frac{\sigma^2}{P\epsilon^2} \right),$$

imply

$$\min_r \mathbb{E}\left[\|\nabla f(\bar{\boldsymbol{x}}_r)\|^2\right] \leq \mathcal{O}\left(\epsilon^2\right).$$

**Cyclic Participation**   We can compute $v$ in terms of the parameters of the participation pattern: number of groups $\bar{K}$ and number of participating clients $S$ in each round. Although we chose $P = \bar{K}$ for Amplified SCAFFOLD, we can satisfy Assumption(b) 2 by choosing $P = m\bar{K}$ for any $m \in \mathbb{N}$, and indeed to achieve $\mathbb{E}[\tilde{\delta}^2(P)] \leq \epsilon^2$ we must choose $P = m\bar{K}$ with $m$ depending on $\epsilon$.

Let $A(i)$ denote the set of rounds in $\{r_0, \ldots, r_0 + P - 1\}$ during which client $i$ is available. Note that $A(i)$ has size $m$, since $P = m\bar{K}$. Then

$$\bar{q}_{r_0}^i - \frac{1}{N} = \frac{1}{P} \sum_{r=r_0}^{r_0+P-1} q_r^i - \frac{1}{N} \stackrel{(i)}{=} \frac{1}{P} \sum_{r \in A(i)} q_r^i - \frac{1}{N} = \frac{1}{m\bar{K}} \sum_{r \in A(i)} q_r^i - \frac{1}{b\bar{K}} = \frac{1}{\bar{K}} \left( \frac{1}{m} \sum_{r \in A(i)} q_r^i - \frac{1}{b} \right),$$

where $(i)$ uses the fact that $q_r^i = 0$ for all $r \notin A(i)$. Therefore

$$v^2 = \frac{1}{\bar{K}^2} \mathbb{E}\left[ \left( \frac{1}{m} \sum_{r \in A(i)} q_r^i - \frac{1}{b} \right)^2 \right]$$

$$\stackrel{(i)}{=} \frac{1}{\bar{K}^2} \frac{1}{m^2} \sum_{r \in A(i)} \mathbb{E}\left[ \left( q_r^i - \frac{1}{b} \right)^2 \right]$$

$$\stackrel{(ii)}{=} \frac{1}{\bar{K}^2} \frac{1}{m^2} \sum_{r \in A(i)} \frac{1}{S^2} \frac{S}{b} \left( 1 - \frac{S}{b} \right)$$

$$= \frac{1}{Sm\bar{K}^2 b} \left( 1 - \frac{S}{b} \right)$$

$$= \frac{1}{SNP} \left( 1 - \frac{S\bar{K}}{N} \right),$$

where $(i)$ uses the fact that $\{q_r^i\}_{r \in A(i)}$ are independent and $(ii)$ uses the fact that $q_r^i$ equals $\frac{1}{S}$ times a Bernoulli variable for $r \in A(i)$.

With a bound for $v^2$, we can bound the remaining term in Equation 36 as follows:

$$\mathbb{E}[\tilde{\delta}^2(P)] \leq N^2 \kappa^2 v^2 \leq \frac{\kappa^2}{P} \frac{N}{S} \left( 1 - \frac{S\bar{K}}{N} \right).$$

Therefore, Amplified FedAvg under cyclic participation can find an $\epsilon$-stationary point with the choices

$$P \geq \max\left\{ \bar{K}, \frac{\kappa^2}{\epsilon^2} \frac{N}{S} \left( 1 - \frac{S\bar{K}}{N} \right) \right\}$$

$$R \geq \mathcal{O}\left( \frac{\Delta L \bar{K} + \kappa^2}{\epsilon^2} + \frac{\Delta L N \kappa^2}{S\epsilon^4} \left( 1 - \frac{S\bar{K}}{N} \right) \right)$$

$$RI \geq \mathcal{O}\left( \frac{\Delta L \sigma^2}{S\epsilon^4} \right).$$

## D   Experiment Details

Here we discuss experimental details deferred from the main body: client sampling parameters, heterogeneity protocol, hyperparameter tuning, definition of the synthetic objective, and specification of the CNN architecture used for the image classification experiments.

### D.1   Client Sampling Parameters

For the synthetic objective, we set the number of groups $\bar{K} = 2$ and the availability time $g = 240$. We set the communication interval $I = 10$ and train for $R = 5000$ rounds.

For Fashion-MNIST, we set the communication interval $I = 30$ and train a logistic regression model for $R = 2000$ rounds, with availability time $g = 4$.

For CIFAR-10, we set the communication interval $I = 5$ and train a two-layer CNN for $R = 12000$ rounds, with availability time $g = 10$.

Table 2: Hyperparameter search ranges and final values.

| | $\gamma$ values | $\gamma\eta$ values | $\mu$ values |
|---|---|---|---|
| **Synthetic** | | | |
| FedAvg | $\{\mathbf{1}\}$ | $\{10^{-6}, \mathbf{10^{-5}}, 10^{-4}, 10^{-3}\}$ | $\{\mathbf{0}\}$ |
| FedProx | $\{\mathbf{1}\}$ | $\{10^{-6}, \mathbf{10^{-5}}, 10^{-4}, 10^{-3}\}$ | $\{\mathbf{0.01}, 0.1, 1.0, 10.0\}$ |
| SCAFFOLD | $\{\mathbf{1}\}$ | $\{10^{-6}, 10^{-5}, \mathbf{10^{-4}}, 10^{-3}\}$ | $\{\mathbf{0}\}$ |
| Amplified FedAvg | $\{1.25, 1.5, 2, \mathbf{3}\}$ | $\{10^{-6}, \mathbf{10^{-5}}, 10^{-4}, 10^{-3}\}$ | $\{\mathbf{0}\}$ |
| Amplified SCAFFOLD | $\{1.25, \mathbf{1.5}, 2, 3\}$ | $\{10^{-6}, 10^{-5}, \mathbf{10^{-4}}, 10^{-3}\}$ | $\{\mathbf{0}\}$ |
| **Fashion-MNIST** | | | |
| FedAvg | $\{\mathbf{1}\}$ | $\{10^{-5}, \mathbf{10^{-4}}, 10^{-3}, 10^{-2}\}$ | $\{\mathbf{0}\}$ |
| FedProx | $\{\mathbf{1}\}$ | $\{10^{-5}, \mathbf{10^{-4}}, 10^{-3}, 10^{-2}\}$ | $\{0.01, 0.1, 1.0, \mathbf{10.0}\}$ |
| SCAFFOLD | $\{\mathbf{1}\}$ | $\{10^{-5}, 10^{-4}, 10^{-3}, \mathbf{10^{-2}}\}$ | $\{\mathbf{0}\}$ |
| Amplified FedAvg | $\{1.25, 1.5, \mathbf{2}, 3\}$ | $\{10^{-5}, \mathbf{10^{-4}}, 10^{-3}, 10^{-2}\}$ | $\{\mathbf{0}\}$ |
| Amplified SCAFFOLD | $\{1.25, \mathbf{1.5}, 2, 3\}$ | $\{10^{-5}, 10^{-4}, 10^{-3}, \mathbf{10^{-2}}\}$ | $\{\mathbf{0}\}$ |
| **CIFAR-10** | | | |
| FedAvg | $\{\mathbf{1}\}$ | $\{10^{-6}, \mathbf{10^{-5}}, 10^{-4}, 10^{-3}\}$ | $\{\mathbf{0}\}$ |
| FedProx | $\{\mathbf{1}\}$ | $\{10^{-6}, \mathbf{10^{-5}}, 10^{-4}, 10^{-3}\}$ | $\{\mathbf{0.01}, 0.1, 1.0, 10.0\}$ |
| SCAFFOLD | $\{\mathbf{1}\}$ | $\{10^{-6}, 10^{-5}, \mathbf{10^{-4}}, 10^{-3}\}$ | $\{\mathbf{0}\}$ |
| Amplified FedAvg | $\{1.25, 1.5, 2, \mathbf{3}\}$ | $\{10^{-6}, 10^{-5}, \mathbf{10^{-4}}, 10^{-3}\}$ | $\{\mathbf{0}\}$ |
| Amplified SCAFFOLD | $\{1.25, \mathbf{1.5}, 2, 3\}$ | $\{10^{-6}, 10^{-5}, 10^{-4}, \mathbf{10^{-3}}\}$ | $\{\mathbf{0}\}$ |

## D.2 Heterogeneity Protocol

The following protocol is commonly used in the literature [17, 39] to convert a dataset into a collection of heterogeneous local datasets according to a data similarity parameter $s$, where $s = 0\%$ creates maximal data heterogeneity across clients, and $s = 100\%$ means that data is allocated to each client uniformly at random.

A single, non-federated dataset (e.g. CIFAR-10) is partitioned into two subsets: $s\%$ of the samples are allocated to an i.i.d. pool and randomly shuffled, and the remaining $(100 - s)\%$ of the sampled are allocated to a non-i.i.d. pool and are sorted by label. Samples are allocated to each client so that $s\%$ of each local dataset comes from the i.i.d. pool, and the remaining $(100 - s)\%$ comes from the non-i.i.d. pool, so that with a small $s$, the majority of each local dataset consists of a small number of labels.

## D.3 Hyperparameter Tuning

For all three experiments in the main body (synthetic objective, Fashion-MNIST, and CIFAR-10), each of the four baselines are individually tuned with grid search. For algorithms that use amplified updates (Amplified FedAvg and Amplified SCAFFOLD), we tune the amplification rate $\gamma$ by searching over a fixed range of values. For the other algorithms (FedAvg and SCAFFOLD), we indicate the lack of amplified updates by setting $\gamma = 1$. We tune $\eta$ by allowing $\gamma\eta$ over a fixed range of values. For FedProx's $\mu$ parameter, we also search over a fixed range of values. The search range and final values for each parameter are written in Table 2, along with the final values adopted for each algorithm.

## D.4 Synthetic Objective

For the synthetic experiment, we use a difficult objective from a lower bound analysis of FedAvg [43]. As defined in the lower bound analysis, the objective is parameterized by $H, \kappa, \sigma, c, \mu,$ and $L$.

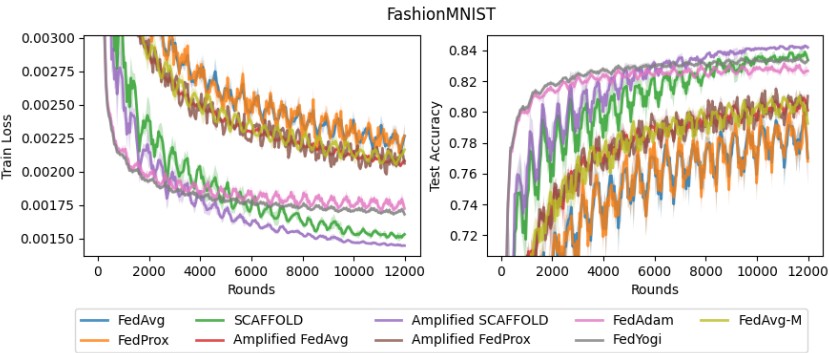

Figure 3: FashionMNIST with additional baselines. Amplified SCAFFOLD maintains the best performance.

The objective maps $\mathbb{R}^4$ to $\mathbb{R}$, and there are only two clients with corresponding local objectives

$$f_1(\boldsymbol{x}) = \frac{\mu}{2}(x_1 - c)^2 + \frac{H}{2}\left(x_2 - \frac{\sqrt{\mu}c}{\sqrt{H}}\right)^2 + \frac{H}{8}\left(x_3^2 + [x_3]_+^2\right) + \kappa x_4$$

$$f_2(\boldsymbol{x}) = \frac{\mu}{2}(x_1 - c)^2 + \frac{H}{2}\left(x_2 - \frac{\sqrt{\mu}c}{\sqrt{H}}\right)^2 + \frac{H}{8}\left(x_3^2 + [x_3]_+^2\right) - \kappa x_4,$$

where $[x]_+ := \max\{x, 0\}$. The stochastic gradients for $f_1$ and $f_2$ are sampled from the distributions $\nabla f_1(\boldsymbol{x}) + \xi e_3$ and $\nabla f_2(\boldsymbol{x}) + \xi e_3$, respectively, where $\xi \sim \mathcal{N}(0, \sigma^2)$ and $e_3$ denotes the third standard basis vector in $\mathbb{R}^4$. We set the parameters of the objective as follows:

$$H = 16, \quad \kappa = 16, \quad \sigma = 1$$
$$c = 1, \quad \mu = 2, \quad L = 2.$$

### D.5 CNN Architecture

We use a simple 2-layer CNN for CIFAR-10. The first layer is a convolutional layer with 64 channels, a $5 \times 5$ kernel, stride of 2, padding of 2, and a ReLU activation. The second layer is a fully connected layer with no activation.

## E  Additional Experimental Results

In this section, we provide two additional experimental results. First, in Section E.1, we add four baselines to the experimental settings of the main paper: FedAdam [32], FedYogi [32], FedAvg-M [5], and Amplified FedAvg with FedProx regularization. Second, in Section E.2, we evaluate all nine algorithms (five from the main paper and the four additional baselines) under another non-i.i.d. client participation pattern for the CIFAR-10 dataset, which we refer to as Stochastic Client Availability (SCA).

### E.1  Additional Baselines

We evaluate the four additional baselines for the Fashion-MNIST and CIFAR-10 experiments from Section 5, keeping the same experimental setup. We tuned the hyperparameters of all baselines according to the hyperparameter ranges suggested in the original paper of each algorithm, and we allow the same compute budget for tuning each baseline as we did for tuning the algorithms in the original paper, in terms of the total number of hyperparameter combinations evaluated. Also, the results are averaged over five random seeds. The results are shown in Figures 3 and 4.

For FashionMNIST, FedAdam and FedYogi reach moderate training loss quickly, but are soon overtaken by Amplified SCAFFOLD and later by SCAFFOLD. FedAvg-M exhibits a minor advantage over FedAvg, but performs about the same as Amplified FedAvg. Amplified FedProx (i.e. Amplified FedAvg with FedProx regularization) performs nearly identically to Amplified FedAvg.

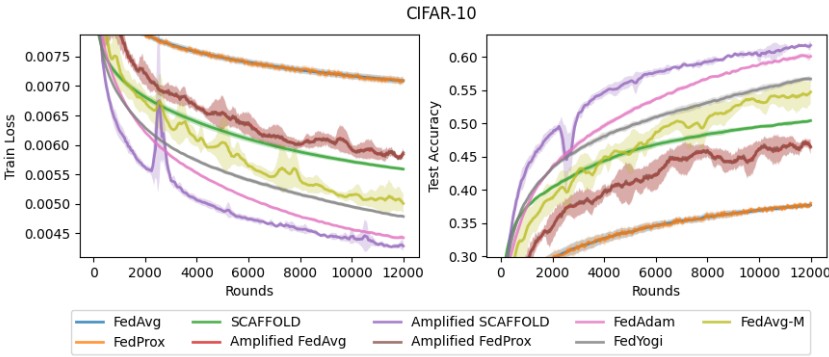

Figure 4: CIFAR-10 with additional baselines. FedAdam is competitive, but Amplified SCAFFOLD maintains superiority.

For CIFAR-10, FedAdam is more competitive, but is still outperformed by Amplified SCAFFOLD. FedYogi and FedAvg-M are further behind, though both still outperform SCAFFOLD. Amplified FedProx is again nearly identical to Amplified FedAvg.

These additional comparisons demonstrate that Amplified SCAFFOLD outperforms strong empirical baselines (FedAdam, FedYogi) under cyclic client participation, reinforcing the empirical validation of our algorithm. This performance is consistent with the fact that Amplified SCAFFOLD has convergence guarantees under periodic participation, while FedAdam and FedYogi were not designed for settings beyond i.i.d. client sampling.

### E.2   CIFAR-10 with Stochastic Client Availability

Here, we include an evaluation under another non-i.i.d. participation pattern, which we refer to as Stochastic Cyclic Availability (SCA). SCA models device availability which is both periodic and unreliable. Similarly to cyclic participation, the set of clients is divided into groups, and at each round one group is deemed the "active" group, while the others are inactive. Unlike cyclic participation, in SCA not every client in the active group is always available: Instead, when a group becomes active, the clients in that group become available for sampling with probability $80\%$, while clients in inactive groups have probability $5\%$ to be available for participation. The active group changes every $g$ rounds. This stochastic availability models the real-life situation where a client device can be unavailable at a time of day when it is usually available, or vice versa. In this way, SCA is more flexible than cyclic participation and better captures the unreliability of client devices. Lastly, we reused the remaining settings ($g$, $\bar{K}$, $I$, etc.) and the tuned hyperparameters for each baseline from the CIFAR-10 experiment under cyclic participation. Again, we average each algorithm's performance over five random seeds.

Results for CIFAR-10 under SCA participation are shown in Figure 5. Again, Amplified SCAFFOLD outperforms all baselines under SCA participation. The relative performance of each baseline is similar as under cyclic participation, with FedAdam staying competitive with Amplified SCAFFOLD, followed by FedYogi and FedAvg-M. The remainder of the baselines have significantly worse performance, and again Amplified FedAvg has not benefitted by adding FedProx regularization.

## F   Extended Comparison with Baselines

Here we include an extended comparison against two relevant prior works.

**FedAvg with Cyclic Participation**   [8] analyzes FedAvg under cyclic participation, but the resulting convergence rate does not benefit from local steps unless regularized participation is satisfied. They analyze FedAvg for $L$-smooth and $\mu$-PL objectives. Using their notation, $\bar{K}$ is the number of client groups, $\kappa$ is the condition number of the objective, $\gamma$ is the intra-group heterogeneity, $M$ is the total number of clients, $N$ is the number of clients that participate in each round, and $T$ is the number of communication rounds. Then the dominating term in their convergence rate for FedAvg under cyclic

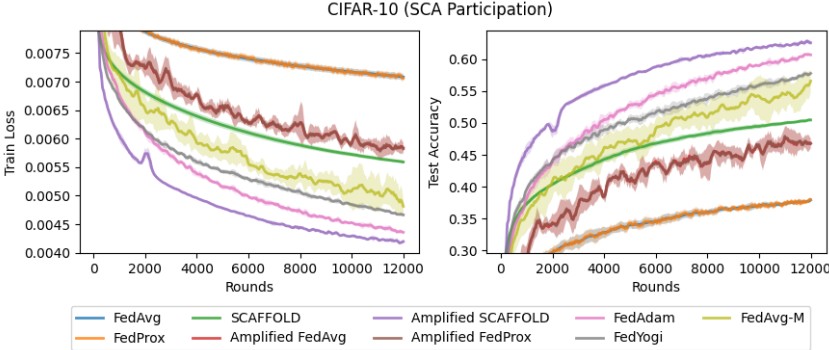

Figure 5: CIFAR-10 under SCA (stochastic cyclic availability). Amplified SCAFFOLD converges fastest.

participation (Theorem 2) is

$$\tilde{\mathcal{O}}\left(\frac{\bar{K}\kappa\gamma^2}{\mu NT}\left(\frac{M/\bar{K}-N}{M/\bar{K}-1}\right)\right).$$

Notice that the number of local steps (denoted $\tau$) does not appear in this dominating term, so there is no way to reduce the communication complexity (compared to parallel SGD) by taking local steps. The only exception is when this term is zero from $N = M/\bar{K}$, which is equivalent to the condition that every client participates within every cycle of availability. Therefore, this result cannot show a benefit from local steps unless the client participation is regularized.

**SCAFFOLD**   In Section 4.3, we mentioned a discrepancy between the complexity of SCAFFOLD vs. Amplified SCAFFOLD in terms of the dependence on $N/S$. In Table 1, the communication complexity of SCAFFOLD and Amplified SCAFFOLD under i.i.d. participation differs by their dependence on $\frac{N}{S}$. The complexity of Amplified SCAFFOLD is $\mathcal{O}\left(\frac{N}{S}\right)$, while that of SCAFFOLD is $\mathcal{O}\left(\left(\frac{N}{S}\right)^{2/3}\right)$. This difference in the order of $\frac{N}{S}$ is due to a potential small issue in the analysis of SCAFFOLD, which we intentionally avoided by accepting a slightly worse dependence on $\frac{N}{S}$.

This difference stems from an apparent mistake in the original SCAFFOLD analysis. In the proof of Lemma 16 (PMLR version of SCAFFOLD), the second-to-last equation of page 32 is obtained with an incorrect step. Namely, while the current version includes

$$\Xi_r = \frac{1}{KN}\sum_{i,k}\mathbb{E}\|\alpha_{i,k-1}^r - \mathbf{x}^r\|^2$$

$$= \left(1 - \frac{S}{N}\right)\frac{1}{KN}\sum_{i,k}\mathbb{E}\|\alpha_{i,k-1}^{r-1} - \mathbf{x}^r\|^2 + \frac{S}{N}\frac{1}{KN}\sum_{i,k}\mathbb{E}\|\mathbf{y}_{i,k-1}^r - \mathbf{x}^r\|^2, \qquad (37)$$

where the last line is obtained by conditioning on the event $\alpha_{i,k-1}^r = \alpha_{i,k-1}^{r-1}$ and the complement $\alpha_{i,k-1}^{r-1} = y_{i,k-1}^r$, which have probabilities $1 - \frac{S}{N}$ and $\frac{S}{N}$, respectively. However, this condition is not denoted in Equation 37. A corrected version should be written

$$\Xi_r = \left(1 - \frac{S}{N}\right)\frac{1}{KN}\sum_{i,k}\mathbb{E}\left[\|\alpha_{i,k-1}^{r-1} - \mathbf{x}^r\|^2 \mid \alpha_{i,k-1}^r = \alpha_{i,k-1}^{r-1}\right]$$

$$+ \frac{S}{N}\frac{1}{KN}\sum_{i,k}\mathbb{E}\left[\|\mathbf{y}_{i,k-1}^r - \mathbf{x}^r\|^2 \mid \alpha_{i,k-1}^{r-1} = y_{i,k-1}^r\right],$$

but this conditional expectation is not specified in the proof of SCAFFOLD. For the remainder of the proof of Lemma 16, these terms are treated as total expectation, leading to an inconsistency. Lemma 16 concludes by applying Lemma 15 in order to bound the term $\mathbb{E}\left[\|\mathbb{E}_{r-1}[\Delta\mathbf{x}^r]\|^2\right]$. However, when we make the correction to include the conditional expectation, we do not have a bound of $\Xi_r$ in terms

of $\mathbb{E}\left[\left\|\mathbb{E}_{r-1}\left[\Delta\mathbf{x}^r\right]\right\|^2\right]$, we instead have a bound in terms of $\mathbb{E}\left[\left\|\mathbb{E}_{r-1}\left[\Delta\mathbf{x}^r \mid \alpha_{i,k-1}^{r-1} = y_{i,k-1}^r\right]\right\|^2\right]$.
But this term with a conditional expectation inside the norm can't be bounded with Lemma 15.

A pessimistic solution is to use Jensen's inequality to bound

$$\mathbb{E}\left[\left\|\mathbb{E}_{r-1}\left[\Delta\mathbf{x}^r \mid \alpha_{i,k-1}^{r-1} = y_{i,k-1}^r\right]\right\|^2\right] \leq \mathbb{E}\left[\mathbb{E}_{r-1}\left[\left\|\Delta\mathbf{x}^r\right\|^2 \mid \alpha_{i,k-1}^{r-1} = y_{i,k-1}^r\right]\right]$$
$$= \mathbb{E}\left[\left\|\Delta\mathbf{x}^r\right\|^2\right],$$

where the second line follows from the tower property. This is the step that we perform in our analysis of Amplified SCAFFOLD, and this results in the $\frac{N}{S}$ dependence.

Fixing their analysis to recover the same $\left(\frac{N}{S}\right)^{2/3}$ dependence may be possible, but we have instead focused on achieving the best known complexity in terms of $\epsilon, \kappa, \sigma$, and other problem parameters.

