# OpenReview forum: "Federated Learning under Periodic Client Participation and Heterogeneous Data: A New Communication-Efficient Algorithm and Analysis"
_NeurIPS.cc/2024/Conference — NeurIPS 2024 poster_

### Official Review · Reviewer_2Xef · 2024-07-08

**Soundness:** 3
**Presentation:** 3
**Contribution:** 3
**Rating:** 6
**Confidence:** 3

**Summary:**

This paper introduces Amplified SCAFFOLD, which is an optimization algorithm for federated learning under periodic client participation. The authors prove that it achieves reduced communication cost, linear speedup, and resilience to data heterogeneity. Numerical experiments are provided to evaluate the performance of Amplified SCAFFOLD.

**Strengths:**

For federated learning under periodic client participation, Amplified SCAFFOLD is proposed and proven to exhibit reduced communication, linear speedup, and resilience to data heterogeneity. Experimental results show that Amplified SCAFFOLD converges faster than baseline algorithms and is robust to changes in data heterogeneity and the number of participating clients.

**Weaknesses:**

1. It would be better to provide a detailed description of the algorithm rather than just an overview.
2. Except for the synthetic data, the experiments are based only on two image datasets (Fashion-MNIST and CIFAR-10) with classification tasks. Adding more experiments would better demonstrate the performance of Amplified SCAFFOLD.

Minor:
1. Line 60-61: Delete "non-uniform average of".
2. Line 99: Assumption 1 (a) is not precise. It should be for all $y$, not for all $x,y$.
3. Line 110: $P$ first appears without explanation.
4. Line 145: Delete "is".

**Questions:**

1. For cycle participation, why are the subsets equally sized? Can they have different sizes? In addition, can the clients sampled with replacement?
2. In the experiments, $\bar{K}$ is chosen to be 5. What about the other choices? What is its influence?

**Limitations:**

As the authors stated, this paper only considers periodic client participation. However, this may not always be true in practice. Clients may arbitrarily join and leave the training, and they may perform different numbers of local steps in each round.

---

> ### Author Rebuttal · Authors · 2024-08-07
>
> Thank you for your helpful comments on our paper. Below we have responded to your comments and questions.
>
> Weaknesses:
>
> 1. **Detailed description of the algorithm.** Thank you for the feedback on our presentation. Section 4.1 is meant to describe the key algorithmic components of Amplified Scaffold, while the concrete definition of the algorithm is left to pseudocode in Algorithm 1. To improve the clarity of the updated version, we can include a detailed description of the algorithm, such as:
>
>     Amplified SCAFFOLD has a similar high-level structure as many FL optimization algorithms: in each round, participating clients individually perform local steps without communication (line 8), then aggregate their updates to the global model (line 12). However, the local steps (line 8) include the control variates $G_{r_0}^i$ and $G_{r_0}$, which are used to modify the local steps in order to approximate the gradient of the global objective. The control variate $G_{r_0}^i$ is the average of stochastic gradients seen by client $i$, but the average is not just taken over the last round: instead, the control variates are computed over windows of $P$ rounds (lines 17-21), in order to ensure that all clients are equally represented (in expectation). The other deviation from the conventional structure of FL algorithms is in the amplified updates (line 15). Over each window of $P$ rounds, the updates from each round are accumulated in the variable $u$. At the end of each window, all clients will have participated with equal frequency (in expectation), so $u$ should contain information from all clients. We then use $u$ to update the global model with a potentially large multiplier $\gamma$ (line 15), leveraging the fact that $u$ contains global information. Together, these two components allow Amplified SCAFFOLD to simultaneously handle non-i.i.d. client participation and heterogeneous objectives.
>
>     Please let us know if this explanation will improve the clarity of our presentation.
>
> 2. **Additional experiments.** In response to your comment about additional experiments, and comments from other reviewers, we have made two additions to the experiments of our submission. First, we added four new baselines (FedAdam, FedYogi, FedAvg-M, and Amplified FedAvg + FedProx) to the evaluation of FashionMNIST and CIFAR-10. Second, we evaluated each algorithm (including these new baselines) under a new non-i.i.d. participation pattern, which models user device availability that is both periodic and unreliable. Both of these experiments are described in detail in the general response, and the loss curves are shown in Figures 1-3 of the 1-page PDF. Here, we highlight the main findings.
>
>     The new baselines FedAdam and FedYogi are competitive for both FashionMNIST and CIFAR-10, but **Amplified SCAFFOLD still maintains the best performance in terms of training loss and testing accuracy across all baselines.**. This is consistent with the fact that FedAdam and FedYogi were designed for i.i.d. client sampling, whereas Amplified SCAFFOLD enjoys convergence guarantees under periodic participation.
>
>     We also evaluate all algorithms under a non-i.i.d. participation pattern which we refer to as Stochastic Client Availability (SCA). On CIFAR-10 with SCA participation, Amplified SCAFFOLD again reaches the best training loss and testing accuracy, showing that **Amplified SCAFFOLD performs well under multiple non-i.i.d. participation patterns**.
>
> Questions:
>
> 1. **Variations on group subsampling.** In your review, you asked "For cycle participation, why are the subsets equally sized? Can they have different sizes?" In our paper and in related works [4, 34], the subsets in cyclic participation are usually equally sized for simplicity. However, this is not a requirement. In fact, for our additional experiment (see general response \#2), the number of available clients randomly sampled and changes throughout training. This participation pattern is more flexible than cyclic participation, and may better model the unpredictability of real-life client devices.
>
>     Also, you asked "In addition, can the clients sampled with replacement?" If we understand your question correctly, you are asking whether the same client can be sampled between two consecutive rounds. If so, the answer is yes. At every round, $S$ clients are sampled, and these clients are not removed from the set of available clients for the next round. In other words, the sampling mechanism does not cycle through all clients before return to previous clients.
>
> 2. **Effect of $\bar{K}$.** In your review, you asked "In the experiments, $\bar{K}$ is chosen to be 5. What about the other choices? What is its influence?" We chose $\bar{K} = 5$ following [34], but this parameter can vary. One way to understand the role of this parameter is notice that $\bar{K} = 1$ corresponds to full client availability, while large values of $\bar{K}$ mean that there are many small groups of clients which are usually not available. Therefore, larger values of $\bar{K}$ means that participation is in some sense ``further from i.i.d.". We expect that larger $\bar{K}$ means that the optimization problem is more difficult, and will require more communication rounds, which aligns with the communication cost of Amplified SCAFFOLD in the last row of Table 1.
>
>     In our experiments, data heterogeneity is created by allocating different data distributions to different clients. With $N=250$ clients and $C=10$ classes, the first 25 clients have a majority of their data from class 1, the next 25 clients have a majority of class 2, etc. The choice $\bar{K} = 5$ yields 50 clients per group, so that each group contains clients with a majority label from two different classes. We believe that this is a good intermediate value for $\bar{K}$: it is not so small that we are too close to the i.i.d. regime ($\bar{K} = 1$), but it is not so large that one class out of ten dominates training in each round.

---

> > ### Author Response · Authors · 2024-08-11
> >
> > Thank you for your efforts in reviewing our paper. In our rebuttal, we responded to the questions and concerns in your review. In particular, we included a detailed description of the algorithm which we can use to improve presentation, additional experiments including an evaluation under a new client participation pattern, and a discussion of the parameters determining participation. Please let us know if we have addressed your concerns, or if you have any more questions. We are happy to continue discussion.
> >
> > Best,
> > Authors

---

> > > ### Comment · Reviewer_2Xef · 2024-08-13
> > >
> > > Thank the authors for the rebuttal. I still have some concerns about $K$. Does a larger $K$ imply greater data heterogeneity? If so, it would be better to experimentally demonstrate the impact of $K$, as this would better show the effectiveness of Amplified SCAFFOLD.

---

> > > > ### Author Response · Authors · 2024-08-13
> > > >
> > > > "Does a larger $K$ imply a greater data heterogeneity?" In our experiments, $K$ is not related to the data heterogeneity. Once the data similarity parameter $s$ is fixed, the partitioning of data to individual clients is determined, regardless of the value of $K$. This means that even with different values of $K$, all clients have the same objective function, so heterogeneity remains the same.
> > > >
> > > > Further, our choice of $K$ follows the experimental setup of [(Wang & Ji, 2022)](https://arxiv.org/abs/2205.13648), who also evaluated with a single choice of $K$.

---

> > > > > ### Comment · Reviewer_2Xef · 2024-08-13
> > > > >
> > > > > Thank the authors for the clarification.  Just to confirm, does $\bar{K}$ represent the heterogeneity between participations? Additionally, $\bar{K}$ impacts the communication complexity of Amplified SCAFFOLD. I still believe that evaluating different values of $\bar{K}$ is necessary.

---

> ### Author Response · Authors · 2024-08-13
>
> There seems to be some confusion around the notation, so let us clarify. In our paper, $\bar{K}$ represents the number of groups for a cyclic participation pattern, and we do not use the variable $K$ anywhere in the paper (other than when using the notation of related works in Appendix E, where $K$ represents the number of local steps). In your previous comment you asked about $K$, and we assumed that you meant $\bar{K}$, and we used the notation $K$ to remain consistent with your question. We apologize for the inconsistency. With this in mind, we can restate the answer to your previous question as: the number of groups $\bar{K}$ does not affect the heterogeneity among client data, it only affects how clients participate in training. Again, we follow the experimental setting of [(Wang & Ji, 2022)](https://arxiv.org/abs/2205.13648), which uses a single choice for $\bar{K}$.
>
> The communication complexity of Amplified SCAFFOLD depends on $\bar{K}$, and the same dependence is present in the complexity of Amplified FedAvg, while Amplified FedAvg also contains a much larger term of order $\epsilon^{-4}$. Therefore changes to $\bar{K}$ should not affect the relative performance of Amplified SCAFFOLD compared to Amplified FedAvg. Also, the choice of a single $\bar{K}$ in both our experiments and those of [(Wang & Ji, 2022)](https://arxiv.org/abs/2205.13648) is consistent with practical scenarios where cyclic participation might arise: when device availability corresponds to geographic location, the number of groups $\bar{K}$ should remain fixed even as the number of users $N$ increases, because there are a limited number of time-zones around the world.

---

> > ### Comment · Reviewer_2Xef · 2024-08-13
> >
> > Thank the authors for response. I acknowledge that $\bar{K}$ should be fixed in one setting. However, evaluating different values of $\bar{K}$ does not imply that $\bar{K}$ is dynamic during a single training session. Instead, it allows for a more comprehensive comparison of the performance of Amplified SCAFFOLD across different settings. Since the experiments are simulations, it is both possible and reasonable to choose a value for $\bar{K}$ other than 5.
> >
> > I also acknowledge that changes to $\bar{K}$ should not affect the relative performance of Amplified SCAFFOLD compared to Amplified FedAvg **theoretically**. However, there may be inconsistencies between empirical and theoretical performance. Therefore, I believe it is still necessary to evaluate different values of $\bar{K}$.

---

> ### Author Response · Authors · 2024-08-14
>
> We agree that the theory may not perfectly predict the practical performance, so that it is important to evaluate different values of $\bar{K}$. Below we included the results of an additional experiment that evaluates Amplified SCAFFOLD and the four major baselines considered in the paper: FedAvg, SCAFFOLD, and Amplified FedAvg. We evaluated each algorithm for the Fashion-MNIST dataset, keeping all of the same settings as stated in the main paper, other than setting $P = 24$ and varying $\bar{K}$ over $\bar{K} = 2, 4, 6, 8$. We only change $P$ from $20$ to $24$ so that $P / \bar{K}$ is an integer, for the sake of simplicity. The training loss and testing accuracy reached at the end of training is shown in the tables below:
>
> | Train Loss | $\bar{K}=2$ | $\bar{K}=4$ | $\bar{K}=6$ | $\bar{K}=8$ |
> |---|---|---|---|---|
> | FedAvg | 0.00216 | 0.00222 | 0.00229 | 0.00236 |
> | SCAFFOLD | 0.00156 | 0.00154 | 0.00161 | 0.00160 |
> | Amplified FedAvg | 0.00206 | 0.00208 | 0.00210 | 0.00213 |
> | Amplified SCAFFOLD | **0.00150** | **0.00147** | **0.00150** | **0.00148** |
>
> | Test Accuracy | $\bar{K}=2$ | $\bar{K}=4$ | $\bar{K}=6$ | $\bar{K}=8$ |
> |---|---|---|---|---|
> | FedAvg | 79.596% | 78.527% | 77.69% | 76.999% |
> | SCAFFOLD | 83.064% | 83.357% | 82.695% | 83.072% |
> | Amplified FedAvg | 80.902% | 80.507% | 80.243% | 79.891% |
> | Amplified SCAFFOLD | **83.643%** | **83.964%** | **83.715%** | **83.957%** |
>
> We draw several conclusions from these results:
> 1. Amplified SCAFFOLD achieves the best training accuracy and testing loss across all algorithms, for every choice of $\bar{K}$.
> 2. FedAvg and Amplified FedAvg get worse as $\bar{K}$ increases, while SCAFFOLD and Amplified SCAFFOLD get better as $\bar{K}$ increases. It makes intuitive sense for an algorithm to degrade as $\bar{K}$ increases, since a larger $\bar{K}$ means that the participation is in some sense "further" from i.i.d. participation. Still, Amplified SCAFFOLD (and SCAFFOLD) are able to maintain performance even as $\bar{K}$ increases.
> 3. Amplified SCAFFOLD is the only algorithm whose final training loss with $\bar{K} = 8$ is better than the final training loss with $\bar{K} = 2$.
>
> These experimental results show that Amplified SCAFFOLD is robust to changes in the number of groups $\bar{K}$ under cyclic participation. While the worst-case communication complexity of Amplified SCAFFOLD (listed in Table 1) actually increases with $\bar{K}$, our experiments demonstrate that in practice, Amplified SCAFFOLD can maintain performance as $\bar{K}$ increases. We hope that this addresses your concern that different values of $\bar{K}$ should be evaluated: please let us know if you are satisfied.
>
> Best,
> Authors

---

> > ### Comment · Reviewer_2Xef · 2024-08-14
> >
> > Thank the authors for conducting the additional experiments. It's great to see that Amplified SCAFFOLD outperforms all the baselines across different values of $\bar{K}$. I am willing to increase my score.
> >
> > The conclusion that "SCAFFOLD and Amplified SCAFFOLD get better as $\bar{K}$ increases" might not be accurate. I am curious about how Amplified SCAFFOLD and SCAFFOLD are able to maintain performance as $\bar{K}$ increases. Could the authors provide some insights or reasons behind this observation?

---

> ### Author Response · Authors · 2024-08-14
>
> It is difficult to explain this observation from a theoretical perspective: SCAFFOLD does not have any theoretical guarantees under cyclic participation, while the communication complexity of Amplified SCAFFOLD indeed increases with $\bar{K}$. One explanation is that the complexity in Table 1 represents the worst-case over all objectives satisfying the assumptions, and the performance for this particular task could be much better than the worst case.
>
> Empirically, it appears that the information contained the control variates of SCAFFOLD and Amplified SCAFFOLD is sufficient to avoid the affect of longer participation cycles. Essentially, the modified update direction using the control variate appears to be a good estimator for the global gradient, even when the control variates are not updated for a long time due to larger values of $\bar{K}$. However, it is difficult to say for sure without further investigation.

---

> > ### Comment · Reviewer_2Xef · 2024-08-14
> >
> > Thank the authors for the explanation. It might be interesting to further investigate the underlying reasons.

---

### Official Review · Reviewer_bD8t · 2024-07-12

**Soundness:** 3
**Presentation:** 3
**Contribution:** 3
**Rating:** 6
**Confidence:** 3

**Summary:**

The paper proposes a new algorithm named Amplified SCAFFOLD for federated learning in environments with periodic client participation and heterogeneous data. The authors address the realistic setting where clients (e.g., mobile devices) are not always available for participation. The proposed algorithm aims to achieve linear speedup, reduced communication rounds, and resilience to data heterogeneity under non-convex optimization. The paper includes theoretical analysis demonstrating the benefits of Amplified SCAFFOLD over existing methods and provides experimental results to validate the proposed approach.

**Strengths:**

1. The introduction of Amplified SCAFFOLD is well-motivated. The algorithm's design to handle non-i.i.d. client participation and its ability to provide tighter guarantees than previous work is well-supported by their theoretical analysis.
2. The experiments with synthetic and real-world data (e.g., Fashion-MNIST and CIFAR-10) demonstrate the algorithm's effectiveness and robustness under various conditions.

**Weaknesses:**

1. Some assumption is too strong, i.e., bounded objective function gap $f(\\mathbf{x}) - f\_{\\min}$ for all $\\mathbf{x} \\in \\mathbb{R}^d$ in assumption 1(a), which is not satisfied for coercive objective function, a commonly used condition in optimization. In addition, previous works [1,2] only utilized the gap between the initial point and the optimum $f(\\mathbf{x}_0) - f\_{\\min}$.
2. While the paper provides comparisons with several baselines, it would be helpful to see related works with more recent state-of-the-art methods that might also address similar challenges, e.g., [3].


[1]. Karimireddy, S. P., Kale, S., Mohri, M., Reddi, S., Stich, S., & Suresh, A. T. (2020). Scaffold: Stochastic controlled averaging for federated learning. International conference on machine learning.

[2]. Wang, S., & Ji, M. (2022). A unified analysis of federated learning with arbitrary client participation. Advances in Neural Information Processing Systems.

[3]. Crawshaw, M., Bao, Y., & Liu, M. (2024). Federated learning with client subsampling, data heterogeneity, and unbounded smoothness: A new algorithm and lower bounds. Advances in Neural Information Processing Systems.

**Questions:**

Can the authors explain from a technical point of view how their amplified SCAFFOLD algorithm is robust against data heterogeneity? I can see that the weights of the participating customers are amplified, but no other mechanism to mitigate data heterogeneity has been identified.

**Limitations:**

The authors adequately addressed the limitations.

---

> ### Author Rebuttal · Authors · 2024-08-07
>
> Thank you for you effort in reviewing our paper. Below we have responded to your thoughts and answered your questions.
>
> Weaknesses:
>
> 1. **Meaning of Assumption 1(a).** We addressed this point in the general response: our Assumption 1(a) contains a typo. Actually, the correct version of our Assumption 1(a) matches the references you mentioned, and we will fix this typo in the updated version.
>
> 2. **Comparison against similar methods** In your review, you wrote "it would be helpful to see related works with more recent state-of-the-art methods that might also address similar challenges, e.g. [3]". From the empirical perspective, we included additional experiments to compare against several baselines; see \#1 of the general response for more information. From the theoretical perspective, we are not aware of additional works not cited in our paper that achieve convergence guarantees under non-i.i.d. client participation. The reference you mentioned [3] provides an algorithm for FL under $(L_0, L_1)$-smoothness with i.i.d. client sampling, which is an orthogonal problem to the non-i.i.d. client participation which motivates our paper. In fact, in the smooth setting (i.e. $L_1 = 0$), the algorithm of [3] degenerates to the SCAFFOLD algorithm with i.i.d. client sampling, which we have already compared against. We can refer to this work in the updated version of our paper, and discuss its relation to our work.
>
> Questions:
>
> 1. **How does Amplified SCAFFOLD avoid heterogeneity?** The main algorithmic component that creates robustness against heterogeneity is the joint effect of control variates with amplified updates. We use the control variates to modify the local update for each client in order to approximate an update on the global objective (see Line 8 of Algorithm 1). Our algorithm combines these control variates with amplified updates across windows of multiple rounds; see Section 4.1 for a description of these algorithmic components. Section 4.2 presents convergence guarantees for this algorithm (reduced communication, robustness to heterogeneity, linear speedup), and Section 4.3 explores the implications for various non-i.i.d. participation patterns.

---

> > ### Author Response · Authors · 2024-08-11
> >
> > Thank you for reviewing our paper. In our rebuttal, we responded to your questions and concerns. We pointed out that your skepticism of assumption 1(a) is due to a typo, included an empirical comparison against more baselines (general response) and a discussion of [3] (weakness #2), and described our algorithm's mechanism for avoiding heterogeneity. Please let us know if we have addressed your concerns.
> >
> > Best,
> > Authors

---

> > > ### Comment · Reviewer_bD8t · 2024-08-11
> > >
> > > Thanks to the authors for their response. I appreciate additional experiments made by the authors. However, I'm a bit confused when the authors responded #2: *'[3] provides an algorithm for FL under $(L\_0,L\_1)$-smoothness with i.i.d. client sampling, which is an **orthogonal problem** to the non-i.i.d. client participation'*. They mentioned around line 216 that *'i.i.d. participation is a special case of cyclic participation'*, implying that i.i.d. client sampling should be a subset of non-i.i.d. client participation. Therefore, I think it is appropriate to ask for clarification on the theoretical advancement of this work over the result in [3] in the case of i.i.d. participation. In other words, will the error bound be tighter than what has been proposed in [3] under i.i.d. participation?

---

> > > > ### Author Response · Authors · 2024-08-11
> > > >
> > > > To fully clarify the comparison between our submission and [3], we must fully state the settings of the two papers. Our paper focuses on periodic participation with smooth functions, while [3] focuses on i.i.d. participation with relaxed smooth functions. It is true that i.i.d. participation is a special case of periodic participation, and similarly the class of smooth functions is a subset of the class of relaxed smooth functions. The special case which is common to the settings of these works is i.i.d. participation for smooth functions. In this special case, our work achieves communication cost of $\Delta L \epsilon^{-2} N/S$ (see Table 1), while [3] requires communication cost of at least $O(\epsilon^{-3})$ (see Table 1 of [3]). Notice that [3] reports their communication complexity in terms of $I$, but their best complexity can be computed by choosing the value of $I$ reported in the fourth column of their Table 1. Therefore, **when we reduce to the special case of smooth functions with i.i.d. participation, our algorithm's communication of $O(\epsilon^{-2})$ is a significant improvement over the complexity $O(\epsilon^{-3})$ of [3]**.

---

> > > > > ### Author Response · Authors · 2024-08-13
> > > > >
> > > > > Please let us know if we have answered your question, and if you have any remaining concerns. Since we addressed both weaknesses and the question in your original review, we kindly request that you consider raising the score. Thank you.
> > > > >
> > > > > Best,
> > > > > Authors

---

> > > > > > ### Comment · Reviewer_bD8t · 2024-08-14
> > > > > >
> > > > > > Thanks for the further clarification. This clears my confusion and I do not have questions. Thus, I will increase my score accordingly.

---

### Official Review · Reviewer_ewwQ · 2024-07-22

**Soundness:** 2
**Presentation:** 2
**Contribution:** 2
**Rating:** 6
**Confidence:** 3

**Summary:**

This work addresses the limitations of federated learning under realistic client participation patterns, specifically focusing on nonconvex optimization. The proposed algorithm, Amplified SCAFFOLD, achieves linear speedup, reduced communication, and resilience to data heterogeneity without requiring strong assumptions. Compared to previous methods, it significantly reduces the required communication rounds for finding a $\epsilon$-stationary point in cyclic participation scenarios. The analysis provides tighter guarantees, and experimental results on both synthetic and real-world data demonstrate the algorithm's effectiveness.

**Strengths:**

1.) The paper is well written
2.) There is extensive theoretical analysis
3.) Supported by experiments on baselines.

**Weaknesses:**

See the questions section.

**Questions:**

1.) While setting up the problem in section 3, I think having a notation table would have been beneficial. Because there are a ton of notations in the paper, many of which are defined later.

2.) In Assumption 1.(a) what is the $\Delta$ and what is the intuitive meaning behind this assumption? Does it have anything to do with the non-negativity of the loss function?

3.) Typo: In line 133, the authors have mentioned $\bar{q}_{r0}$.

4.) Do the authors use the assumption of data heterogeneity mentioned in Table 1 in their proof? If not, then how did they avoid to use it?

5.) As the authors say, the work is a culmination of algorithmic components from references [14] and [34] (references from the paper), and I agree that analysis is by no means trivial. So, combining the algorithmic component of Amplified FedAvg paper and Fedprox would also provide a better result. Can this be checked experimentally, or did the authors observe anything?

6.) Since the papers signify the main contribution to achieving communication efficiency, I would also like to see some experiments with algorithms like FedAdam and FedYogi. Along similar lines of work, in [1] (check below), the authors showed that momentum-based methods can reduce client drift and perform better than methods like SCAFFOLD for full participation.

7.) Did the authors also check the efficacy of the proposed algorithm with different participation strategies, as mentioned in the paper?


[1] Cheng, Ziheng, et al. "Momentum benefits non-iid federated learning simply and provably." arXiv preprint arXiv:2306.16504 (2023).

**Limitations:**

Yes, the authors have mentioned the limitations of the work.

---

> ### Author Rebuttal · Authors · 2024-08-07
>
> Thank you for your insightful suggestions. We have responded to your thoughts and questions below.
>
> Questions:
>
> 1\. **Table of notation.** Thank you for this suggestion. We agree that it would make the presentation more clear and we will provide a table of notation in the updated version.
>
> 2\. **Meaning of Assumption 1(a).** As we said in the general repsonse, our Assumption 1(a) actually contains a typo: the correct version should say $f(x_0) - \min_{x \in \mathbb{R}^d} f(x) \leq \Delta$, which is a standard condition in convergence proofs.
>
> 3\. **Typo on line 133.** Thank you for pointing this out. The variable $\bar{q}_{r_0}^i$ first appears on Line 132, and immediately after we refer to its definition in Algorithm 1, but the reference should point to line 17 of Algorithm 1, not line 18.
>
> 4\. **Usage of heterogeneity assumption.** Since multiple reviewers addressed this point, we discussed it in our general response. Our main message is that we do not use any heterogeneity assumption in the proof, because the necessity of this assumption is completely eliminated by the use of control variates. This circumstance is the same as in SCAFFOLD: the analysis of SCAFFOLD does not require a heterogeneity assumption, but such an assumption is referenced in their paper because it is used by baselines.
>
> 5\. and 6\. **Comparison with additional baselines.** In your review, you requested that we compare against additional baselines such as FedAdam, FedYogi, FedAvg-M, and the combination of Amplified FedAvg with FedProx. We agree that these are important baselines to consider, so we have evaluated all four of these baselines in the experimental settings of the main paper. The results are thoroughly discussed in the general response, and the loss curves are shown in Figures 1 and 2, but we highlight the main findings here.
>
> For both FashionMNIST and CIFAR-10, **Amplified SCAFFOLD achieves the best training loss and testing accuracy among all baselines, including the four new baselines.** In general, FedAdam and FedYogi were competitive, outperforming every algorithm except for Amplified SCAFFOLD for CIFAR-10, and every algorithm except for Amplified SCAFFOLD and SCAFFOLD for FashionMNIST. Ultimately these algorithms fell short of Amplified SCAFFOLD, which is consistent with the fact that these baselines were designed for i.i.d. client participation, whereas Amplified SCAFFOLD has convergence guarantees under periodic participation.
>
> In response to your question about combining Amplified FedAvg with FedProx, we note that Amplified FedProx did not perform better than Amplified FedAvg. This means that combining amplified updates with control variates is much more effective than combining amplified updates with FedProx regularization.
>
> Please see the general response for a detailed discussion of this additional experiment.
>
> 7\. **Experiments with different participation strategies.** In response to your question about different participation strategies, we have included an additional experiment under a new non-i.i.d. participation pattern. We refer to this pattern as Stochastic Cyclic Availability (SCA), and we believe that it captures user device availability which is both periodic and unreliable. We thoroughly discussed the details of this participation pattern and the results of the experiment in the general response, and we highlight the results below. Figure 3 of the 1-page PDF shows the loss curves for all algorithms (including the four additional baselines) for CIFAR-10 under SCA participation.
>
> From Figure 3, we conclude that **Amplified SCAFFOLD outperforms all baselines under a second non-i.i.d. participation pattern**. Amplified SCAFFOLD reaches the lowest training loss and highest training accuracy out of all algorithms, and even reaches a slightly lower training loss under SCA participation than cyclic participation (shown in Figure 2). We believe that this experiment bolsters the empirical validation of our proposed algorithm, demonstrating that Amplified SCAFFOLD performs well under multiple non-i.i.d. participation patterns.
>
> Please see the general response for the full details of this experiment.

---

> ### Author Response · Authors · 2024-08-11
>
> Thank you again for your efforst in the review process. We responded to your questions and concerns in our rebuttal. Specifically, we addressed your request for additional experiments (#5-7) and discussed the heterogeneity assumption (#4). Please let us know if we have addressed your concerns, and if you have any more questions. We are happy to continue discussion.
>
> Best,
> Authors

---

> > ### Comment · Reviewer_ewwQ · 2024-08-12
> >
> > Thank you for your response; my concerns have been addressed. I want to keep my score.

---

> > > ### Author Response · Authors · 2024-08-12
> > >
> > > Thank you for the feedback. Do you have any additional concerns which limit the paper from receiving a higher score? If not, we kindly suggest that you consider raising the score, since we have fulfilled the request for more experiments and answered all questions. Thank you.
> > >
> > > Best,
> > > Authors

---

> > > > ### Comment · Reviewer_ewwQ · 2024-08-13
> > > >
> > > > There are a couple of reasons why I did not increase my score. Firstly, I am not sure about the results of Amplified Fedprox for the Fashion-MNIST dataset. Can you please describe the algorithm that you used for Amplified Fedprox? The reason for my concern is that a standard setting [1] shows that the test accuracy is close. Secondly, I understand that the SCAFFOLD did not use the data heterogeneity assumption, but how did they or the authors avoid using it?
> > > >
> > > >
> > > >
> > > > [1] Li, Qinbin, et al. "Federated learning on non-iid data silos: An experimental study." 2022 IEEE 38th international conference on data engineering (ICDE). IEEE, 2022.

---

> ### Author Response · Authors · 2024-08-13
>
> **Amplified FedProx**: Our implementation of Amplified FedProx is obtained by starting from the implementation of Amplified FedAvg [(Wang & Ji, 2022)](https://arxiv.org/abs/2205.13648), and adding the FedProx regularization term $\mu/2 \lVert x - x_r \rVert^2$ to the objective of each local client, where $x_r$ denotes the global model from the last synchronization step. Table III from [1] shows that FedAvg and FedProx perform very closely in most settings without systems heterogeneity, and this is consistent with our results: The curves for FedProx are almost overlapping with those of FedAvg, and the curves for Amplified FedProx are almost overlapping with those of Amplified FedAvg. We believe that the experimental results of [1] are consistent with ours.
>
> **Avoiding heterogeneity assumption**: The essential technique that allows SCAFFOLD (and Amplified SCAFFOLD) to avoid the heterogeneity assumption is to modify the direction of each local update to approximate an update on the global gradient: we show some details below.
>
> The standard analysis of FedAvg wants to show descent of the global objective $f$, but the update directions depend on $\nabla f_i$ (i.e. the local gradients), so the rate of descent will be slowed down by the difference between $\nabla f$ and $\nabla f_i$. More concretely, the analysis involves the difference between the global gradient $\nabla f(x_{r,k}^i)$ and the update direction $\nabla f_i(x_{r,k}^i)$, which is: $\lVert \nabla f_i(x_{r,k}^i) - \nabla f(x_{r,k}^i) \rVert$. **Bounding this term requires the heterogeneity assumption**, but this might not be necessary if we change the update direction...
>
> For SCAFFOLD, the update direction is changed to $\nabla f_i(x_{r,k}^i) - G_r^i + G_r$, where $G_r^i$ is the average of stochastic gradients encountered by client $i$ during the round before $r$, and $G_r$ is the average of $G_r^i$ over $i$. Therefore, we can bound $\lVert G_r^i - \nabla f_i(x_{r,k}^i) \Vert$ and $\lVert G_r - \nabla f(x_{r,k}^i) \rVert$ **by smoothness alone, without requiring any additional assumptions**. This is very helpful for the convergence analysis for the following reason: when we have to compare the descent direction $\nabla f(x_{r,k}^i)$ against the local update direction $\nabla f_i(x_{r,k}^i) - G_r^i + G_r$, we can use the triangle inequality to bound $$
>     \lVert (\nabla f_i(x_{r,k}^i) - G_r^i + G_r) - \nabla f(x_{r,k}^i) \rVert \leq \lVert \nabla f_i(x_{r,k}^i) - G_r^i \rVert + \lVert \nabla f(x_{r,k}^i) - G_r \rVert.
> $$ Again, both of the terms on the RHS can be bounded with smoothness alone. Therefore, **the difference between the global gradient and the update direction can be bounded only with smoothness: no heterogeneity assumption necessary**.
>
> Please note that the above analysis omits certain details for the sake of brevity, but the key idea is there: control variates eliminate the need for bounded gradient dissimilarity. We use a similar essential idea in our analysis, but the proof is technically more complicated due to the added difficulty of non-i.i.d. participation. Please let us know if you have further questions on this topic.

---

> > ### Comment · Reviewer_ewwQ · 2024-08-13
> >
> > Thank you for the response. I understand that FedAvg and Fedprox perform very closely without system heterogeneity, but according to [1], SCAFFOLD also performs very closely in settings without system heterogeneity. I am not convinced about this gap in Amplified SCAFFOLD and Amplified Fedprox for the FMNIST dataset. I am sorry if I am missing something.

---

> > > ### Author Response · Authors · 2024-08-13
> > >
> > > You're right, thank you for clarifying your question: the results in [1] do show that SCAFFOLD is close to FedAvg and FedProx. The discrepancy between [1] and our results is likely due to differences in neural network architectures, due to the fact that the performance of SCAFFOLD seems to be very sensitive to network depth [(Yu et al, 2022)](https://arxiv.org/abs/2207.06343). Indeed, Table 3 of the original SCAFFOLD paper [(Karimireddy et al, 2019)](https://arxiv.org/abs/1910.06378) shows that SCAFFOLD outperforms FedAvg and FedProx when training a logistic regression model (1 layer), while [(Yu et al, 2022)](https://arxiv.org/abs/2207.06343) showed that SCAFFOLD can struggle even with 4 layer networks. Our Fashion-MNIST results you refer to are using a logistic regression model, so our results are consistent with Table 3 of [(Karimireddy et al, 2019)](https://arxiv.org/abs/1910.06378). Notice that in our CIFAR-10 experiments, we use a two layer NN and already SCAFFOLD not much better than FedAvg and FedProx, which is consistent with Table III of [1], while Amplified SCAFFOLD maintains a significant advantange.
> > >
> > > Thank you for the thought-provoking question! We hope that this clears up our experimental results in the context of the literature.

---

> > > > ### Comment · Reviewer_ewwQ · 2024-08-13
> > > >
> > > > Thank you for the clarification. I am going to raise my score accordingly.

---

### Official Review · Reviewer_3Kpj · 2024-07-28

**Soundness:** 3
**Presentation:** 2
**Contribution:** 2
**Rating:** 5
**Confidence:** 3

**Summary:**

In this paper, the authors examine realistic participation scenarios, including cyclic client participation and arbitrary participation patterns. They focus on a non-convex optimization setting, which is common in practical applications but challenging to address. To tackle this, they introduce a novel method called Amplified Scaffold, designed to effectively correct client drift—a common issue where clients' updates diverge from the global model. The Amplified Scaffold method builds upon existing techniques but enhances their effectiveness in this more complex setting. The authors not only provide theoretical convergence guarantees, ensuring that their method is mathematically sound, but also back up their claims with extensive experimental results. These experiments demonstrate the practical effectiveness of Amplified Scaffold in various scenarios, highlighting its potential for real-world applications.

**Strengths:**

In this work, the authors provide a detailed convergence analysis of a modified version of the well-known Scaffold method. While the theoretical framework appears to be sound, I did not thoroughly review all the proofs, so there may be some oversights. Nonetheless, the results obtained seem normal, adequate, and align with expectations.

The authors consider a wide range of settings and sampling schemes, which makes their work relevant to a broad audience, including both researchers and practitioners. This comprehensive approach addresses a significant problem in the field, highlighting its importance for real-world applications.

Additionally, the proposed method undergoes extensive experimental testing. These experiments validate the method's effectiveness and demonstrate its practical applicability in various scenarios. The combination of theoretical analysis and experimental validation strengthens the credibility and utility of the proposed approach.

**Weaknesses:**

1) The related work section in your paper is limited and could be significantly strengthened by including discussions of several closely related methods. For instance, a closely related method to Scaffold is ProxSkip (also referred to as Scaffnew), which was proposed as an improved version of Scaffold. Additionally, the Tamuna method, which incorporates partial participation, is relevant. Including a discussion about the connection to these methods will provide a broader context and highlight the advancements in the field.

- Mishchenko, Konstantin, et al. "Proxskip: Yes! Local gradient steps provably lead to communication acceleration! Finally!" International Conference on Machine Learning. PMLR, 2022.
- Condat, Laurent Pierre, Grigory Malinovsky, and Peter Richtárik. "Tamuna: Accelerated federated learning with local training and partial participation." (2023).

Furthermore, an arbitrary sampling framework has been studied in several recent papers. It would be beneficial to compare the assumptions on the sampling schemes used in these studies with those in your work. This will help clarify the differences and similarities, as well as the advantages and limitations of the various approaches.

- Tyurin, Alexander, et al. "Sharper Rates and Flexible Framework for Nonconvex SGD with Client and Data Sampling." Transactions on Machine Learning Research.
- Grudzień, Michał, Grigory Malinovsky, and Peter Richtárik. "Improving Accelerated Federated Learning with Compression and Importance Sampling." Federated Learning and Analytics in Practice: Algorithms, Systems, Applications, and Opportunities.

Additionally, it is worth discussing the paper that addresses optimal complexity for the non-convex federated learning setting. This paper provides insights into achieving optimal communication complexity in distributed non-convex optimization, which is highly relevant to your study.

- Patel, Kumar Kshitij, et al. "Towards optimal communication complexity in distributed non-convex optimization." Advances in Neural Information Processing Systems 35 (2022): 13316-13328.

By incorporating these discussions, your related work section will be more comprehensive and provide a clearer understanding of how your proposed method fits within the broader landscape of federated learning research. This will not only strengthen your paper but also demonstrate the relevance and impact of your contributions.

2) The major issue in your paper is that in Table 1, the comparison between methods and data heterogeneity is defined using the condition $ \sup_x \left\Vert\nabla f_i(x) - \nabla f(x)\right\Vert \leq \kappa $, which implies uniformly bounded heterogeneity. However, in the original Scaffold paper, the authors used a different measure for bounded gradient dissimilarity:
$$ \frac{1}{N} \sum_{i=1}^N \left\Vert \nabla f_i(x) \right\Vert^2 \leq G^2 + B^2 \left\Vert \nabla f(x) \right\Vert^2, \forall x. $$

This discrepancy is significant because uniformly bounded heterogeneity is a more restrictive assumption compared to bounded gradient dissimilarity. The latter is a more common and practical assumption in federated learning literature, as it better captures the variations in real-world data distributions.

I strongly recommend revising Table 1 to use the less restrictive assumption of bounded gradient dissimilarity instead of uniformly bounded heterogeneity. This will make your comparisons more relevant and applicable to a broader range of scenarios.

Additionally, the notation $\kappa$ used for bounded heterogeneity is potentially confusing. Typically, $\kappa$ is used to denote the condition number, $\kappa = \frac{L}{\mu}$, in the context of strongly convex functions. To avoid this confusion, I suggest changing the notation for bounded heterogeneity to something more standard and distinct, which will make your paper clearer and more consistent with established conventions in the field.

3) Assumption 2 is quite complex and may be challenging for readers to fully grasp. I recommend providing a more detailed discussion of the class of sampling schemes considered under this assumption. Specifically, elaborating on the types of sampling schemes included, their characteristics, and why they are relevant to your study will help clarify the underlying assumptions. This additional context will make it easier for readers to understand the scope and implications of your assumptions, thereby enhancing the clarity and impact of your paper.

4) In the experimental section, the rationale behind the choice of hyperparameters is not clearly explained. To improve clarity, I recommend providing a detailed explanation of the specific hyperparameters selected for your experiments. This should include the criteria used to choose these values, how they were tuned, and any relevant considerations or trade-offs. Offering this additional context will help readers understand why particular hyperparameter settings were used and how they influence the outcomes of your experiments. This level of detail will enhance the reproducibility of your work and provide valuable insights for others seeking to build upon your research.

5) The main issue with the theoretical results is that they do not generalize the Scaffold method in the i.i.d. setting. For Scaffold, the convergence rate is $ \frac{\Delta L}{\varepsilon^2} \left(\frac{N}{S}\right)^{\frac{2}{3}} $, while for Amplified Scaffold, the rate is $\frac{\Delta L}{\varepsilon^2} \left(\frac{N}{S}\right) $. This discrepancy means that Amplified Scaffold does not recover the more favorable convergence rate of the original Scaffold method under i.i.d. conditions.

It would be a significant improvement if Amplified Scaffold could match the original Scaffold's rate in the i.i.d. setting. Demonstrating this equivalence would not only strengthen the theoretical results but also enhance the practical applicability of Amplified Scaffold. If you can achieve this and show that Amplified Scaffold can indeed recover the original Scaffold's rate in the i.i.d. setting, I am willing to increase the score. This adjustment would highlight the robustness and efficiency of the Amplified Scaffold method, making it a more compelling contribution to the field.

**Questions:**

Please check the Weaknesses section.

**Limitations:**

The limitations section is well-written, but it should also note that the Amplified Scaffold method does not recover the rate of the original Scaffold method in the i.i.d. setting.

---

> ### Author Rebuttal · Authors · 2024-08-07
>
> Thank you for your helpful comments on our paper. Below we have responded to your questions and concerns.
>
> Weaknesses:
>
> 1. **Missing related work.** Thank you for pointing out these works. We agree that including these works in the discussion of our paper will establish a broader context of the FL literature, especially around problems of client sampling. Below we've included a brief comparison of each work you listed against our submission:
>
>     (a) Mischenko et al, 2022 (ProxSkip/ScaffNew): ScaffNew achieves reduced communication rounds for FL with full participation on strongly convex problems.
>
>     (b) Condat el al, 2023 (Tamuna): Tamuna uses communication compression and local steps together to reduce communication in FL with i.i.d. participation on strongly convex problems.
>
>     (c) Tyurin et al, 2023: Improves convergence rate of nonconvex SGD for finite sum problems in terms of the smoothness constants, based on various unbiased data sampling protocols. Includes an application to FL with various client sampling protocols, which is related to our motivating problem of non-i.i.d. client sampling. These client sampling protocols are required to be unbiased.
>
>     (d) Grudzien et al, 2023: Combines local training, client sampling, and communication compression to accelerate convergence in terms of condition number and dimension (strongly convex problems). Similarly to [Tyurin et al, 2023], the client sampling schemes from this paper are required to be unbiased.
>
>     (e) Patel et al, 2022: This paper provides lower bounds for distributed non-convex, stochastic, smooth optimization with intermittent communication, both in the full and partial participation settings. They also include algorithms employing variance reduction which match (or closely match) lower bounds in the full and partial participation settings. These lower bounds are generally important to understand the limits of optimization under i.i.d. client sampling.
>
>     In summary, these works are all relevant for FL in general. With respect to client sampling, (a) applies to full participation, (b) and (e) apply for i.i.d. client sampling, and (c) and (d) apply to unbiased sampling. The analysis of these papers cannot be directly applied to our setting of periodic participation, and our work aims to fill this gap by providing an algorithm with non-convex convergence guarantees even with non-i.i.d. client participation. We will cite and discuss these works in our updated paper.
>
> 2. **Different heterogeneity assumptions.** We addressed this point in the general response, but we can further touch on it here. The main point is that we do not use any heterogeneity assumption in our paper. The only reason that the condition $\lVert \nabla f_i(x) - \nabla f(x) \rVert \leq \kappa$ is stated in our table is that this assumption is used by the baselines we compare in Table 1. Our analysis avoids any heterogeneity assumption because our use of control variates eliminates the need for one (this is also the case in the SCAFFOLD analysis). We agree that bounded gradient dissimilarity is less restrictive than uniformly bounded heterogeneity, but even less restrictive is no heterogeneity assumption at all!
>
> 3. **Complexity of assumption 2.** We agree that Assumption 2 is somewhat dense, although we have already included in Section 3.2 a detailed discussion of different sampling schemes which satisfy assumption 2. If you have additional suggestions to improve the clarity of this assumption, please let us know.
>
> 4. **Explanation of hyperparameters.** As stated on Line 269, all of the hyperparameters for every baseline are tuned with grid search. The search ranges and final values for each hyperparameter are given in Table 2 of Appendix D. To evaluate each hyperparameter combination in the grid, we run the training setup described in the main body (with only one random seed instead of five) and choose the hyperparameter combination that reaches the smallest training loss by the end of training. After each hyperparameter is tuned, we evaluated each algorithm over five random seeds.
>
> 5. **Recovering SCAFFOLD's rate.** You are correct that our convergence rate as a slightly worse dependence on $N/S$ than SCAFFOLD. We pointed this out in Section 4.3 (line 223), and we provide a detailed explanation of this discrepancy in Appendix E. Essentially, there is a potential small issue in the analysis of SCAFFOLD for the case of partial participation. Instead of repeating this issue in our analysis, we accept a slightly worse dependence on $N/S$. It is likely that the issue can be fixed to recover the $(N/S)^{2/3}$ rate for both their algorithm and ours, but this is not the focus of our paper. In this paper, we focused on achieving convergence under periodic participation with reduced communication, linear speedup, and resilience to heterogeneity, and by doing so we have improved over previous work. If the $N/S$ dependence is an important issue to you, please read our discussion of the details, which is 1 page in Appendix E (lines 807-829).

---

> > ### Author Response · Authors · 2024-08-11
> >
> > Thank you again for reviewing our paper. In our rebuttal, we responded to the points from your review. Of particular importance is our discussion of the different heterogeneity assumptions (#2) and a comparison with SCAFFOLD's rate in the i.i.d. case (#5). Also, we included a discussion of new experimental results in the general response. Please let us know if we have addressed your concerns. We are happy to continue discussion.
> >
> > Best, Authors

---

> ### Comment · Reviewer_3Kpj · 2024-08-12
> **Response to authors**
>
> Thank you for your rebuttal!
>
> You have addressed my concerns, and as a result, I will be increasing my score accordingly.
>
> Let me clarify the concerns I had and highlight the aspects that addressed them:
>
> 1) Thank you for providing a discussion of related work. This aspect is resolved.
>
> 2) Let me clarify the aspect of the heterogeneity assumption. I agree that in your results, you do not use the heterogeneity assumption. This is straightforward since you use a client drift reduction mechanism, such as SCAFFOLD. The idea behind the SCAFFOLD mechanism is to allow a method to work without the heterogeneity assumption, and this is clear.
>
> However, I must mention that when you provide information about previous results, you must be precise in describing the details and assumptions made in those results. This helps prevent confusion for readers, as they are not expected to read all the papers in the field. I requested clarification on the aspects related to the table.
>
> 3) Under Assumption 2, you briefly described what conditions (a), (b), and (c) mean, with basically one sentence for each. I suggest providing more detailed explanations. For example, why did you specifically use equal frequency in condition (b)? Additionally, there is no mention of condition (d). While I understand the meaning of this assumption, it is not easy to grasp with such a compressed explanation. In Section 3.2, you provide examples, which are valuable, but the examples do not fully explain the assumptions themselves.
>
> 4) Let me clarify this aspect once again to avoid any confusion. I understand that you used a grid search for step sizes. What I meant is that the selection of client sampling parameters—such as the number of groups, availability time, communication interval, and number of communication rounds—is not described. These could also be considered hyperparameters. To avoid confusion, let us refer to them as experimental setup parameters. All I asked for was some explanation of how these parameters were selected. In Section D.1, these parameters are simply stated without clarification (lines 756-762).
>
> 5) I did indeed miss the explanation in Appendix E. Thank you for pointing this out. This clarification is indeed valuable. I recommend mentioning it earlier in the text (not at the end of the entire paper).
>
> Since aspects 1 and 5 are fully resolved, I have increased my score.

---

### Author Rebuttal · Authors · 2024-08-07

Thank you to all of the reviewers for your time and effort in the review process. Here we describe additional experimental results that we have added to address the reviewer comments, and give answers to common questions. We have also responded individually to each review below.

1. **Experiments with additional baselines.** For the FashionMNIST and CIFAR-10 experiments from the original paper, we have evaluated four additional baselines: FedAdam [28], FedYogi [28], FedAvg-M [Cheng, 2023] (see citation below), and Amplified FedAvg + FedProx [34, 19]. The results are shown in Figures 1 and 2 of the 1-page PDF, where you can see that **Amplified SCAFFOLD maintains superior performance against all of the added baselines in terms of both training loss and testing accuracy.**

    We tuned the hyperparameters of all baselines according to the hyperparameter ranges suggested in the original paper of each algorithm, and we allow the same compute budget for tuning each baseline as we did for tuning the algorithms in the original paper, in terms of the total number of hyperparameter combinations evaluated. Also, the results are averaged over five random seeds.

    For FashionMNIST: FedAdam and FedYogi reach moderate training loss quickly, but are soon overtaken by Amplified SCAFFOLD and later by SCAFFOLD. FedAvg-M exhibits a minor advantage over FedAvg, but performs about the same as Amplified FedAvg. Amplified FedProx (i.e. Amplified FedAvg with FedProx regularization) performs nearly identically to Amplified FedAvg.

    For CIFAR-10: FedAdam is more competitive, but is still outperformed by Amplified SCAFFOLD. FedYogi and FedAvg-M are further behind, though both still outperform SCAFFOLD. Amplified FedProx is again nearly identical to Amplified FedAvg.

    These new experiments demonstrate that **Amplified SCAFFOLD outperforms strong empirical baselines (FedAdam, FedYogi) under cyclic client participation**, reinforcing the empirical validation of our algorithm. This performance is consistent with the fact that Amplified SCAFFOLD has convergence guarantees under periodic participation, while FedAdam and FedYogi were not designed for settings beyond i.i.d. client sampling.

2. **Experiments with different participation strategies.** We also added an evaluation under another non-i.i.d. participation pattern. Figure 3 of the 1-page PDF shows the evaluation of all baselines (including the additional baselines from \#1) for CIFAR-10 under this new participation pattern, which is described in the next paragraph. The results show that **Amplified SCAFFOLD outperforms all baselines under a second non-i.i.d. participation pattern**.

    We refer to this new pattern as Stochastic Cyclic Availability (SCA), and it models device availability which is both periodic and unreliable. Similarly to cyclic participation, the set of clients is divided into $\bar{K}$ groups, and at each round one group is deemed the "active" group, while the others are inactive. Unlike cyclic participation, in SCA not every client in the active group is always available: Instead, when a group becomes active, the clients in that group become available for sampling with probability $80\%$, while clients in inactive groups have probability $5\%$ to be available for participation. The active group changes every $g$ rounds. This stochastic availability models the real-life situation where a client device can be unavailable at a time of day when it is usually available, or vice versa. In this way, SCA is more flexible than cyclic participation and better captures the unreliability of client devices. Lastly, we reused the remaining settings ($g, \bar{K}, I$, etc.) and the tuned hyperparameters for each baseline from the CIFAR-10 experiment under cyclic participation. Again, we average each algorithm's performance over five random seeds.

    Results for CIFAR-10 under SCA participation are shown in Figure 3. Again, **Amplified SCAFFOLD outperforms all baselines under SCA participation**. The relative performance of each baseline is similar, with FedAdam staying competitive with Amplified SCAFFOLD, followed by FedYogi and FedAvg-M. The remainder of the baselines have significantly worse performance, and again Amplified FedAvg has not benefitted by adding FedProx regularization.

    This new experiment shows that **Amplified SCAFFOLD performs well in other non-i.i.d. participation patterns beyond cyclic participation.** We will include these new results in the updated version of the paper.

3. **Meaning of assumption 1(a).** Several reviewers pointed out that Assumption 1(a) is too strong or counterintuitive. This confusion is actually just due to a typo in the statement of Assumption 1(a). The correct version should say that $f(x_0) - \min_{x \in \mathbb{R}^d} f(x) \leq \Delta$, which is standard for convergence proofs.

4. **Heterogeneity assumptions.** Two reviewers asked whether the heterogeneity assumption $\lVert \nabla f_i(x) - \nabla f(x) \rVert \leq \kappa$ is used for our proof, and whether this assumption is reasonable in our setting. In short, our proof does not use this assumption or any assumption on the heterogeneity; as stated in Theorem 1, the only assumptions we need are Assumption 1, Assumption 2, and Equations (1) and (2) (both only depend on the client sampling distribution). The heterogeneity assumption with $\kappa$ is only stated in Table 1 because it is used by several baselines. Similarly to SCAFFOLD in the case of i.i.d. sampling, we can avoid this assumption through the use of control variates to approximate the global gradient at each local step.

[Cheng, 2023] Cheng, Ziheng, et al. "Momentum Benefits Non-iid Federated Learning Simply and Provably." The Twelfth International Conference on Learning Representations.

---

### Decision · Program_Chairs · 2024-09-25

**Decision:**

Accept (poster)

**Comment:**

The authors examine realistic partial participation scenarios in federated learning, including cyclic client participation and arbitrary client participation patterns. The focus is on the non-convex optimization setting which is common in practical applications but challenging to address. To tackle this, they introduce a novel method called Amplified Scaffold, designed to effectively correct client drift --- a common issue where clients' updates diverge from the global model. The Amplified Scaffold method builds upon existing techniques but enhances their effectiveness in this more complex setting. The authors not only provide theoretical convergence guarantees, ensuring that their method is mathematically sound, but also back up their claims with extensive experimental results. These experiments demonstrate the practical effectiveness of Amplified Scaffold in various scenarios, highlighting its potential for real-world applications.

All reviewers proposed the paper to be accepted, with scores 5, 6, 6, 6. The work was praised in several ways, including:
- the problem and method are well motivated
- the theoretical results obtained look adequate, and align with expectations
- the authors consider a wide range of settings and sampling schemes, which makes their work relevant to a broad audience
- extensive experimental testing
- the text is well written

The observed weaknesses were considered minor in comparison to the strengths, and some were addressed in the rebuttal (e.g., insufficient discussion of related work), and the previously lower scores were increased.